# GENERALIZED VARIATIONAL CONTINUAL LEARNING

**Noel Loo, Siddharth Swaroop & Richard E. Turner**
University of Cambridge
`{nl355,ss2163,ret26}@cam.ac.uk`

## ABSTRACT

Continual learning deals with training models on new tasks and datasets in an online fashion. One strand of research has used probabilistic regularization for continual learning, with two of the main approaches in this vein being Online Elastic Weight Consolidation (Online EWC) and Variational Continual Learning (VCL). VCL employs variational inference, which in other settings has been improved empirically by applying likelihood-tempering. We show that applying this modification to VCL recovers Online EWC as a limiting case, allowing for interpolation between the two approaches. We term the general algorithm Generalized VCL (GVCL). In order to mitigate the observed overpruning effect of VI, we take inspiration from a common multi-task architecture, neural networks with task-specific FiLM layers, and find that this addition leads to significant performance gains, specifically for variational methods. In the small-data regime, GVCL strongly outperforms existing baselines. In larger datasets, GVCL with FiLM layers outperforms or is competitive with existing baselines in terms of accuracy, whilst also providing significantly better calibration.

## 1 INTRODUCTION

Continual learning methods enable learning when a set of tasks changes over time. This topic is of practical interest as many real-world applications require models to be regularly updated as new data is collected or new tasks arise. Standard machine learning models and training procedures fail in these settings (French, 1999), so bespoke architectures and fitting procedures are required.

This paper makes two main contributions to continual learning for neural networks. First, we develop a new regularization-based approach to continual learning. Regularization approaches adapt parameters to new tasks while keeping them close to settings that are appropriate for old tasks. Two popular approaches of this type are Variational Continual Learning (VCL) (Nguyen et al., 2018) and Online Elastic Weight Consolidation (Online EWC) (Kirkpatrick et al., 2017; Schwarz et al., 2018). The former is based on a variational approximation of a neural network's posterior distribution over weights, while the latter uses Laplace's approximation. In this paper, we propose Generalized Variational Continual Learning (GVCL) of which VCL and Online EWC are two special cases. Under this unified framework, we are able to combine the strengths of both approaches. GVCL is closely related to likelihood-tempered Variational Inference (VI), which has been found to improve performance in standard learning settings (Zhang et al., 2018; Osawa et al., 2019). We also see significant performance improvements in continual learning.

Our second contribution is to introduce an architectural modification to the neural network that combats the deleterious overpruning effect of VI (Trippe & Turner, 2018; Turner & Sahani, 2011). We analyze pruning in VCL and show how task-specific FiLM layers mitigate it. Combining this architectural change with GVCL results in a hybrid architectural-regularization based algorithm. This additional modification results in performance that exceeds or is within statistical error of strong baselines such as HAT (Serra et al., 2018) and PathNet (Fernando et al., 2017).

The paper is organized as follows. Section 2 outlines the derivation of GVCL, shows how it unifies many continual learning algorithms, and describes why it might be expected to perform better than them. Section 3 introduces FiLM layers, first from the perspective of multi-task learning, and then through the lens of variational over-pruning, showing how FiLM layers mitigate this pathology of VCL. Finally, in Section 5 we test GVCL and GVCL with FiLM layers on many standard bench-

marks, including ones with few samples, a regime that could benefit more from continual learning. We find that GVCL with FiLM layers outperforms existing baselines on a variety of metrics, including raw accuracy, forwards and backwards transfer, and calibration error. In Section 5.4 we show that FiLM layers provide a disproportionate improvement to variational methods, confirming our hypothesis in Section 3[1].

## 2 GENERALIZED VARIATIONAL CONTINUAL LEARNING

In this section, we introduce Generalized Variational Continual Learning (GVCL) as a likelihood-tempered version of VCL, with further details in Appendix C. We show how GVCL recovers Online EWC. We also discuss further links between GVCL and the Bayesian cold posterior in Appendix D.

### 2.1 LIKELIHOOD-TEMPERING IN VARIATIONAL CONTINUAL LEARNING

**Variational Continual Learning (VCL).** Bayes' rule calculates a posterior distribution over model parameters $\theta$ based on a prior distribution $p(\theta)$ and some dataset $D_T = \{X_T, y_T\}$. Bayes' rule naturally supports online and continual learning by using the previous posterior $p(\theta|D_{T-1})$ as a new prior when seeing new data (Nguyen et al., 2018). Due to the intractability of Bayes' rule in complicated models such as neural networks, approximations are employed, and VCL (Nguyen et al., 2018) uses one such approximation, Variational Inference (VI). This approximation is based on approximating the posterior $p(\theta|D_T)$ with a simpler distribution $q_T(\theta)$, such as a Gaussian. This is achieved by optimizing the ELBO for the optimal $q_T(\theta)$,

$$\text{ELBO}_{\text{VCL}} = \mathbb{E}_{\theta \sim q_T(\theta)}[\log p(D_T|\theta)] - D_{\text{KL}}(q_T(\theta)\|q_{T-1}(\theta)), \tag{1}$$

where $q_{T-1}(\theta)$ is the approximation to the previous task posterior. Intuitively, this refines a distribution over weight samples that balances good predictive performance (the first expected prediction accuracy term) while remaining close to the prior (the second KL-divergence regularization term).

**Likelihood-tempered VCL.** Optimizing the ELBO will recover the true posterior if the approximating family is sufficiently rich. However, the simple families used in practice typically lead to poor test-set performance. Practitioners have found that performance can be improved by down-weighting the KL-divergence regularization term by a factor $\beta$, with $0 < \beta < 1$. Examples of this are seen in Zhang et al. (2018) and Osawa et al. (2019), where the latter uses a "data augmentation factor" for down-weighting. In a similar vein, sampling from "cold posteriors" in SG-MCMC has also been shown to outperform the standard Bayes posterior, where the cold posterior is given by $p_T(\theta|D) \propto p(\theta|D)^{\frac{1}{T}}, T < 1$ (Wenzel et al., 2020). Values of $\beta > 1$ have also been used to improve the disentanglement variational autoencoder learned models (Higgins et al., 2017). We down-weight the KL-divergence term in VCL, optimizing the $\beta$-ELBO[2],

$$\beta\text{-ELBO} = \mathbb{E}_{\theta \sim q_T(\theta)}[\log p(D_T|\theta)] - \beta D_{\text{KL}}(q_T(\theta)\|q_{T-1}(\theta)).$$

VCL is trivially recovered when $\beta = 1$. We will now show that surprisingly as $\beta \to 0$, we recover a special case of Online EWC. Then, by modifying the term further as required to recover the full version of Online EWC, we will arrive at our algorithm, Generalized VCL.

### 2.2 ONLINE EWC IS A SPECIAL CASE OF GVCL

We analyze the effect of KL-reweighting on VCL in the case where the approximating family is restricted to Gaussian distributions over $\theta$. We will consider training all the tasks with a KL-reweighting factor of $\beta$, and then take the limit $\beta \to 0$, recovering Online EWC. Let the approximate posteriors at the previous and current tasks be denoted as $q_{T-1}(\theta) = \mathcal{N}(\theta; \mu_{T-1}, \Sigma_{T-1})$ and $q_T(\theta) = \mathcal{N}(\theta; \mu_T, \Sigma_T)$ respectively, where we are learning $\{\mu_T, \Sigma_T\}$. The optimal $\Sigma_T$ under the $\beta$-ELBO has the form (see Appendix C),

$$\Sigma_T^{-1} = \frac{1}{\beta}\nabla_{\mu_T}\nabla_{\mu_T}\mathbb{E}_{q_T(\theta)}[-\log p(D_T|\theta)] + \Sigma_{T-1}^{-1}. \tag{2}$$

---

[1]Code is available at `https://github.com/yolky/gvcl`
[2]We slightly abuse notation by writing the likelihood as $p(D_T|\theta)$ instead of $p(y_T|\theta, X_T)$.

Now take the limit $\beta \to 0$. From Equation 2, $\Sigma_T \to 0$, so $q_T(\theta)$ becomes a delta function, and

$$\Sigma_T^{-1} = -\frac{1}{\beta}\nabla_{\mu_T}\nabla_{\mu_T}\log p(D_T|\theta = \mu_T) + \Sigma_{T-1}^{-1} = \frac{1}{\beta}H_T + \Sigma_{T-1}^{-1} = \frac{1}{\beta}\sum_{t=1}^{T}H_t + \Sigma_0^{-1}, \quad (3)$$

where $H_T$ is the $T$th task Hessian[3]. Although the learnt distribution $q_T(\theta)$ becomes a delta function (and not a full Gaussian distribution as in Laplace's approximation), we will see that a cancellation of $\beta$ factors in the $\beta$-ELBO will lead to the eventual equivalence between GVCL and Online EWC. Consider the terms in the $\beta$-ELBO that only involve $\mu_T$:

$$\beta\text{-ELBO} = \mathbb{E}_{\theta \sim q_T(\theta)}[\log p(D_T|\theta)] - \frac{\beta}{2}(\mu_T - \mu_{T-1})^\top \Sigma_{T-1}^{-1}(\mu_T - \mu_{T-1})$$

$$= \log p(D_T|\theta = \mu_T) - \frac{1}{2}(\mu_T - \mu_{T-1})^\top \left(\sum_{t=1}^{T-1}H_t + \beta\Sigma_0^{-1}\right)(\mu_T - \mu_{T-1}), \quad (4)$$

where we have set the form of $\Sigma_{T-1}$ to be as in Equation 3. Equation 4 is an instance of the objective function used by a number of continual learning methods, most notably Online EWC[4] (Kirkpatrick et al., 2017; Schwarz et al., 2018), Online-Structured Laplace (Ritter et al., 2018), and SOLA (Yin et al., 2020). These algorithms can be recovered by changing the approximate posterior class $\mathcal{Q}$ to Gaussians with diagonal, block-diagonal Kronecker-factored covariance matrices, and low-rank precision matrices, respectively (see Appendices C.4 and C.5).

Based on this analysis, we see that $\beta$ can be seen as interpolating between VCL, with $\beta = 1$, and continual learning algorithms which use point-wise approximations of curvature as $\beta \to 0$. In Appendix A we explore how $\beta$ controls the scale of the quadratic curvature approximation, verifying with experiments on a toy dataset.. Small $\beta$ values learn distributions with good *local* structure, while higher $\beta$ values learn distributions with a more *global* structure. We explore this in more detail in Appendices A and B, where we show the convergence of GVCL to Online-EWC on a toy experiment.

**Inference using GVCL.** When performing inference with GVCL at test time, we use samples from the unmodified $q(\theta)$ distribution. This means that when $\beta = 1$, we recover the VCL predictive, and as $\beta \to 0$, the posterior collapses as described earlier, meaning that the weight samples are effectively deterministic. This is in line with the inference procedure given by Online EWC and its variants. In practice, we use values of $\beta = 0.05 - 0.2$ in Section 5, meaning that some uncertainty is retained, but not all. We can increase the uncertainty at inference time by using an additional tempering step, which we describe, along with further generalizations in Appendix D.

## 2.3 REINTERPRETING $\lambda$ AS COLD POSTERIOR REGULARIZATION

As described above, the $\beta$-ELBO recovers instances of a number of existing second-order continual learning algorithms including Online EWC as special cases. However, the correspondence does not recover a key hyperparameter $\lambda$ used by these methods that up-weights the quadratic regularization term. Instead, our derivation produces an implicit value of $\lambda = 1$, i.e. equal weight between tasks of equal sample count. In practice it is found that algorithms such as Online EWC perform best when $\lambda > 1$, typically $10 - 1000$. In this section, we view this $\lambda$ hyperparameter as a form of cold posterior regularization.

In the previous section, we showed that $\beta$ controls the length-scale over which we approximate the *curvature* of the posterior. However, the *magnitude* of the quadratic regularizer stays the same, because the $O(\beta^{-1})$ precision matrix and the $\beta$ coefficient in front of the KL-term cancel out. Taking inspiration from cold posteriors (Wenzel et al., 2020), which temper both the likelihood and the prior and improve accuracy with Bayesian neural networks, we suggest tempering the prior in GVCL.

Therefore, rather than measuring the KL divergence between the posterior and prior, $q_T$ and $q_{T-1}$, respectively, we suggest regularizing towards *tempered* version of the prior, $q_{T-1}^\lambda$. However, this

---

[3]The actual Hessian may not be positive semidefinite while $\Sigma$ is, so here we refer to a positive semidefinite approximation of the Hessian.

[4]EWC uses the Fisher information, but our derivation results in the Hessian. The two matrices coincide when the model has near-zero training loss, as is often the case (Martens, 2020).

form of regularization has a problem: in continual learning, over the course of many tasks, old tasks will be increasingly (exponentially) tempered. In order to combat this, we also use the tempered version of the posterior in the KL divergence, $q_T^\lambda$. This should allow us to gain benefits from tempering the prior while being stable over multiple tasks in continual learning.

As we now show, tempering in this way recovers the $\lambda$ hyperparameter from algorithms such as Online EWC. Note that raising the distributions to the power $\lambda$ is equivalent to tempering by $\tau = \lambda^{-1}$. For Gaussians, tempering a distribution by a temperature $\tau = \lambda^{-1}$ is the same as scaling the covariance by $\lambda^{-1}$. We can therefore expand our new KL divergence,

$$
\begin{aligned}
D_{\mathrm{KL}}\left(q_T^\lambda \| q_{T-1}^\lambda\right) &= \tfrac{1}{2}\big((\mu_T - \mu_{T-1})^\top \lambda \Sigma_{T-1}^{-1}(\mu_T - \mu_{T-1}) + \mathrm{Tr}(\lambda \Sigma_{T-1}^{-1} \lambda^{-1} \Sigma_T) + \log \tfrac{|\Sigma_{T-1}|\lambda^{-d}}{|\Sigma_T|\lambda^{-d}} - d\big) \\
&= \tfrac{1}{2}\big((\mu_T - \mu_{T-1})^\top \lambda \Sigma_{T-1}^{-1}(\mu_T - \mu_{T-1}) + \mathrm{Tr}(\Sigma_{T-1}^{-1}\Sigma_T) + \log \tfrac{|\Sigma_{T-1}|}{|\Sigma_T|} - d\big) \\
&= D_{\mathrm{KL}\lambda}(q_T \| q_{T-1}).
\end{aligned}
$$

In the limit of $\beta \to 0$, our $\lambda$ coincides with Online EWC's $\lambda$, if the tasks have the same number of samples. However, this form of $\lambda$ has a slight problem: it increases the regularization strength of the initial prior $\Sigma_0$ on the mean parameter update. We empirically found that this negatively affects performance. We therefore propose a different version of $\lambda$, which only up-weights the "data-dependent" parts of $\Sigma_{T-1}$, which can be viewed as likelihood tempering the previous task posterior, as opposed to tempering both the initial prior and likelihood components. This new version still converges to Online EWC as $\beta \to 0$, since the $O(1)$ prior becomes negligible compared to the $O(\beta^{-1})$ Hessian terms. We define,

$$
\tilde{\Sigma}_{T,\lambda}^{-1} := \frac{\lambda}{\beta} \sum_{t=1}^{T} H_t + \Sigma_0^{-1} = \lambda(\Sigma_T^{-1} - \Sigma_0^{-1}) + \Sigma_0^{-1}.
$$

In practice, it is necessary to clip negative values of $\Sigma_T^{-1} - \Sigma_0^{-1}$ to keep $\tilde{\Sigma}_{T,\lambda}^{-1}$ positive definite. This is only required because of errors during optimization. We then use a modified KL-divergence,

$$
D_{\mathrm{KL}\tilde\lambda}(q_T \| q_{T-1}) = \tfrac{1}{2}\left((\mu_T - \mu_{T-1})^\top \tilde{\Sigma}_{T-1,\lambda}^{-1}(\mu_T - \mu_{T-1}) + \mathrm{Tr}(\Sigma_{T-1}^{-1}\Sigma_T) + \log \tfrac{|\Sigma_{T-1}|}{|\Sigma_T|} - d\right).
$$

Note that in Online EWC, there is another parameter $\gamma$, that down-weights the previous Fisher matrices. As shown in Appendix C, we can introduce this hyperparameter by taking the KL divergence priors and posteriors at different temperatures: $q_{T-1}^\lambda$ and $q_T^{\gamma\lambda}$. However, we do not find that this approach improves performance. Combining everything, we have our objective for GVCL,

$$
\mathbb{E}_{\theta \sim q_T(\theta)}[\log p(D_T|\theta)] - \beta D_{\mathrm{KL}\tilde\lambda}(q_T(\theta) \| q_{T-1}(\theta)).
$$

## 3 FiLM Layers for Continual Learning

The Generalized VCL algorithm proposed in Section 2 is applicable to any model. Here we discuss a multi-task neural network architecture that is especially well-suited to GVCL when the task ID is known at both training and inference time: neural networks with task-specific FiLM layers.

### 3.1 Background to FiLM Layers

The most common architecture for continual learning is the multi-headed neural network. A shared set of body parameters act as the feature extractor. For every task, features are generated in the same way, before finally being passed to separate head networks for each task. This architecture does not allow for task-specific differentiation in the feature extractor, which is limiting (consider, for example, the different tasks of handwritten digit recognition and image recognition). FiLM layers (Perez et al., 2018) address this limitation by linearly modulating features for each specific task so that useful features can be amplified and inappropriate ones ignored. In fully-connected layers, the transformation is applied element-wise: for a hidden layer with width $W$ and activation values $h_i$, $1 \le i \le W$, FiLM layers perform the transformation $h_i' = \gamma_i h_i + b_i$, before being passed on to the remainder of the network. For convolutional layers, transformations are applied filter-wise. Consider a layer with $N$ filters of size $K \times K$, resulting in activations $h_{i,j,k}$, $1 \le i \le N, 1 \le j \le W, 1 \le k \le H$, where $W$ and $H$ are the dimensions of the resulting feature map. The transformation has the

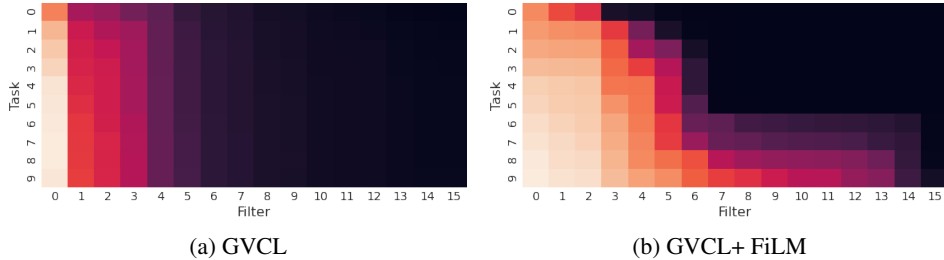

(a) GVCL                                      (b) GVCL+ FiLM

Figure 1: Visualizations of deviation from the prior distribution for filters in the first layer of a convolutional networks trained on Hard-CHASY. Lighter colours indicate an active filter for that task. Models are trained either (a) sequentially using GVCL, or (b) sequentially with GVCL + FiLM. FiLM layers increase the number of active units.

form $h'_{i,j,k} = \gamma_i * h_{i,j,k} + b_i$. The number of required parameters scales with the number of filters, as opposed to the full activation dimension, making them computationally cheap and parameter-efficient. FiLM layers have previously been shown to help with fine-tuning for transfer learning (Rebuffi et al., 2017), multi-task meta-learning (Requeima et al., 2019), and few-shot learning (Perez et al., 2018). In Appendix F, we show how FiLM layer parameters are interpretable, with similarities between FiLM layer parameters for similar tasks in a multi-task setup.

## 3.2 COMBINING GVCL AND FiLM LAYERS

It is simple to apply GVCL to models which utilize FiLM layers. Since these layers are specific to each task they do not need a distributional treatment or regularization as was necessary to support continual learning of the shared parameters. Instead, point estimates are found by optimising the GVCL objective function. This has a well-defined optimum unlike joint MAP training when FiLM layers are added (see Appendix E for a discussion). We might expect an improved performance for continual learning by introducing task-specific FiLM layers as this results in a more suitable multi-task model. However, when combined with GVCL, there is an additional benefit.

When applied to multi-head networks, VCL tends to prune out large parts of the network (Trippe & Turner, 2018; Turner & Sahani, 2011) and GVCL inherits this behaviour. This occurs in the following way: First, weights entering a node revert to their prior distribution due to the KL-regularization term in the ELBO. These weights then add noise to the network, affecting the likelihood term of the ELBO. To avoid this, the bias concentrates at a negative value so that the ReLU activation effectively shuts off the node. In the single task setting, this is often relatively benign and can even facilitate compression (Louizos et al., 2017; Molchanov et al., 2017). However, in continual learning the effect is pathological: the bias remains negative due to its low variance, meaning that the node is effectively shut off from that point forward, preventing the node from re-activating. Ultimately, large sections of the network can be shut off after the first task and cannot be used for future tasks, which wastes network capacity (see Figure 1a).

In contrast, when using task-specific FiLM layers, pruning can be achieved by either setting the FiLM layer scale to 0 or the FiLM layer bias to be negative. Since there is no KL-penalty on these parameters, it is optimal to prune in this way. Critically, both the incoming weights and the bias of a pruned node can then return to the prior without adding noise to the network, meaning that the node can be re-activated in later tasks. The increase in the number of unpruned units can be seen in Figure 1b. In Appendix G we provide more evidence of this mechanism.

## 4 RELATED WORK

**Regularization-based continual learning.** Many algorithms attempt to regularize network parameters based on a metric of importance. Section 2 shows how some methods can be seen as special cases of GVCL. We now focus on other related methods. Lee et al. (2017) proposed IMM, which is an extension to EWC which merges posteriors based on their Fisher information matrices. Ahn et al.

(2019), like us, use regularizers based on the ELBO, but also measure importance on a per-node basis rather than a per-weight one. SI (Zenke et al., 2017) measures importance using "Synaptic Saliency," as opposed to methods based on approximate curvature.

**Architectural approaches to continual learning.** This family of methods modifies the standard neural architecture by adding components to the network. Progressive Neural Networks (Rusu et al., 2016) adds a parallel column network for every task, growing the model size over time. PathNet (Fernando et al., 2017) fixes the model size while optimizing the paths between layer columns. Architectural approaches are often used in tandem with regularization based approaches, such as in HAT (Serra et al., 2018), which uses per-task gating parameters alongside a compression-based regularizer. Adel et al. (2020) propose CLAW, which also uses variational inference alongside per-task parameters, but requires a more complex meta-learning based training procedure involving multiple splits of the dataset. GVCL with FiLM layers adds to this list of hybrid architectural-regularization based approaches. See Appendix H for a more comprehensive related works section.

## 5 EXPERIMENTS

We run experiments in the small-data regime (Easy-CHASY and Hard-CHASY) (Section 5.1), on *Split*-MNIST (Section 5.1), on the larger Split CIFAR benchmark (Section 5.2), and on a much larger Mixed Vision benchmark consisting of 8 different image classification datasets (Section 5.3). In order to compare continual learning performance, we compare final average accuracy, forward transfer (the improvement on the current task as number of past tasks increases (Pan et al., 2020)) and backward transfer (the difference in accuracy between when a task is first trained and its accuracy after the final task (Lopez-Paz & Ranzato, 2017)). We compare to many baselines, but due to space constraints, only report the best-performing baselines in the main text. We also compare to two offline methods: an upper-bound "joint" version trained on all tasks jointly, and a lower-bound "separate" version with each task trained separately (no transfer). Further baseline results are in Appendix J. The combination of GVCL on task-specific FiLM layers (GVCL-F) outperforms baselines on the smaller-scale benchmarks and outperforms or performs within statistical error of baselines on the larger Mixed Vision benchmark. We also report calibration curves, showing that GVCL-F is well-calibrated. Full experimental protocol and hyperparameters are reported in Appendix I.

### 5.1 CHASY AND *Split*-MNIST

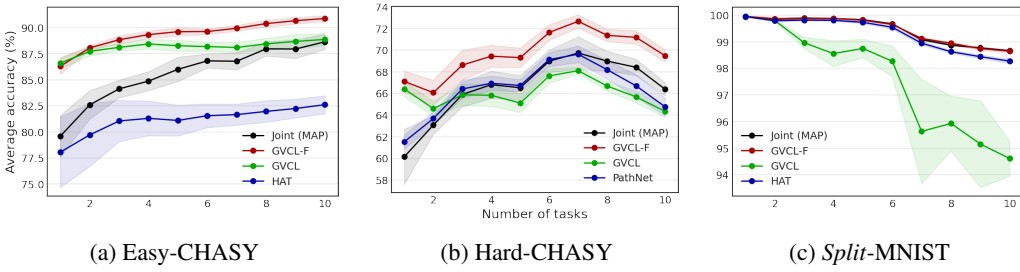

|                |                |                  |
| :------------: | :------------: | :--------------: |
| (a) Easy-CHASY | (b) Hard-CHASY | (c) *Split*-MNIST |

Figure 2: Running average accuracy of Easy-CHASY, Hard-CHASY and *Split*-MNIST trained continually. GVCL-F and GVCL are compared to the best performing baseline algorithm. GVCL-F and GVCL both significantly outperform HAT on Easy-CHASY. On Hard-CHASY, GVCL-F still manages to perform as well joint MAP training, while GVCL performs as well as PathNet. In Split-MNIST, GVCL-F narrowly outperforms HAT, with both performing nearly as well as joint training. The CHASY benchmark consists of a set of tasks specifically designed for multi-task and continual learning, with detailed explanation in Appendix K. It is derived from the HASYv2 dataset (Thoma, 2017), which consists of 32x32 handwritten latex characters. Easy-CHASY was designed to maximize transfer between tasks and consists of similar tasks with 20 classes for the first task, to 11 classes for the last. Hard-CHASY represents scenarios where tasks are very distinct, where tasks range from 18 to 10 classes. Both versions have very few samples per class. Testing our algorithm on these datasets tests two extremes of the continual learning spectrum. For these two datasets we use a small convolutional network comprising two convolutions layers and a fully connected layer.

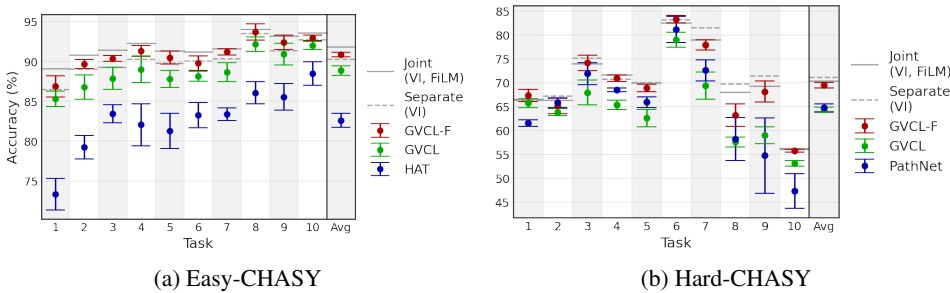

|                | (a) Easy-CHASY | (b) Hard-CHASY |
|---|---|---|

Figure 3: Accuracy of Easy-CHASY and Hard-CHASY trained models at the end of learning all 10 tasks continually. Performance of GVCL-F, GVCL and the best performing baselines (HAT and Pathnet) are compared to Joint and Separate training. GVCL-F again strongly outperforms the baselines and performs similar to the upper-bound VI joint training.

For our *Split*-MNIST experiment, in addition to the standard 5 binary classification tasks for *Split*-MNIST, we add 5 more binary classification tasks by taking characters from the KMNIST dataset (Clanuwat et al., 2018). For these experiments we used a 2-layer fully-connected network, as in common in continual learning literature (Nguyen et al., 2018; Zenke et al., 2017).

Figure 2 shows the raw accuracy results. As the CHASY datasets have very few samples per class (16 per class, resulting in the largest task having a training set of 320 samples), it is easy to overfit. This few-sample regime is a key practical use case for continual learning as it is essential to transfer information between tasks. In this regime, continual learning algorithms based on MAP-inference overfit, resulting in poor performance. As GVCL-F is based on a Bayesian framework, it is not as adversely affected by the low sample count, achieving 90.9% accuracy on Easy-CHASY compared to 82.6% of the best performing MAP-based CL algorithm, HAT. Hard-CHASY tells a similar story, 69.1% compared to PathNet's 64.8%. Compared to the full joint training baselines, GVCL-F achieves nearly the same accuracy (Figure 3). The gap between GVCL-F and GVCL is larger for Easy-CHASY than for Hard-CHASY, as the task-specific adaptation that FiLM layers provide is more beneficial when tasks require contrasting features, as in Hard-CHASY. With *Split*-MNIST, GVCL-F also reaches the same performance as joint training, however it is difficult to distinguish approaches on this benchmark as many achieve near maximal accuracy.

|  |  | GVCL-F | GVCL | HAT | PathNet | VCL | Online EWC |
|---|---|---|---|---|---|---|---|
| Easy-CHASY | ACC (%) | **90.9 ± 0.3** | 88.9 ± 0.6 | 82.6 ± 0.9 | 82.4 ± 0.9 | 78.4 ± 1.0 | 73.4 ± 3.4 |
|  | BWT (%) | **0.2 ± 0.1** | −0.8 ± 0.4 | −1.6 ± 0.6 | 0.0 ± 0.0 | −4.1 ± 1.2 | −8.9 ± 2.9 |
|  | FWT (%) | **0.4 ± 0.3** | −0.6 ± 0.5 | **0.4 ± 1.4** | −1.5 ± 0.9 | −7.9 ± 0.8 | −1.5 ± 0.5 |
| Hard-CHASY | ACC (%) | **69.5 ± 0.6** | 64.4 ± 0.6 | 62.5 ± 5.4 | 64.8 ± 0.8 | 45.8 ± 1.4 | 56.4 ± 1.7 |
|  | BWT (%) | **−0.1 ± 0.1** | −0.6 ± 0.2 | −0.8 ± 0.4 | **0.0 ± 0.0** | −11.9 ± 1.6 | −7.1 ± 1.7 |
|  | FWT (%) | **−1.6 ± 0.7** | −6.3 ± 0.6 | −3.7 ± 5.5 | −2.2 ± 0.8 | −13.5 ± 2.2 | −3.4 ± 1.3 |
| *Split*-MNIST (10 Tasks) | ACC (%) | **98.6 ± 0.1** | 94.6 ± 0.7 | 98.3 ± 0.1 | 95.2 ± 1.8 | 92.4 ± 1.2 | 94.0 ± 1.4 |
|  | BWT (%) | **0.0 ± 0.0** | −4.0 ± 0.7 | −0.2 ± 0.0 | **0.0 ± 0.0** | −5.5 ± 1.1 | −3.8 ± 1.4 |
|  | FWT (%) | **−0.1 ± 0.1** | **−0.0 ± 0.0** | −0.1 ± 0.1 | −3.3 ± 1.8 | −0.8 ± 0.1 | −0.8 ± 0.1 |
| *Split*-CIFAR | ACC (%) | **80.0 ± 0.5** | 70.6 ± 1.7 | 77.3 ± 0.3 | 68.7 ± 0.8 | 44.2 ± 14.2 | 77.1 ± 0.2 |
|  | BWT (%) | −0.3 ± 0.2 | −2.3 ± 1.4 | **−0.1 ± 0.1** | **0.0 ± 0.0** | −23.9 ± 12.2 | −0.5 ± 0.3 |
|  | FWT (%) | **8.8 ± 0.5** | 1.3 ± 1.0 | 6.8 ± 0.2 | −1.9 ± 0.8 | −3.5 ± 2.1 | 6.9 ± 0.3 |
| Mixed Vision Tasks | ACC (%) | 80.0 ± 1.2 | 49.0 ± 2.8 | **80.3 ± 1.0** | 76.8 ± 2.0 | 26.9 ± 2.1 | 62.8 ± 5.2 |
|  | BWT (%) | −0.9 ± 1.3 | −13.1 ± 1.6 | **−0.1 ± 0.1** | **0.0 ± 0.0** | −35.0 ± 5.6 | −18.7 ± 5.8 |
|  | FWT (%) | **−4.8 ± 1.6** | −23.5 ± 3.4 | −5.8 ± 1.0 | −9.5 ± 2.0 | −23.7 ± 3.8 | **−4.8 ± 0.7** |

Table 1: Performance metrics of GVCL-F and GVCL compared to baselines (more in Appendix J). GVCL-F obtains the best accuracy and backwards/forwards transfer on many datasets/architectures.

## 5.2 *Split*-CIFAR

The popular *Split*-CIFAR dataset, introduced in Zenke et al. (2017), has CIFAR10 as the first task, and then 5 tasks as disjoint 10-way classifications from the first 50 classes of CIFAR100, giving a

total of 6 tasks. We use the same architecture as in other papers (Zenke et al., 2017; Pan et al., 2020). Like with Easy-CHASY, jointly learning these tasks significantly outperforms networks separately trained on the tasks, indicating potential for forward and backward transfer in a continual learning algorithm. Results are in Figure 4. GVCL-F is able to achieve the same final accuracy as joint training with FiLM layers, achieving 80.0±0.5%, beating all baseline algorithms by at least 2%. This confirms that our algorithm performs well in larger settings as well as the previous smaller-scale benchmarks, with minimal forgetting. While the backwards transfer metric for many of the best performing continual learning algorithms is near 0, GVCL-F has the highest forward transfer, achieving 8.5%.

GVCL consistently outperforms VCL, but unlike in the CHASY experiments, it does not outperform Online EWC. This also occurs in the Mixed Vision tasks considered next. Theoretically this should not happen, but GVCL's hyperparameter search found $\beta = 0.2$ which is far from Online EWC. We believe this is because optimizing the GVCL cost for small $\beta$ is more challenging (see Appendix B). However, since intermediate $\beta$ settings result in more pruning, FiLM layers then bring significant improvement.

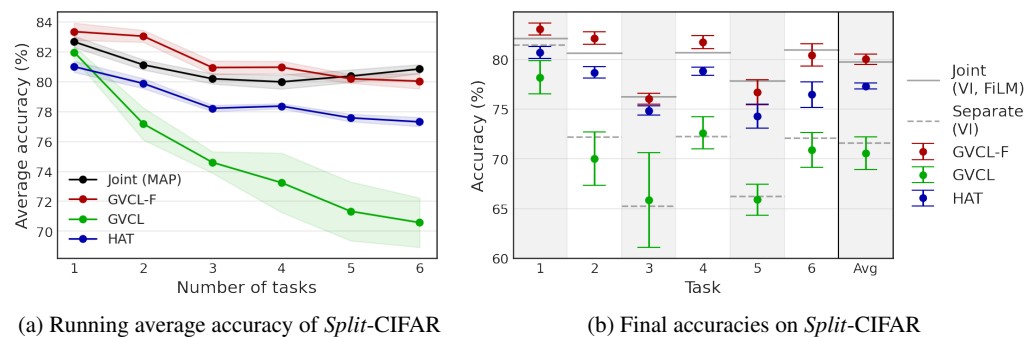

(a) Running average accuracy of *Split*-CIFAR

(b) Final accuracies on *Split*-CIFAR

Figure 4: Running average accuracy of *Split*-CIFAR and final accuracies after continually training on 6 tasks for GVCL-F, GVCL, and HAT. GVCL-F achieves the maximum amount of forwards transfer, and achieves close to the upper-bound joint performance.

## 5.3 MIXED VISION TASKS

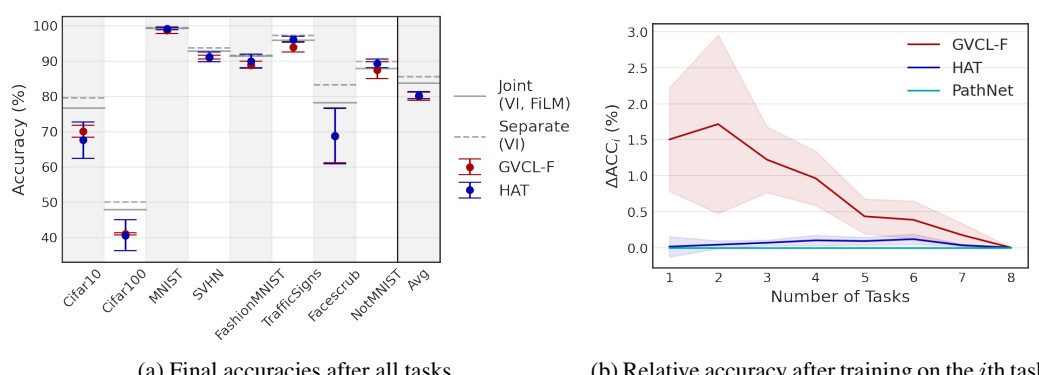

(a) Final accuracies after all tasks

(b) Relative accuracy after training on the $i$th task

Figure 5: (a) Average accuracy of mixed vision tasks at the end of training for GVCL-F and HAT. Both algorithms perform nearly equally well in this respect. (b) GVCL-F gracefully forgets, with higher intermediate accuracies, while HAT has a lower initial accuracy but does not forget.

We finally test on a set of mixed vision datasets, as in Serra et al. (2018). This benchmark consists of 8 image classification datasets with 10-100 classes and a range of dataset sizes, with the order of tasks randomly permuted between different runs. We use the same AlexNet architecture as in Serra et al. (2018). Average accuracies of the 8 tasks after continual training are shown in Figure 5.

GVCL-F's final accuracy matches that of HAT, with similar final performances of 80.0±1.2% and 80.3±1.0% for the two methods, respectively. Figure 5b shows the relative accuracy of the model after training on intermediate tasks compared to its final accuracy. A positive relative accuracy after $t$ tasks means that the method performs better on the tasks seen so far than it does on the same tasks after seeing all 8 tasks (Appendix I contains a precise definition). HAT achieves its continual learning performance by compressing earlier tasks, hindering their performance in order to reserve capacity for later tasks. In contrast, GVCL-F attempts to maximize the performance for early tasks, but allows performance to gradually decay, as shown by the gradually decreasing relative accuracy in Figure 5b. While both strategies result in good final accuracy, one could argue that pre-compressing a network in anticipation of future tasks which may or may not arrive is an impractical real-world strategy, as the number of total tasks may be unknown *a priori*, and therefore one does not know how much to compress the network. The approach taken by GVCL-F is then more desirable, as it ensures good performance after any number of tasks, and frees capacity by "gracefully forgetting".

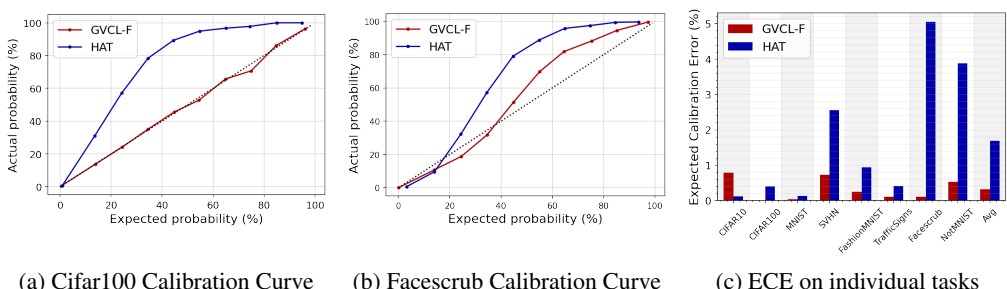

(a) Cifar100 Calibration Curve     (b) Facescrub Calibration Curve     (c) ECE on individual tasks

Figure 6: Calibration curves and Expected Calibration Error for GVCL-F and HAT trained on the Mixed Vision Tasks benchmark. GVCL-F achieves much lower Expected Calibration Error, attaining a value averaged across all tasks of 0.3% compared to HAT's 1.7%.

**Uncertainty calibration.** As GVCL-F is based on a probabilistic framework, we expect it to have good uncertainty calibration compared to other baselines. We show this for the Mixed Vision tasks in Figure 6. Overall, the average Expected Calibration Error for GVCL-F (averaged over tasks) is 0.32%, compared to HAT's 1.69%, with a better ECE on 7 of the 8 tasks. These results demonstrate that GVCL-F is generally significantly better calibrated than HAT, which can be extremely important in decision critical problems where networks must know when they are likely to be uncertain.

## 5.4   RELATIVE GAIN FROM ADDING FILM LAYERS

| Algorithm | Easy-CHASY | Hard-CHASY | *Split*-MNIST (10 tasks) | *Split*-CIFAR | Mixed Vision Tasks | Average |
|---|---|---|---|---|---|---|
| GVCL | 2.0 ± 0.5% | 5.1 ± 0.4% | 4.0 ± 0.7% | 9.5 ± 1.4% | 31.0 ± 2.2% | 10.3 ± 10.6% |
| VCL | 1.5 ± 1.2% | 19.2 ± 1.4% | 2.4 ± 1.5% | 12.0 ± 16.4% | 28.6 ± 3.6% | 12.8 ± 10.3% |
| Online EWC | 2.6 ± 3.3% | 0.3 ± 7.1% | 0.1 ± 1.2% | 0.1 ± 0.1% | 7.7 ± 2.1% | 2.2 ± 2.9% |

Table 2: Relative performance improvement from adding FiLM layers on several benchmarks, for VI and non-VI based algorithms. VI-based approaches see a much more significantly gain over EWC, suggesting that FiLM layers synergize very well with VI and address the pruning issue.

In Section 3, we suggested that adding FiLM layers to VCL in particular would result in the largest gains, since it addresses issues specific to VI, and that FiLM parameter values were automatically best allocated based on the prior. In Section 5.4, we compare the relative gain of adding FiLM layers to VI-based approaches and Online EWC. We omitted HAT, since it already has per-task gating mechanisms, so FiLM layers would be redundant. We see that the gains from adding FiLM layers to Online EWC are limited, averaging 2.2% compared to over 10% for both VCL and GVCL. This suggests that the strength of FiLM layers is primarily in how they interact with variational methods for continual learning. As described in Section 3, with VI we do not need any special algorithm to encourage pruning and how to allocate resources, as they are done automatically by VI. This contrasts HAT, where specific regularizers and gradient modifications are necessary to encourage the use of FiLM parameters.

# 6    CONCLUSIONS

We have developed a framework, GVCL, that generalizes Online EWC and VCL, and we combined it with task-specific FiLM layers to mitigate the effects of variational pruning. GVCL with FiLM layers outperforms strong baselines on a number of benchmarks, according to several metrics. Future research might combine GVCL with memory replay methods, or find ways to use FiLM layers when task ID information is unavailable.

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

## A  LOCAL VS GLOBAL CURVATURE IN GVCL

In this section, we look at the effect of $\beta$ on the approximation of local curvature found from optimizing the $\beta$-ELBO by analyzing its effect on a toy dataset. In doing so, we aim to provide intuition why different values of $\beta$ might outperform $\beta = 1$. We start by looking at the equation of the fixed point of $\Sigma$.

$$\Sigma_T^{-1} = \frac{1}{\beta} \nabla_{\mu_T} \nabla_{\mu_T} \mathbb{E}_{q_T(\theta)}[-\log p(D_T|\theta)] + \Sigma_{T-1}^{-1}. \tag{5}$$

We consider the $T = 1$ case. We can interpret this as roughly measuring the curvature of $\log p(D_T|\theta)$ at different samples of $\theta$ drawn from the distribution $q_T(\theta)$. Based on this equation, we know $\Sigma_T^{-1}$ increases as $\beta$ decreases, so samples from $q_T(\theta)$ are more localized, meaning that the curvature is measured closer to the mean, forming a local approximation of curvature. Conversely, if $\beta$ is larger, $\Sigma_T^{-1}$ broadens and the approximation of curvature is on a more global scale. For simplicity, we write $\nabla_{\mu_T} \nabla_{\mu_T} \mathbb{E}_{q_T(\theta)}[-\log p(D_T|\theta)]$ as $\tilde{H}_T$.

To test this explanation of $\beta$, we performed $\beta$-VI on a simple toy dataset.

We have a true data generative distribution $X \sim \mathcal{N}(0, 1)$, and we sample 1000 points forming the dataset, $D$. Our model is a generative model with $X \sim \mathcal{N}(f(\theta), \sigma_0^2 = 30)$, with $\theta$ being the model's only parameter and $f(\theta)$ an arbitrary fixed function. With $\beta$-VI, we aim to approximate $p(\theta|D)$ with $q(\theta) = \mathcal{N}(\theta; \mu, \sigma^2)$ with a prior $p(\theta) = \mathcal{N}(\theta; 0, 1)$. We choose three different equations for $f(\theta)$:

1. $f_1(\theta) = |\theta|^{1.6}$
2. $f_2(\theta) = \sqrt[4]{|\theta|}$
3. $f_3(\theta) = \sqrt[3]{(|\theta| - 0.5)^3 + 0.4}$

We visualize $\log p(D|\theta)$ for each of these three functions in Figure 7. Here, we see that the data likelihoods have very distinct shapes. $f_1$ results in a likelihood that is flat locally but curves further away from the origin. $f_2$ is the opposite: there is a cusp at 0 then flattens out. $f_3$ is a mix, where at a very small scale it has high curvature, then flattens, then curves again. Now, we perform $\beta$-VI to get $\mu$ and $\sigma^2$, for $\beta \in \{0.1, 1, 10\}$. We then have values for $\sigma^2$, which acts as $\Sigma_T^{-1}$ in Equation 5. We want to extract $\tilde{H}_T^{-1}$ from these values, so we perform the operation $\tilde{\sigma}^2 = \frac{\beta}{\frac{1}{\sigma^2} - 1}$, which represents our estimate of the curvature of $\log p(D|\theta)$ at the mean. This operation also "cancels" the scaling effect of $\beta$. We then plot these approximate log-likelihood functions $\log \tilde{p}(D|\theta) = \mathcal{N}(\theta; \mu, \tilde{\sigma}^2)$ in Figure 8.

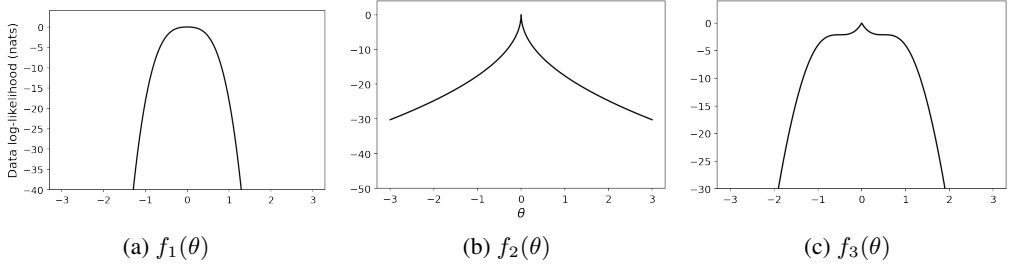

|   |   |   |
|---|---|---|
| (a) $f_1(\theta)$ | (b) $f_2(\theta)$ | (c) $f_3(\theta)$ |

Figure 7: True data log-likelihoods of a generative model of the form $p(x|\theta) = \mathcal{N}(x; f(\theta), \sigma_0^2)$. Curves are shifted so that they pass through the origin

From these figures, we see a clear trend: small values of $\beta$ cause the approximate curvature to be measured locally while larger values cause it to be measured globally, confirming our hypothesis. Most striking is Figure 8c, where the curvature is not strictly increasing or decreasing further from the origin. Here, we see that the curvature first is high for $\beta = 0.1$, then flattens out for $\beta = 1$ then becomes high again for $\beta = 10$. Now imagine in continual learning our posterior for a parameter whose posterior looks like Figure 8a. Here, the parameter would be under-regularized with $\beta = 1$,

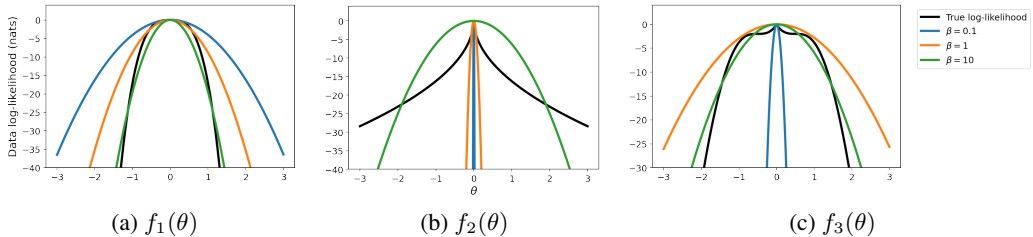

(a) $f_1(\theta)$         (b) $f_2(\theta)$         (c) $f_3(\theta)$

Figure 8: Approximate data log-likelihoods found using $\beta$-VI for various values of $\beta$ for three different generative models. Small values of $\beta$ cause local approximations of curvature and large values cause global ones.

so the parameter will drift far away, significantly affecting performance. Equally, if the posterior was like Figure 8b, then values of $\beta = 1$ would cause the parameter to be over-regularized, limiting model capacity than in practice could be freed. In practice we found that $\beta$ values of $0.05 - 0.2$ worked the best. We leave finding better ways of quantifying the posterior's variable curvature and ways of selecting appropriate values of $\beta$ as future work.

## B CONVERGENCE TO ONLINE-EWC ON A TOY EXAMPLE

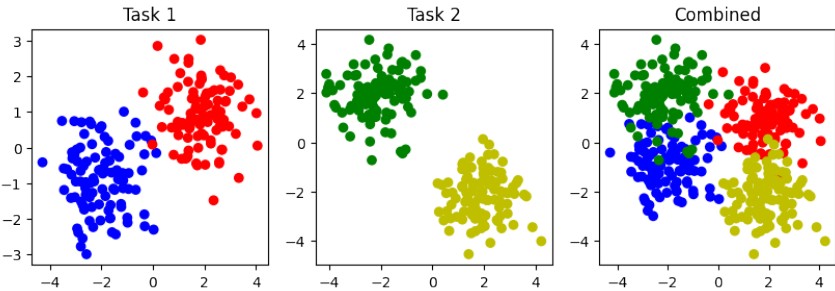

Figure 9: Visualization of a simple 2d logistic regression clustering task. The first task is distinguishing blue and red, classes 1 and 2 respectively. The second task is distinguishing green (class 1) from yellow (class 2). The combined task is shown on the left

Here, we demonstrate convergence of GVCL to Online-EWC for small $\beta$. In this problem, we deal with 2d logistic regression on a toy dataset consisting of separated clusters. The clusters are shown in Figure 9. The first set of tasks is separating the red/blue clusters, then the second is the yellow/green clusters. Blue and green are the first class and red and yellow are the second. Or model is given by the equation

$$p(y_i = 1|w, b, x_i) = \sigma(w^\top x_i + b)$$

Where $x_i$ are our datapoints and $w$ and $b$ are our parameters. $y_i = 1$ means class 2 (and $y_i = 0$ means class 1). $x$ is 2-dimensional so we have a total of 3 parameters.

Next, we ran GVCL with decreasing values of $\beta$ and compared the resulting values of $w$ and $b$ after the second task to solution generated by Online-EWC. For both cases, we set $\lambda = 1$. For our prior, we used the unit normal prior on both $w$ and $b$, our approximating distribution was a fully factorized Gaussian. We ran this experiment for 5 random seeds (of the parameters, not the clusters) and plotted the results.

Figure 10 shows the result. Evidently, the values of the parameters approach those of Online-EWC as we decrease $\beta$, in line with our theory. However, it is worth noting that to get this convergent behaviour, we had to run this experiment for very long. For the lowest $\beta$ value, it took 17 minutes to

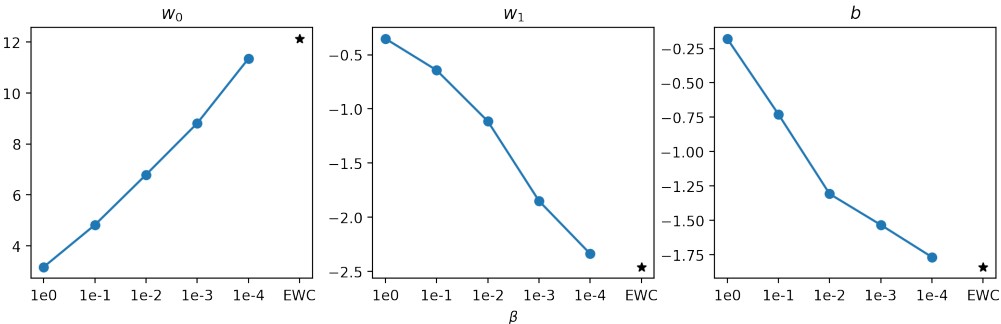

Figure 10: Convergence of GVCL parameter values to Online-EWC parameter values for decreasing values of $\beta$ for a toy 2d logistic regression problem

converge compared to 1.7 for $\beta = 1$. A small learning rate of 1e-4 with 100000 iteration steps was necessary for the smallest $\beta =$1e-4. If the optimization process was run for shorter, or too large a learning rate was used, we would observe convergent behaviour for the first few values of $\beta$, but the smallest values of $\beta$ would result in completely different values.

This shows that while in theory, for small $\beta$, GVCL should approach Online-EWC, it is extremely hard to achieve in practice. Given that it takes so long to achieve convergent behaviour on a model with 3 parameters, it is unsurprising that we were not able to achieve the same performance as Online-EWC for our neural networks, and explains why despite GVCL, in theory, encompassing Online-EWC, can sometimes perform worse.

## C   FURTHER DETAILS ON RECOVERING ONLINE EWC

Here, we show the full derivation to recover Online EWC from GVCL, as $\beta \to 0$. First, we expand the $\beta$-ELBO which for Gaussian priors and posteriors has the form:

$$
\begin{aligned}
\beta\text{-ELBO} &= \mathbb{E}_{\theta \sim q_T(\theta)} \log p(D_T|\theta) - \beta D_{\text{KL}}(q_T(\theta)||q_{T-1}(\theta)) \\
&= \mathbb{E}_{\theta \sim q_T(\theta)}[\log p(D_T|\theta)] - \frac{\beta}{2}\bigg( \log|\Sigma_{T-1}| - \log|\Sigma_T| - d \\
&\quad + \text{Tr}(\Sigma_{T-1}^{-1}\Sigma_T) + (\mu_T - \mu_{T-1})^\top \Sigma_T^{-1}(\mu_T - \mu_{T-1}) \bigg),
\end{aligned}
$$

where $q_T(\theta)$ is our approximate distribution with means and covariance $\mu_T$ and $\Sigma_T$, and our prior distribution $q_{T-1}(\theta)$ has mean and covariance $\mu_{T-1}$ and $\Sigma_{T-1}$. $D_T$ refers to the $T$th dataset and $d$ the dimension of $\mu$. Next, take derivatives wrt $\Sigma_T$ and set to 0:

$$
\nabla_{\Sigma_T}\beta\text{-ELBO} = \nabla_{\Sigma_T}\mathbb{E}_{\theta \sim q_T(\theta)}[\log p(D_T|\theta)] + \frac{\beta}{2}\Sigma_T^{-1} - \frac{\beta}{2}\Sigma_{T-1}^{-1} \tag{6}
$$

$$
0 = \frac{1}{2}\nabla_\mu \nabla_\mu \mathbb{E}_{q_T(\theta)}[\log p(D_T|\theta)] + \frac{\beta}{2}\Sigma_T^{-1} - \frac{\beta}{2}\Sigma_{T-1}^{-1} \tag{7}
$$

$$
\Rightarrow \Sigma_T^{-1} = \frac{1}{\beta}\nabla_{\mu_T}\nabla_{\mu_T}\mathbb{E}_{q_T(\theta)}[-\log p(D_T|\theta)] + \Sigma_{T-1}^{-1}. \tag{8}
$$

We move from Equation 6 to Equation 7 using Equation 19 in Opper & Archambeau (2008). From Equation 8, we see that as $\beta \to 0$, the precision grows indefinitely, so $q_T(\theta)$ approaches a delta function centered at its mean. We give a more precise explanation of this argument in Appendix C.1.

We have

$$\Sigma_T^{-1} = -\frac{1}{\beta}\nabla_{\mu_T}\nabla_{\mu_T}\log p(D_T|\theta = \mu_T) + \Sigma_{T-1}^{-1}$$

$$\Sigma_T^{-1} = \frac{1}{\beta}H_T + \Sigma_{T-1}^{-1}, \tag{9}$$

where $H_T$ is the Hessian of the $T$th dataset log-likelihood. This recursion of $\Sigma_T^{-1}$ gives

$$\Sigma_T^{-1} = \frac{1}{\beta}\sum_{t=1}^{T} H_t + \Sigma_0^{-1}.$$

Now, optimizing the $\beta$-ELBO for $\mu_T$ (ignoring terms that do not depend on $\mu_T$):

$$\beta\text{-ELBO} = \mathbb{E}_{\theta\sim q(\theta)}[\log p(D|\theta)] - \frac{\beta}{2}(\mu_T - \mu_{T-1})^\top \Sigma_{T-1}^{-1}(\mu_T - \mu_{T-1}) \tag{10}$$

$$= \log p(D|\theta = \mu_T) - \frac{1}{2}(\mu_T - \mu_{T-1})^\top \left(\sum_{t=1}^{T-1} H_t + \beta\Sigma_0^{-1}\right)(\mu_T - \mu_{T-1}). \tag{11}$$

Which is the exact optimization problem for Laplace Propagation (Smola et al., 2003). If we note that $H_T \approx N_T F_T$ (Martens, 2020), where $N_T$ is the number of samples in the $T$th dataset and $F_T$ is the Fisher information matrix, we recover Online EWC with $\lambda = 1$ when $N_1 = N_2 = ... = N_T$ (with $\gamma = 1$).

## C.1 CLARIFICATION OF THE DELTA-FUNCTION ARGUMENT

In C, we argued,

$$\Sigma_T^{-1} = \frac{1}{\beta}\nabla_{\mu_T}\nabla_{\mu_T}\mathbb{E}_{q_T(\theta)}[-\log p(D_T|\theta)] + \Sigma_{T-1}^{-1}$$

$$\approx \frac{1}{\beta}H_T + \Sigma_{T-1}^{-1}$$

for small $\beta$. We argued that for small $\beta$, $q(\theta)$ collapsed to its mean and it is safe to treat the expectation as sampling only from the mean. In this section, we show that this argument is justified.

**Lemma 1.** *If $q(\theta)$ has mean and covariance parameters $\mu$ and $\Sigma$, and $\Sigma^{-1} = \frac{1}{\beta}\nabla_\mu\nabla_\mu\mathbb{E}_{\theta\sim q(\theta)}[f(\theta)] + C$, $C = O(\frac{1}{\beta})$, then for small $\beta$, $\Sigma^{-1} \approx \frac{1}{\beta}H_\mu + C$, where $H_\mu$ is the Hessian of $f(\theta)$ evaluated at $\mu$, assuming $H_\mu = O(1)$*

*Proof.* We first assume that $f(\theta)$ admits a Taylor expansion around $\mu$. For notational purposes, we define,

$$T_{k_1,...,k_n}\Big|_{\theta=\mu} = \frac{\partial f}{\partial\theta^{(k_1)}\ldots\partial\theta^{(k_n)}}\Big|_{\theta=\mu}$$

For our notation, upper indices in brackets indicate vector components (not powers), and lower indices indicate covector components. Note that, $H_{\mu,i,j} = T_{i,j}\Big|_{\theta=\mu}$. [5]

Then, a Taylor expansion centered at $\mu$ has the form

$$f(\theta) = f(\mu) + \sum_{n=1}^{\infty}\frac{1}{n!}T_{k_1,...,k_n}\Big|_{\theta=\mu}(\theta - \mu)^{(k_1)}\ldots(\theta - \mu)^{(k_n)}$$

---

[5] In this case, the $\mu$ in $H_{\mu,i,j}$ refers to the Hessian evaluated at $\mu$, while $i, j$ refers to the indices

Where we use Einstein notation, so

$$T_{k_1,\ldots,k_n}\Big|_{\theta=\mu}(\theta-\mu)^{(k_1)}\ldots(\theta-\mu)^{(k_n)} = \sum_{k_1,\ldots,k_n=1}^{D} T_{k_1,\ldots,k_n}\Big|_{\theta=\mu}(\theta-\mu)^{(k_1)}\ldots(\theta-\mu)^{(k_n)}$$

$$(12)$$

With $D$ the dimension of $\theta$. To denote the central moments of $q(\theta)$, we define

$$\tilde{\mu}^{(k_1,\ldots,k_n)} := \mathbb{E}_{\theta\sim q(\theta)}\left[(\theta-\mu)^{(k_1)}\ldots(\theta-\mu)^{(k_n)}\right]$$

These moments can be computed using Isserlis' theorem. Notably, for a Gaussian, if $n$ is odd, $\tilde{\mu}^{(k_1,\ldots,k_n)} = 0$

Now, we can compute our expectation as an infinite sum:

$$\nabla_\mu\nabla_\mu\mathbb{E}_{\theta\sim q(\theta)}[f(\theta)] = \nabla_\mu\nabla_\mu\mathbb{E}_{\theta\sim q(\theta)}\left[f(\mu)+\sum_{n=1}^{\infty}\frac{1}{n!}T_{k_1,\ldots,k_n}\Big|_{\theta=\mu}(\theta-\mu)^{(k_1)}\ldots(\theta-\mu)^{(k_n)}\right]$$

$$= \nabla_\mu\nabla_\mu\left[f(\mu)+\sum_{n=1}^{\infty}\frac{1}{n!}T_{k_1,\ldots,k_n}\Big|_{\theta=\mu}\tilde{\mu}^{(k_1,\ldots,k_n)}\right]$$

$$= \nabla_\mu\nabla_\mu\left[f(\mu)+\sum_{n=1}^{\infty}\frac{1}{2n!}T_{k_1,\ldots,k_{2n}}\Big|_{\theta=\mu}\tilde{\mu}^{(k_1,\ldots,k_{2n})}\right] \quad \text{(odd moments are 0)}$$

$$= A \quad \text{for notational simplicity}$$

We can look at individual components of $A$:

$$A_{i,j} = \frac{\partial}{\partial\mu^{(i)}}\frac{\partial}{\partial\mu^{(j)}}\left[f(\mu)+\sum_{n=1}^{\infty}\frac{1}{2n!}T_{k_1,\ldots,k_{2n}}\Big|_{\theta=\mu}\tilde{\mu}^{(k_1,\ldots,k_{2n})}\right]$$

$$= T_{i,j}\Big|_{\theta=\mu}+\sum_{n=1}^{\infty}\frac{1}{2n!}T_{i,j,k_1,\ldots,k_{2n}}\Big|_{\theta=\mu}\tilde{\mu}^{(k_1,\ldots,k_{2n})}$$

Now we can insert this into our original equation.

$$\Sigma^{-1} = \frac{1}{\beta}\nabla_\mu\nabla_\mu\mathbb{E}_{\theta\sim q(\theta)}[f(\theta)] + C$$

$$\Sigma^{-1} = \frac{1}{\beta}A + C$$

$$\Sigma_{i,j}^{-1} = \frac{1}{\beta}A_{i,j} + C_{i,j} \quad \text{looking at individual indices}$$

$$\underbrace{\Sigma_{i,j}^{-1}}_{O(\frac{1}{\beta})} = \frac{1}{\beta}\left(\underbrace{T_{i,j}\Big|_{\theta=\mu}}_{O(1)}+\underbrace{\sum_{n=1}^{\infty}\frac{1}{2n!}T_{i,j,k_1,\ldots,k_{2n}}\Big|_{\theta=\mu}\tilde{\mu}^{(k_1,\ldots,k_{2n})}}_{O(\beta)}\right) + \underbrace{C_{i,j}}_{O(\frac{1}{\beta})}$$

Now we assumed that $H_\mu$ is $O(1)$ (so $T_{i,j}\Big|_{\theta=\mu}$ is too), which means that $\Sigma_{i,j}^{-1}$ must be at least $O(\frac{1}{\beta})$. If $\Sigma^{-1} = O(\frac{1}{\beta})$, then $\Sigma = O(\beta)$. From Isserlis' theorem, we know that $\tilde{\mu}^{(k_1,\ldots,k_{2n})}$ is composed of the product of $n$ elements of $\Sigma$, so $\tilde{\mu}^{(k_1,\ldots,k_{2n})} = O(\beta^n)$. $T_{i,j,k_1,\ldots,k_{2n}}\Big|_{\theta=\mu}$ is constant with respect to $\beta$, so is $O(1)$. Hence, the summation is $O(\beta)$, which for small $\beta$ is negligible compared to the $O(1)$ term $T_{i,j}\Big|_{\theta=\mu}$, so can therefore be ignored. Then, keeping only $O(\frac{1}{\beta})$ terms,

$$\overbrace{\tilde{\Sigma}_{i,j}^{-1}}^{O(\frac{1}{\beta})} = \frac{1}{\beta}\left(\overbrace{T_{i,j}\Big|_{\theta=\mu}}^{O(1)} + \overbrace{\sum_{n=1}^{\infty}\frac{1}{2n!}T_{i,j,k_1,..,k_{2n}}\Big|_{\theta=\mu}\tilde{\mu}^{(k_1,...,k_{2n})}}^{O(\beta)}\right) + \overbrace{C_{i,j}}^{O(\frac{1}{\beta})}$$

$$\overbrace{\tilde{\Sigma}_{i,j}^{-1}}^{O(\frac{1}{\beta})} = \overbrace{\frac{1}{\beta}T_{i,j}\Big|_{\theta=\mu}}^{O(\frac{1}{\beta})} + \overbrace{\frac{1}{\beta}\left(\sum_{n=1}^{\infty}\frac{1}{2n!}T_{i,j,k_1,...,k_{2n}}\Big|_{\theta=\mu}\tilde{\mu}^{(k_1,...,k_{2n})}\right)}^{O(1)} + \overbrace{C_{i,j}}^{O(\frac{1}{\beta})}$$

$$\approx \frac{1}{\beta}T_{i,j}\Big|_{\theta=\mu} + C_{i,j}$$

$$= \frac{1}{\beta}H_{\mu,i,j} + C_{i,j}$$

$$\Sigma^{-1} \approx \frac{1}{\beta}H_\mu + C$$

$\square$

## C.2 CORRESPONDING GVCL'S $\lambda$ AND ONLINE EWC'S $\lambda$

We use $D_{\mathrm{KL}\tilde{\lambda}}$ in place of $D_{\mathrm{KL}}$, with $D_{\mathrm{KL}\tilde{\lambda}}$ defined as

$$D_{\mathrm{KL}\tilde{\lambda}}(q_T\|q_{T-1}) = \frac{1}{2}\Big((\mu_T - \mu_{T-1})^\top\tilde{\mathbf{\Sigma}}_{\mathbf{T-1},\lambda}^{-1}\mu_T - \mu_{T-1}) + \mathrm{Tr}(\Sigma_{T-1}^{-1}\Sigma_T)$$
$$+ \log|\Sigma_{T-1}| - d - \log|\Sigma_T|\Big),$$

with

$$\tilde{\Sigma}_{T,\lambda}^{-1} := \frac{\lambda}{\beta}\sum_{t=1}^{T}H_t + \Sigma_0^{-1} = \lambda(\Sigma_T^{-1} - \Sigma_0^{-1}) + \Sigma_0^{-1}.$$

Now, the fixed point for $\Sigma_T$ is still given by Equation 9, but the $\beta$-ELBO for for terms involving $\mu_T$ has the form,

$$\beta\text{-ELBO} = \mathbb{E}_{\theta\sim q(\theta)}[\log p(D|\theta)] - \frac{\beta}{2}(\mu_T - \mu_{T-1})^\top\tilde{\Sigma}_{T-1,\lambda}^{-1}(\mu_T - \mu_{T-1})$$

$$= \log p(D|\theta = \mu_T) - \frac{1}{2}(\mu_T - \mu_{T-1})^\top\left(\lambda\sum_{t=1}^{T}H_t + \beta\Sigma_0^{-1}\right)(\mu_T - \mu_{T-1}),$$

which upweights the quadratic terms dependent on the data (and not the prior), similarly to $\lambda$ in Online EWC.

## C.3 RECOVERING $\gamma$ FROM TEMPERING

In order to recover $\lambda$, we used the KL-divergence between tempered priors and posteriors $q_{T-1}^\lambda$ and $q_T^\lambda$. Recovering $\gamma$ can be done using the same trick, except we temper the posterior to $q_T^{\gamma\lambda}$:

$$D_{\mathrm{KL}}(q_T^\lambda\|q_{T-1}^{\gamma\lambda}) = \tfrac{1}{2}\big((\mu_T - \mu_{T-1})^\top\lambda\Sigma_{T-1}^{-1}(\mu_T - \mu_{T-1})$$
$$+ \mathrm{Tr}(\gamma\lambda\Sigma_{T-1}^{-1}\lambda^{-1}\Sigma_T) + \log\frac{|\lambda^{-1}\Sigma_{T-1}|}{|(\gamma\lambda)^{-1}\Sigma_T|} - d\big)$$
$$= \tfrac{1}{2}\big((\mu_T - \mu_{T-1})^\top\lambda\Sigma_{T-1}^{-1}(\mu_T - \mu_{T-1}) + \gamma\mathrm{Tr}(\Sigma_{T-1}^{-1}\Sigma_T) - \log|\Sigma_T|\big) + \text{cons.}$$
$$= D_{\mathrm{KL}\lambda,\gamma}(q_T\|q_{T-1})$$

We can apply the same $\lambda$ to $\tilde{\lambda}$ as before to get $D_{\mathrm{KL}\tilde{\lambda},\gamma}(q_T \| q_{T-1})$. Plugging this into the $\beta$-ELBO and solving yields the recursion for $\Sigma_T$ to be

$$\Sigma_T^{-1} = \frac{1}{\beta} H_T + \gamma \Sigma_{T-1}^{-1},$$

which is exactly that of Online EWC.

## C.4 GVCL RECOVERS THE SAME APPROXIMATION OF $F_T$ AS ONLINE EWC

The earlier analysis dealt with full rank $\Sigma_T$. In practice, however, $\Sigma_T$ is rarely full rank and we deal with approximations of $\Sigma_T$. In this subsection, we consider diagonal $\Sigma_T$, like Online EWC, which in practice uses a diagonal approximation of $F_T$. The way Online EWC approximates this diagonal is by matching diagonal entries of $F_T$. There are many ways of producing a diagonal approximation of a matrix, for example matching diagonals of the inverse matrix is also valid, depending on the metric we use. Here, we aim to show that that the diagonal approximation of $\Sigma_T$ that is produced when $\mathcal{Q}$ is the family of diagonal covariance Gaussians is the same as the way Online EWC approximates $F_T$, that is, diagonals of $\Sigma_{T,\mathrm{approx}}^{-1}$ match diagonals of $\Sigma_{T,\mathrm{true}}^{-1}$, i.e. we match the diagonal *precision* entries, not the diagonal covariance entries.

Let $\Sigma_{T,\mathrm{approx}} = \mathrm{diag}(\sigma_1^2, \sigma_2^2, ..., \sigma_d^2)$, with $d$ the dimension of the matrix. Because we are performing VI, we are optimizing the forwards KL divergence, i.e. $D_{\mathrm{KL}}(q_{\mathrm{approx}} \| q_{\mathrm{true}})$. Therefore, ignoring terms that do not depend on $\Sigma_{T,\mathrm{approx}}$,

$$D_{\mathrm{KL}}(q_{\mathrm{approx}} \| q_{\mathrm{true}}) = \frac{1}{2}\mathrm{Tr}(\Sigma_{T,\mathrm{approx}}\Sigma_{T,\mathrm{true}}^{-1}) - \frac{1}{2}\log|\Sigma_{T,\mathrm{approx}}| + (\text{constants wrt } \Sigma_{T,\mathrm{approx}})$$

$$= \frac{1}{2}\sum_{i=1}^{d}(\Sigma_{T,\mathrm{approx}}\Sigma_{T,\mathrm{true}}^{-1})_{i,i} - \frac{1}{2}\sum_{i=1}^{d}\log\sigma_i^2$$

$$= \frac{1}{2}\sum_{i=1}^{d}\left(\sigma_i^2(\Sigma_{T,\mathrm{true}}^{-1})_{i,i} - \log\sigma_i^2\right).$$

Optimizing wrt $\sigma_i^2$:

$$\frac{\partial D_{\mathrm{KL}}(q_{\mathrm{approx}} \| q_{\mathrm{true}})}{\partial \sigma_i^2} = 0 = \frac{1}{2}\left((\Sigma_{T,\mathrm{true}}^{-1})_{i,i} - \frac{1}{\sigma_i^2}\right)$$

$$\Rightarrow \sigma_i^2 = \frac{1}{(\Sigma_{T,\mathrm{true}}^{-1})_{i,i}}.$$

So we have that diagonals of $\Sigma_{T,\mathrm{approx}}^{-1}$ match diagonals of $\Sigma_{T,\mathrm{true}}^{-1}$.

## C.5 GVCL RECOVERS THE SAME APPROXIMATION OF $H_T$ AS SOLA

SOLA approximates the Hessian with a rank-restricted matrix $\tilde{H}$ (Yin et al., 2020). We first consider a relaxation of this problem with full rank, then consider the limit when we reduce this relaxation.

Because we are concerned with limiting $\beta \to 0$, it is sufficient to consider $\Sigma_{\mathrm{true}}^{-1}$ as $H$, the true Hessian. Because $H$ is symmetric (and assuming it is positive-semi-definite), we can also write H as $H = VDV^{\top} = \sum_{i=1}^{p}\lambda_i x_i x_i^{\top}$, with $D$, and $V$ be the diagonal matrix of eigenvalues and a unitary matrix of eigenvectors, respectively. These eigenvalues and eigenvectors are $\lambda_i$ and $x_i$, respectively, and $p$ the dimension of $H$.

For $\tilde{H}$, we first consider full-rank matrix which becomes low-rank as $\delta \to 0$:

$$\tilde{H} = \sum_{i=1}^{k}\tilde{\lambda}_i\tilde{x}_i\tilde{x}_i^{\top} + \sum_{j=k+1}^{p}\delta\tilde{x}_j\tilde{x}_j^{\top}$$

This matrix has $\tilde{\lambda}_i, 1 \le i \le k$ as its first $k$ eigenvalues and $\delta$ as its remaining. We also set $\tilde{x}_i^\top \tilde{x}_i = 1$ and $\tilde{x}_i^\top \tilde{x}_j = 0, i \ne j$.

With KL minimization, we aim to minimize (up to a constant and scalar factor),

$$\text{KL} = \text{Tr}(\Sigma_{\text{approx}} \Sigma_{\text{true}}^{-1}) - \log |\Sigma_{\text{approx}}|$$

In our case, this is Equation 13, which we can further expand as,

$$\text{KL} = \text{Tr}(\tilde{H}^{-1} H) - \log |\tilde{H}^{-1}| \tag{13}$$

$$= \text{Tr}\left(\left(\sum_{i=1}^{k} \frac{1}{\tilde{\lambda}_i} \tilde{x}_i \tilde{x}_i^\top + \sum_{j=k+1}^{p} \frac{1}{\delta} \tilde{x}_j \tilde{x}_j^\top \right) H \right) + \sum_{i=1}^{k} \log(\tilde{\lambda}_i) + \sum_{j=k+1}^{p} \log \delta \tag{14}$$

$$= \text{Tr}\left(\sum_{i=1}^{k} \frac{1}{\lambda_i} \tilde{x}_i \tilde{x}_i^\top H \right) + \text{Tr}\left(\sum_{j=k+1}^{p} \frac{1}{\delta} \tilde{x}_j \tilde{x}_j^\top H \right) + \sum_{i=1}^{k} \log(\tilde{\lambda}_i) + \sum_{j=k+1}^{p} \log \delta \tag{15}$$

$$= \sum_{i=1}^{k} \frac{1}{\lambda_i} \tilde{x}_i^\top H \tilde{x}_i + \sum_{j=k+1}^{p} \frac{1}{\delta} \tilde{x}_j^\top H \tilde{x}_j + \sum_{i=1}^{k} \log(\tilde{\lambda}_i) + \sum_{j=k+1}^{p} \log \delta \tag{16}$$

$$\tag{17}$$

Taking derivatives wrt $\tilde{\lambda}_i$, we have:

$$\frac{\partial KL}{\partial \tilde{\lambda}_i} = 0 = -\frac{1}{\tilde{\lambda}_i^2} \tilde{x}_i^\top H \tilde{x}_i + \frac{1}{\tilde{\lambda}_i} \tag{18}$$

$$\Rightarrow \tilde{\lambda}_i = \tilde{x}_i^\top H \tilde{x}_i \tag{19}$$

Which when put into Equation 16,

$$\text{KL} = \sum_{i=1}^{k} \frac{1}{\lambda_i} \tilde{x}_i^\top H \tilde{x}_i + \sum_{j=k+1}^{p} \frac{1}{\delta} \tilde{x}_j^\top H \tilde{x}_j + \sum_{i=1}^{k} \log(\tilde{\lambda}_i) + \sum_{j=k+1}^{p} \log \delta \tag{20}$$

$$= \sum_{i=1}^{k} \frac{\tilde{x}_i^\top H \tilde{x}_i}{\tilde{x}_i^\top H \tilde{x}_i} + \sum_{j=k+1}^{p} \frac{1}{\delta} \tilde{x}_j^\top H \tilde{x}_j + \sum_{i=1}^{k} \log(\tilde{\lambda}_i) + \sum_{j=k+1}^{p} \log \delta \tag{21}$$

$$= k + \sum_{j=k+1}^{p} \frac{1}{\delta} \tilde{x}_j^\top H \tilde{x}_j + \sum_{i=1}^{k} \log(\tilde{\lambda}_i) + \sum_{j=k+1}^{p} \log \delta \tag{22}$$

$$= \frac{1}{\delta} \sum_{j=k+1}^{p} \tilde{x}_j^\top H \tilde{x}_j + \sum_{i=1}^{k} \log(\tilde{\lambda}_i) \quad \text{(removing constants)} \tag{23}$$

$$= \frac{1}{\delta} \sum_{j=k+1}^{p} \tilde{x}_j^\top H \tilde{x}_j + \sum_{i=1}^{k} \log(\tilde{x}_i^\top H \tilde{x}_i) \tag{24}$$

Now we need to consider the constraints $\tilde{x}_i^\top \tilde{x}_i = 1$ and $\tilde{x}_i^\top \tilde{x}_j = 0, i \ne j$ by adding Lagrange multipliers to our KL cost,

$$L = \frac{1}{\delta} \sum_{j=k+1}^{p} \tilde{x}_j^\top H \tilde{x}_j + \sum_{i=1}^{k} \log(\tilde{x}_i^\top H \tilde{x}_i) - \sum_{i=1}^{k} \phi_{i,i}(\tilde{x}_i^\top \tilde{x}_i - 1) - \sum_{i,j,i \ne j} \phi_{i,j} \tilde{x}_i^\top \tilde{x}_j \tag{25}$$

Taking derivatives wrt $\tilde{x}_i$:

$$\frac{\partial L}{\partial \tilde{x}_i} = 0 = \frac{2H\tilde{x}_i}{\tilde{x}_i^\top H \tilde{x}_i} - 2\phi_{i,i}\tilde{x}_i - 2\sum_{i,j\neq i}\phi_{i,j}\tilde{x}_j \tag{26}$$

$$\sum_{i,j\neq i}\phi_{i,j}\tilde{x}_j = \left(\frac{H}{\tilde{x}_i^\top H \tilde{x}_i} - \phi_{i,i}I_p\right)\tilde{x}_i \tag{27}$$

In Equation 27, we have $\tilde{x}_i$ expressed as a linear combination of $\tilde{x}_j, j \neq i$, but $\tilde{x}_i$ and $\tilde{x}_j$ are orthogonal, so $\tilde{x}_i$ cannot be expressed as such, so $\phi_{i,j} = 0, i \neq j$, and,

$$\frac{H\tilde{x}_i}{\tilde{x}_i^\top H \tilde{x}_i} = \phi_{i,i}\tilde{x}_i \tag{28}$$

Meaning $\tilde{x}_i$ are eigenvectors of $H$ for $1 \leq i \leq k$. We can also use the same Lagrange multipliers to show that $\tilde{x}_i$ for $k+1 \leq i \leq p$ are also eigenvectors of $H$.

This means that our cost,

$$\text{KL} = \frac{1}{\delta}\sum_{j=k+1}^{p}\tilde{x}_j^\top H \tilde{x}_j + \sum_{i=1}^{k}\log(\tilde{x}_i^\top H \tilde{x}_i) \tag{29}$$

$$= \frac{1}{\delta}\sum_{j=k+1}^{p}\tilde{\kappa}_j + \sum_{i=1}^{k}\log(\tilde{\kappa}_i) \tag{30}$$

where the set $(\tilde{\kappa}_1, \tilde{\kappa}_2, ..., \tilde{\kappa}_p)$ is a permutation of $(\lambda_1, \lambda_2, ..., \lambda_p)$ and $\tilde{\kappa}_i = \tilde{\lambda}_i$ for $1 \leq i \leq k$. I.e., $\tilde{H}$ shares $k$ eigenvalues with $H$, and the rest are $\delta$. It now remains to determine which eigenvalues are shared and which are excluded.

Considering only two eigenvalues, $\lambda_i, \lambda_j$, and let $\lambda_i > \lambda_j \geq 0$. Let $r = \frac{\lambda_i}{\lambda_j}$. The relative cost of excluding $\lambda_i$ in the set $\{\tilde{\kappa}_1, \tilde{\kappa}_2, ..., \tilde{\kappa}_k\}$ compared to including it is,

$$\text{Relative Cost} = \frac{\lambda_i - \lambda_j}{\delta} - \log\frac{\lambda_i}{\lambda_j}$$

$$= \frac{\lambda_i(1 - \frac{1}{r})}{\delta} - \log r$$

If the relative cost is positive, then including $\lambda_i$ as one of the eigenvalues of $\tilde{H}$ is the more optimal choice. Now solving the inequality,

$$\text{Relative Cost} > 0$$

$$\frac{\lambda_i(1 - \frac{1}{r})}{\delta} - \log r > 0$$

$$\lambda_i > \delta(1 - \frac{1}{r})\log r$$

Which, for sufficiently small $\delta$ is always true because $r > 1$. Thus, it is always better to swap two eigenvalues which are included/excluded, if the excluded one is larger. This means that $\tilde{H}$ has the $k$ largest eigenvalues of $H$, and we already showed that it shares the same eigenvectors. This maximum eigenvalue/eigenvector pair selection is exactly the procedure used by SOLA.

## D    COLD POSTERIOR VCL AND FURTHER GENERALIZATIONS

The use of KL-reweighting is closely related related to the idea of "cold-posteriors," in which $p_T(\theta|D) \propto p(\theta|D)^{\frac{1}{\tau}}$. Finding this cold posterior is equivalent to find optimal q distributions for maximizing the $\tau$-ELBO:

$$\tau\text{-ELBO} := \mathbb{E}_{\theta \sim q(\theta)}[\log p(D|\theta) + \log p(\theta) - \tau \log q(\theta)]$$

when $\mathcal{Q}$ is all possible distributions of $\theta$. This objective is the same as the standard ELBO with only the entropy term reweighted, and contrasts the $\beta$-ELBO where both the entropy and prior likelihoods are reweighted. Here, $\beta$ acts similarly to $T$ (the temperature, not to be confused with task number). This relationship naturally leads to the transition diagram shown in Figure 11. In this, we can see that we can easily transition between posteriors at different temperatures by optimizing either the $\beta$-ELBO, $\tau$-ELBO, or tempering the posterior.

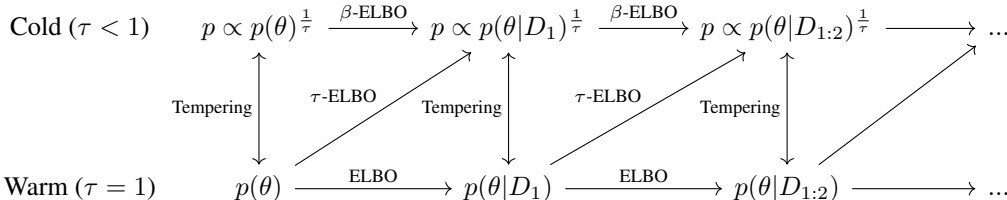

Figure 11: Transitions between posteriors at different temperatures using tempering and optimizing either the $\tau$-ELBO or $\beta$-ELBO

When $\mathcal{Q}$ contains all possible distributions, moving along any path results in the exact same distribution, for example optimizing the $\tau$-ELBO then tempering is the same as directly optimizing the ELBO. However in the case where $Q$ is limited, this transition is not exact, and the resulting posterior is path dependent. In fact, each possible path represents a different valid method for performing continual learning. Standard VCL works by traversing the horizontal arrows, directly optimizing the ELBO, while an alternative scheme of VCL would optimize the $\tau$-ELBO to form cold posteriors, then heat the posterior before optimizing the $\tau$-ELBO for a new task. Inference can be done at either the warm or cold state. Note that for Gaussians, heating the posterior is just a matter of scaling the covariance matrix by a constant factor $\frac{\tau_{\text{after}}}{\tau_{\text{before}}}$.

While warm posteriors generated through this two-step procedure are not optimal under the ELBO, when $\mathcal{Q}$ is limited, they may perform better for continual learning. Similar to Equation 2, the optimal $\Sigma$ when optimizing the $\tau$-ELBO is given by

$$\Sigma_T^{-1} = \frac{1}{\tau}\sum_{t=1}^{T}\tilde{H}_t + \frac{1}{\tau}\Sigma_0^{-1}$$

Where $\tilde{H}_t$ is the approximate curvature for a specific value of $\tau$ for task $t$, which coincides with the true Hessian for $\tau \to 0$, like with the $\beta$-ELBO. Here, both the prior and data-dependent component are scaled by $\frac{1}{\tau}$, in contrast to Equation 2, where only the data-dependent component is reweighted. As discussed in Section 2.2 and further explored in appendix A, this leads to a different scale of the quadratic approximation, which may lend itself better for continual learning. This also results in a second way to recover $\gamma$ in Online EWC by first optimizing the $\beta$-ELBO with $\beta = \gamma$, then tempering by a factor of $\frac{1}{\gamma}$ (i.e. increasing the temperature when $\gamma < 1$).

## E    MAP DEGENERACY WITH FILM LAYERS

Here we describe how training FiLM layer with MAP training leads to degenerate values for the weights and scales, whereas with VI training, no degeneracy occurs. For simplicity, consider only the nodes leading into a single node and let there be $d$ of them, i.e. $\theta$ has dimension $d$. Because we only have one node, our scale parameter $\gamma$ is a single variable.

For MAP training, we have the loss function $L = -p(D|\theta, \gamma) + \frac{\lambda}{2}\theta^2$, with $D$ the dataset and $\lambda$ the L2 regularization hyperparameter. Note that $p(D|\theta, \gamma) = p(D|c\theta, \frac{1}{c}\gamma)$, hence we can scale $\theta$ arbitrarily without affecting the likelihood, so long as $\gamma$ is scaled inversely. If $c < 1$, $\frac{\lambda}{2}\theta^2 < \frac{\lambda}{2}(\frac{1}{c}\theta)^2$, so increasing $c$ decreases the L2 penalty if $\theta$ is inversely scaled by $c$. Therefore the optimal setting of the scale parameter $\gamma$ is arbitrarily large, while $\theta$ shrinks to 0.

At a high level, VI-training (with Gaussian posteriors and priors) does not have this issue because the KL-divergence penalizes the variance of the parameters from deviating from the prior in addition to the mean parameters, whereas MAP training only penalizes the means. Unlike with MAP training, if we downscale the weights, we also downscale the value of the variances, which increases the KL-divergence. The variances cannot revert to the prior either, as when they are up-scaled by the FiLM scale parameter, the noise would increase, affecting the log-likelihood component of the ELBO. Therefore, there exists an optimal amount of scaling which balances the mean-squared penalty component of the KL-divergence and the variance terms.

Mathematically we can derive this optimal scale. Consider the scenario with VI training with Gaussian variational distribution and prior, where our approximate posterior $q(\theta)$ has mean and variance $\mu$ and $\Sigma$ and our prior $p(\theta)$ has parameters $\mu_0$ and $\Sigma_0$. First consider the scenario without FiLM Layers. Now, have our loss function $L = -\mathbb{E}_{\theta \sim q(\theta)} \log p(D|\theta) + D_{KL}(q(\theta)||q_0(\theta))$. For multivariate Gaussians,

$$D_{KL}(q(\theta)||p(\theta)) = \frac{1}{2}(\log|\Sigma_0| - \log|\Sigma| - d + Tr(\Sigma_0^{-1}\Sigma) + (\mu - \mu_0)^T\Sigma_0^{-1}(\mu - \mu_0)).$$

Now consider another distribution $q'(\theta)$, with mean and variance parameters $c\mu$ and $c^2\Sigma$. Now if $q'(\theta)$ is paired with FiLM scale parameter $\gamma$ set at $\frac{1}{c}$, the log-likelihood component is unchanged:

$$\mathbb{E}_{\theta \sim q(\theta)} \log p(D|\theta) = \mathbb{E}_{\theta \sim q'(\theta)} \log p(D|\theta, \gamma = \frac{1}{c}),$$

with $\gamma$ being our FiLM scale parameter and $p(D|\theta, \gamma)$ representing a model with FiLM scale layers. Now consider the $D_{KL}(q'(\theta)||q_0(\theta))$, and optimize $c$ with $\mu$ and $\Sigma$ fixed:

$$D_{KL}(q'(\theta)||p(\theta)) = \frac{1}{2}(\log|\Sigma_0| - \log|c^2\Sigma| - d + Tr(\Sigma_0^{-1}c^2\Sigma) + (c\mu - \mu_0)^T\Sigma_0^{-1}(c\mu - \mu_0))$$

$$= \frac{1}{2}(\log|\Sigma_0| - \log|\Sigma| - 2d\log c - d + c^2Tr(\Sigma_0^{-1}\Sigma)$$
$$+ (c\mu - \mu_0)^T\Sigma_0^{-1}(c\mu - \mu_0))$$

$$\frac{\partial D_{KL}}{\partial c}|_{c=c^*} = 0 = -\frac{d}{c^*} + c^*Tr(\Sigma_0^{-1}\Sigma) + (c^*\mu - \mu_0)^T\Sigma_0^{-1}\mu$$

$$0 = -d + c^{*2}Tr(\Sigma_0^{-1}\Sigma) + c^{*2}\mu^T\Sigma_0^{-1}\mu - c^*\mu_0^T\Sigma_0^{-1}\mu$$

$$0 = c^{*2}(Tr(\Sigma_0^{-1}\Sigma) + \mu^T\Sigma_0^{-1}\mu) - c^*\mu_0^T\Sigma_0^{-1}\mu - d$$

$$\Rightarrow c^* = \frac{\mu_0^T\Sigma_0^{-1}\mu \pm \sqrt{(\mu_0^T\Sigma_0^{-1}\mu)^2 + 4d(Tr(\Sigma_0^{-1}\Sigma) + \mu^T\Sigma_0^{-1}\mu)}}{2(Tr(\Sigma_0^{-1}\Sigma) + \mu^T\Sigma_0^{-1}\mu)}.$$

Also note that $c = 0$ results in an infinitely-large KL-divergence, so there is a barrier at $c = 0$, i.e. If optimized through gradient descent, $c$ should never change sign. Furthermore, note that

$$\frac{\partial^2 D_{KL}}{\partial c^2} = \frac{d}{c^2} + Tr(\Sigma_0^{-1}\Sigma) + \mu^T\Sigma_0^{-1}\mu > 0.$$

So the KL-divergence is concave with respective to $c$, so $c^*$ is a minimizer of $D_{KL}$ and therefore

$$D_{KL}(q(\theta)||p(\theta)) \geq D_{KL}(q'(\theta)||p(\theta))|_{c=c*},$$

which implies the optimal value of the FiLM scale parameter $\gamma$ is $\frac{1}{c^*}$. While no formal data was collected, it was observed that the scale parameters do in fact reach very close to this optimal scale value after training.

## F CLUSTERING OF FiLM PARAMETERS

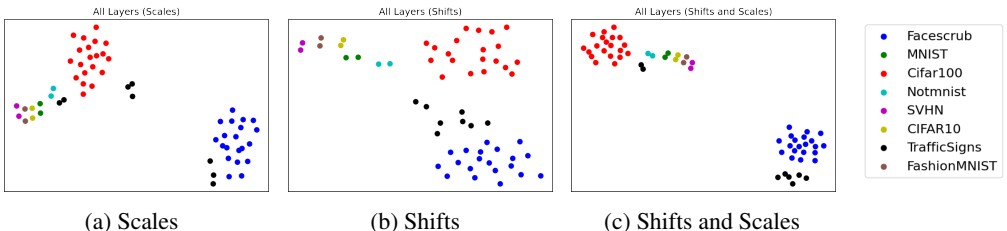

    (a) Scales            (b) Shifts            (c) Shifts and Scales

Figure 12: T-SNE of FiLM layer parameters of 58 tasks coming from different domains. Shift and scale parameters from the same domain are more similar than those from different ones.

In this section, we test the interpretability of learned FiLM Parameters. Such clustering has been done in the past with FiLM parameters, as well as node-wise uncertainty parameters. One would intuitively expect that tasks from similar domains would finds similar features salient, and thus share similar FiLM parameters. To test this hypothesis, we took the 8 mixed vision task from Section 5.3 and split each task into multi 5-way classification tasks so that there were many tasks from similar domains. For example, CIFAR100, which originally had 100 classes, became 20 5-way clasification tasks, Trafficsigns became 8 tasks (7 5-way and 1 8-way), and MNIST 2 (2 5-way). Next, we trained the same architecture used in Section 5.3 except trained all 58 resulting tasks. Joint training was chosen over continual learning to avoid artifacts which would arise from task ordering. Figure 12 shows that the results scale and shift parameters can be clustered and FiLM parameters which arise from the same base task cluster together. Like in Achille et al. (2019), this likely could be used as a means of knowing which tasks to learn continually and which tasks to separate (i.e. tasks from the same cluster would likely benefit from joint training, while tasks from different ones should be separately trained), however we did not explore this idea further.

## G HOW FiLM LAYERS INTERACT WITH PRUNING

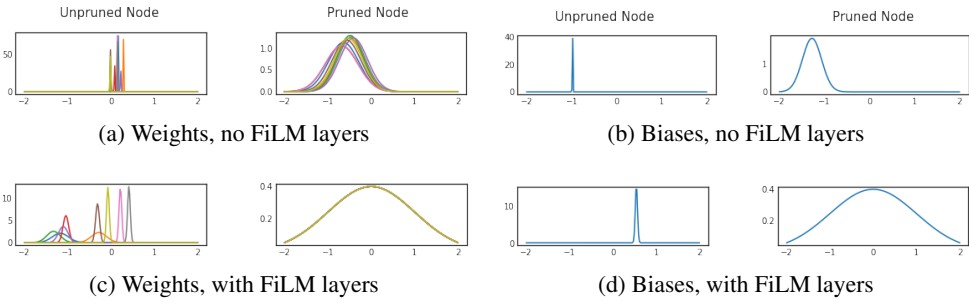

    (a) Weights, no FiLM layers            (b) Biases, no FiLM layers

    (c) Weights, with FiLM layers           (d) Biases, with FiLM layers

Figure 13: Posterior distributions for incoming weights (left) or biases (right) for a node in the first layer. Nodes are either unrpruned (left within a column) or pruned (right within a column). Without FiLM Layers (top row), we see that pruned nodes have their bias concentrated at a negative value, preventing future tasks from reactivating the node. With FiLM Layers, a pruned node prunes using the FiLM parameters rather than the shared ones, allowing the posteriors to revert to the prior distribution, allowing for node reactivation.

In Section 3, we discussed the problem of pruning in variational continual learning and how it prevents nodes from becoming reactivated. To reiterate, pruning broadly occurs in three steps:

1. Weights incoming to a node begin to revert to the prior distribution
2. Noise from these high-variance weights affect the likelihood term in the ELBO

3. To prevent noise, the bias concentrates at a negative value to be cut off by the ReLU activation

Later tasks then are initialized with this negative bias with low variance, meaning that the node has a difficult time reactivating the node without incurring a high prior cost. This results in the effect shown in Figure 1, where after the first task, effectively no more nodes are reactivated. The effect is further exacerbated with larger values of $\beta$, where the pruning effect is stronger. Increasing $\lambda$ worsens this as well, as increasing the quadratic cost further prevents already low-variance negative biases from moving.

We verify that this mechanism is indeed the cause of the limited capacity use by visualizing the posteriors for weights and biases entering a node in the first convolutional layer for a network trained on Easy-CHASY (Figure 13). Here, we see that biases in pruned nodes when there are no FiLM Layers do indeed concentrate at negative values. In contrast, biases in models with FiLM layers are able to revert to their prior because the FiLM parameters perform pruning.

## H    RELATED WORK

**Regularization-based continual learning.** Many algorithms attempt to regularize network parameters based on a metric of importance. The most directly comparable algorithms to GVCL are EWC (Kirkpatrick et al., 2017), Online EWC (Schwarz et al., 2018), and VCL (Nguyen et al., 2018). EWC measures importance based on the Fisher information matrix, while VCL uses an approximate posterior covariance matrix as an importance measure. Online EWC slightly modifies EWC so that there is only a single regularizer based on the cumulative sum of Fisher information matrices. Lee et al. (2017) proposed IMM, which is an extension to EWC which merges posteriors based on their Fisher information matrices. Ritter et al. (2018) and Yin et al. (2020) both aim to approximate the Hessian by using either Kronecker-factored or low-rank forms, using the Laplace approximation to form approximate posteriors of parameters. These methods all use second-order approximations of the loss.Ahn et al. (2019), like us, use regularizers based on the ELBO, but also measure importance on a per-node basis than a per-weight one. SI (Zenke et al., 2017) measures importance using "Synaptic Saliency," as opposed to methods based on approximate curvature.

**Architectural approaches to continual and meta-learning.** This family of methods modifies the standard neural architecture by either adding parallel or series components to the network. Progressive Neural Networks adds a parallel column network for every task. Pathnet (Fernando et al., 2017) can be interpreted as a parallel-network based algorithm, but rather than growing model size over time, the model size remains fixed while paths between layer columns are optimized. FiLM parameters can be interpreted as adding series components to a network, and has been a mainstay in the multitask and meta-learning literature. Requeima et al. (2019) use hypernetworks to amortize FiLM parameter learning, and has been shown to be capable of continual learning. Architectural approaches are often used in tandem with regularization based approaches, such as in HAT (Serra et al., 2018), which uses per-task gating parameters alongside a compression-based regularizer. Adel et al. (2020) propose CLAW, which also uses variational inference alongside per-task parameters, but requires a more complex meta-learning based training procedure involving multiple splits of the dataset. GVCL with FiLM layers adds to this list of hybrid architectural-regularization based approaches.

**Cold Posteriors and likelihood-tempering.** As mentioned in Section 2, likelihood-tempering (or KL-reweighting) has been empirically found to improve performance when using variational inference for Bayesian Neural Networks over a wide number of contexts and papers (Osawa et al., 2019; Zhang et al., 2018). Cold posteriors are closely related to likelihood tempering, except they temper the full posterior rather than only the likelihood term, and often empirically outperform Bayesian posteriors when using MCMC sampling Wenzel et al. (2020). From an information-theoretic perspective, KL-reweighted ELBOs have also studied as compression (Achille et al., 2020). Achille et al. (2019), like us, considers a limiting case of $\beta$, and uses this to measure parameter saliency, but use this information to create a task embedding rather than for continual learning. Outside of the Bayesian Neural Network context, values of $\beta > 1$ have also been explored (Higgins et al., 2017), and more generally different values of $\beta$ trace out different points on a rate-distortion curve for VAEs (Alemi et al., 2018).

# I EXPERIMENT DETAILS

## I.1 REPORTED METRICS

All reported scores and figures present the mean and standard deviation across 5 runs of the algorithm with a different network initialization. For Easy-CHASY and Hard-CHASY, train/test splits are also varied across iterations. For the Mixed Vision tasks, task permutation of the 8 tasks is also randomized between iterations.

Let the matrix $R_{i,j}$ represent the performance of $j$th task after the model was trained on the $i$th task. Furthermore, let $R_j^{ind}$ be the mean performance of the $j$th for a network trained only on that task and let the total number of tasks be $T$. Following Lopez-Paz & Ranzato (2017) and Pan et al. (2020), we define

$$\text{Average Accuracy (ACC)} = \frac{1}{T} \sum_{j=1}^{T} R_{T,j},$$

$$\text{Forward Transfer (FWT)} = \frac{1}{T} \sum_{j=1}^{T} R_{j,j} - R_j^{ind},$$

$$\text{Backward Transfer (BWT)} = \frac{1}{T} \sum_{j=1}^{T} R_{T,j} - R_{j,j}.$$

Note that these metrics are not exactly the same as those presented in all other works, as the FWT and BWT metrics are summed over the indices $1 \leq j \leq T$, whereas Lopez-Paz & Ranzato (2017) and Pan et al. (2020) sum from $2 \leq j \leq T$ and $1 \leq j \leq T - 1$ for FWT and BWT, respectively. For FWT, this definition does not assumes that $R_{1,1} = R_1^{ind}$, and affects algorithms such as HAT and Progressive Neural Networks, which either compress the model, resulting in lower accuracy, or use a smaller architecture for the first task. The modified BWT transfer is equal to the other BWT metrics apart from a constant factor $\frac{T-1}{T}$.

Intuitively, forward transfer equates to how much continual learning has benefited a task when a task is newly learned, while backwards transfer is the accuracy drop as the network learns more tasks compared to when a task was first learned. Furthermore, in the tables in Appendix J, we also present net performance gain (NET), which quantifies the total gain over separate training, at the end of training continually:

$$\text{NET} = \text{FWT} + \text{BWT} = \frac{1}{T} \sum_{j=1}^{T} R_{T,j} - R_j^{ind}.$$

Note that for computation of $R^{ind}$, we compare to models trained under the same paradigm, i.e. MAP algorithms (all baselines except for VCL) are compared to a MAP trained model, and VI algorithms (GVCL-F, GVCL and VCL) are compared to KL-reweighted VI models. This does not make a difference for most of the benchmarks where $R_{\text{MAP}}^{ind} \approx R_{\text{VI}}^{ind}$. However, for Easy and Hard-CHASY, $R_{\text{MAP}}^{ind} < R_{\text{VI}}^{ind}$, so we compare VI to VI and MAP to MAP to obtain fair metrics.

In Figure 5b, we plot $\Delta\text{ACC}_i$, which we define as

$$\Delta\text{ACC}_i = \frac{1}{i} \sum_{j=1}^{i} R_{i,j} - R_{T,j}.$$

This metric is useful when the tasks have very different accuracies and their permutation is randomized, as is the case with the mixed vision tasks. Note that this means that $R_{i,j}$ would refer to a different task for each permutation, but we average over the 5 permutations of the runs. Empirically,

if two algorithms have similar final accuracies, this metric measures how much the network forgets about the first $i$ tasks from that point to the end, and also measures how high the accuracy would have been if training was terminated after $i$ tasks. Plotting this also captures the concept of graceful vs catastrophic forgetting, as graceful forgetting would show up as a smooth downward curve, while catastrophic forgetting would have sudden drops.

## I.2 OPTIMIZER AND TRAINING DETAILS

The implementation of all baseline methods was based on the Github repository[6] for HAT (Serra et al., 2018), except the implementions of IMM-Mode and EWC were modified due to an error in the computation of the Fisher Information Matrix in the original implementation. Baseline MAP algorithms were trained with SGD with a decaying learning starting at 5e-2 with a maximum epochs of 200 per task for the *Split*-MNIST, *Split*-CIFAR and the mixed vision benchmarks. The number of maximum epochs for Easy-CHASY and Hard-CHASY was 1000, due to the small dataset size. Early stopping based on the validation set was used. 10% of the training set was used as validation for these methods, and for Easy and Hard CHASY, 8 samples per class form the validation set (which are disjoint from the training samples or test samples).

For VI models, we used Adam optimizer with a learning rate of 1e-4 for *Split*-MNIST and Mixture, and 1e-3 for Easy-CHASY, Hard-CHASY and *Split*-CIFAR. We briefly tested running the baselines algorithms using Adam rather than SGD and performance did not change. Easy-CHASY and Hard-CHASY were run for 1500 epochs per task, *Split*-MNIST for 100, *Split*-CIFAR for 60, and 180 for Mixture. The number of epochs was changed so that the number of gradient steps for each task was roughly equal. For Easy-CHASY, Hard-CHASY and *Split*-CIFAR, this means that later tasks are run for more epochs, since the largest training sets are at the start. For Mixture, we ran 180 equivalents epochs for Facescrub. For how many epochs this equates to in the other datasets, we refer the reader to Appendix A in Serra et al. (2018). We did not use early stopping for these VI results. While we understand that in some cases we trained for many more epochs than the baselines, the baselines used early stopping and therefore all stopped long before the 200 epoch limit was reached, so allocating more time would not change their results. Swaroop et al. (2019) also finds that allowing VI to converge is crucial for continual learning performance. We leave the discussion of improving this convergence time for future work.

All experiments (both the baselines and VI methods) use a batch size of 64.

## I.3 ARCHITECTURAL DETAILS

**Easy and Hard CHASY.** We use a convolutional architecture with 2 convolutions layers with:

1. 3x3 convolutional layer with 16 filters, padding of 1, ReLU activations
2. 2x2 Max Pooling with stride 2
3. 3x3 convolutional layer with 32 filters, padding of 1, ReLU activations
4. 2x2 Max Pooling with stride 2
5. Flattening layer
6. Fully connected layer with 100 units and ReLU activations
7. Task-specific head layers

*Split*-**MNIST.** We use a standard MLP with:

1. Fully connected layer with 256 units and ReLU activations
2. Fully connected layer with 256 units and ReLU activations
3. Task-specific head layers

*Split*-**CIFAR.** We use the same architecture from Zenke et al. (2017):

1. 3x3 convolutional layer with 32 filters, padding of 1, ReLU activations

---

[6]Repository at `https://github.com/joansj/hat`

2. 3x3 convolutional layer with 32 filters, padding of 1, ReLU activations

3. 2x2 Max Pooling with stride 2

4. 3x3 convolutional layer with 64 filters, padding of 1, ReLU activations

5. 3x3 convolutional layer with 64 filters, padding of 1, ReLU activations

6. 2x2 Max Pooling with stride 2

7. Flattening

8. Fully connected layer with 512 units and ReLU activations

9. Task-specific head layers

**Mixed vision tasks.** We use the same AlexNet architecture from Serra et al. (2018):

1. 4x4 convolutional layer with 64 filters, padding of 0, ReLU activations

2. 2x2 Max Pooling with stride 2

3. 3x3 convolutional layer with 128 filters, padding of 0, ReLU activations

4. 2x2 Max Pooling with stride 2

5. 2x2 convolutional layer with 256 filters, padding of 0, ReLU activations

6. 2x2 Max Pooling with stride 2

7. Flattening

8. Fully connected layer with 2048 units and ReLU activations

9. Fully connected layer with 2048 units and ReLU activations

10. Task-specific head layers

For MAP models, dropout layers with probabilities of either 0.2 or 0.5 were added after convolutional or fully-connected layers. For GVCL-F, FiLM layers were inserted after convolutional/hidden layers, but before ReLU activations.

## I.4   HYPERPARAMETER SELECTION

For all algorithms on Easy-CHASY, Hard-CHASY, *Split*-MNIST and *Split*-CIFAR, hyperparameter selection was done by selecting the combination which produced the best average accuracy on the first 3 tasks. The algorithms were then run on the full number of tasks. For the Mixed Vision tasks, the best hyperparameters for the baselines were taken from the HAT Github repository. For GVCL, we performed hyperparameter selection in the same way as in Serra et al. (2018): we found the best hyperparameters for the average performance on the first random permutation of tasks. Note that in the mixture tasks, we randomly permute the task order for each iteration (with permutations kept consistent between algorithms), whereas for the other 4 benchmarks, the task order is fixed. Hyperparameter searches were performed using a grid search. The best selected hyperparameters are shown in Table 3.

| Algorithm | Hyperparameter | Easy-CHASY | Hard-CHASY | *Split*-MNIST | *Split*-CIFAR | Mixed Vision |
|---|---|---|---|---|---|---|
| GVCL-F | $\beta$ | 0.05 | 0.05 | 0.1 | 0.2 | 0.1 |
| | $\lambda$ | 10 | 10 | 100 | 100 | 50 |
| GVCL | $\beta$ | 0.05 | 0.05 | 0.1 | 0.2 | 0.1 |
| | $\lambda$ | 100 | 100 | 1 | 1000 | 100 |
| HAT | $\lambda$ | 1 | 1 | 0.1 | 0.025 | 0.75* |
| | $s_{max}$ | 10 | 50 | 50 | 50 | 400* |
| PathNet | # of evolutions | 20 | 200 | 10 | 100 | 20* |
| VCL | None | - | - | - | - | - |
| Online EWC | $\lambda$ | 100 | 500 | 10000 | 100 | 5 |
| Progressive | None | - | - | - | - | - |
| IMM-Mean | $\lambda$ | 0.0005 | 1e-6 | 5e-4 | 1e-4 | 0.0001* |
| IMM-Mode | $\lambda$ | 1e-7 | 0.1 | 0.1 | 1e-5 | 1 |
| LWF | $\lambda$ | 0.5 | 0.5 | 2 | 2 | 2* |
| | $T$ | 4 | 2 | 4 | 4 | 1* |

* Best hyperparameters taken from HAT code

Table 3: Best (selected) hyperparameters for continual learning experiments for various algorithms. We fix Online EWC's $\gamma = 1$.

For the Joint and Separate VI baselines, we used the same $\beta$. For the mixed vision tasks, we had to used a prior variance of 0.01 (for both VCl, GVCL and GVCL-F), but for all other tasks we did not need to tune this.

## J  FURTHER EXPERIMENTAL RESULTS

In following section we present more quantitative results of the various baselines on our benchmarks. For brevity, in the main text, we only included the best performing baselines and those which are most comparable to GVCL, which consisted of HAT, PathNet, Online EWC and VCL.

## J.1 EASY-CHASY ADDITIONAL RESULTS

| Metric | ACC (%) | BWT (%) | FWT (%) | NET (%) |
|---|---|---|---|---|
| GVCL-F | $\mathbf{90.9 \pm 0.3}$ | $\mathbf{0.2 \pm 0.1}$ | $0.4 \pm 0.3$ | $\mathbf{0.6 \pm 0.3}$ |
| GVCL | $88.9 \pm 0.6$ | $-0.8 \pm 0.4$ | $-0.6 \pm 0.5$ | $-1.4 \pm 0.6$ |
| | | | | |
| HAT | $82.6 \pm 0.9$ | $-1.6 \pm 0.6$ | $0.4 \pm 1.4$ | $-1.3 \pm 0.9$ |
| PathNet | $82.4 \pm 0.9$ | $0.0 \pm 0.0$ | $-1.5 \pm 0.9$ | $-1.5 \pm 0.9$ |
| VCL | $78.4 \pm 1.0$ | $-4.1 \pm 1.2$ | $-7.9 \pm 0.8$ | $-11.9 \pm 1.0$ |
| VCL-F | $79.9 \pm 1.0$ | $-6.1 \pm 0.9$ | $-4.3 \pm 0.3$ | $-10.4 \pm 1.0$ |
| Online EWC | $73.4 \pm 3.4$ | $-8.9 \pm 2.9$ | $-1.5 \pm 0.5$ | $-10.5 \pm 3.4$ |
| Online EWC-F | $76.0 \pm 1.5$ | $-6.9 \pm 1.6$ | $-1.0 \pm 0.3$ | $-7.9 \pm 1.5$ |
| Progressive | $82.6 \pm 0.6$ | $0.0 \pm 0.0$ | $-1.3 \pm 0.6$ | $-1.3 \pm 0.6$ |
| IMM-mean | $42.3 \pm 1.0$ | $-1.1 \pm 0.6$ | $-40.6 \pm 1.1$ | $-41.6 \pm 1.0$ |
| imm-mode | $74.8 \pm 1.0$ | $-11.2 \pm 0.1$ | $2.1 \pm 0.9$ | $-9.1 \pm 1.0$ |
| LWF | $75.1 \pm 2.4$ | $-12.9 \pm 1.9$ | $\mathbf{4.1 \pm 0.6}$ | $-8.8 \pm 2.4$ |
| SGD | $75.3 \pm 1.8$ | $-11.1 \pm 0.9$ | $2.5 \pm 1.0$ | $-8.6 \pm 1.8$ |
| SGD-Frozen | $81.2 \pm 0.8$ | $0.0 \pm 0.0$ | $-2.7 \pm 0.8$ | $-2.7 \pm 0.8$ |
| | | | | |
| Separate (MAP) | $88.4 \pm 0.8$ | - | - | $0.0 \pm 0.0$ |
| Separate ($\beta$-VI) | $90.3 \pm 0.1$ | - | - | $0.0 \pm 0.0$ |
| | | | | |
| Joint (MAP) | $88.6 \pm 0.7$ | - | - | $4.7 \pm 0.7$ |
| Joint ($\beta$-VI + FiLM) | $91.9 \pm 0.1$ | - | - | $1.6 \pm 0.1$ |

Table 4: Performance metrics of GVCL-F, GVCL and various baseline algorithms on Easy-CHASY. Separate and joint training results for both MAP and $\beta$-VI models are also presented

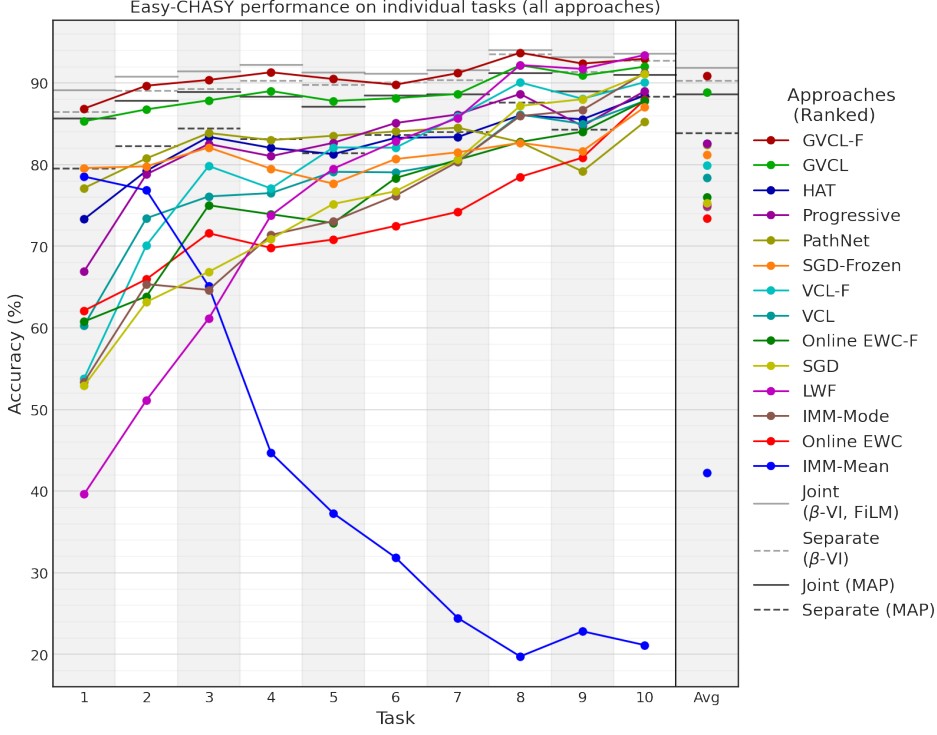

Figure 14: Mean accuracy of individual tasks after training for all approaches on Easy-CHASY

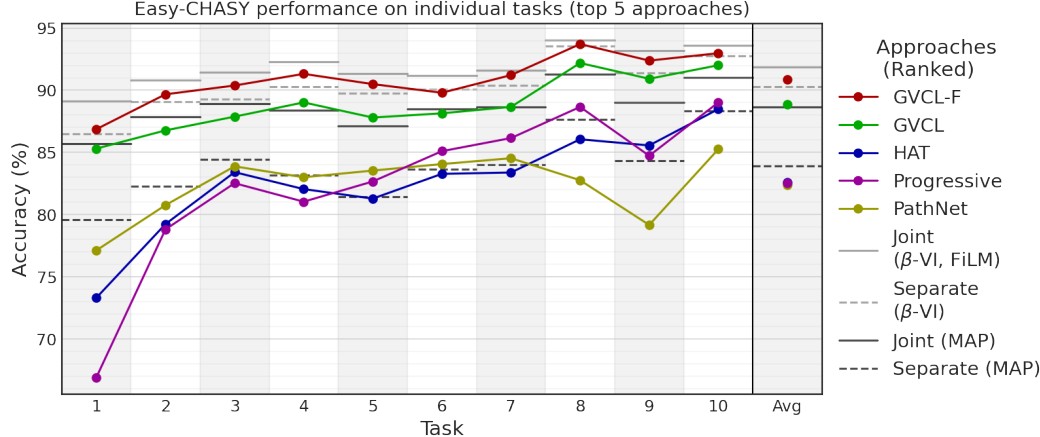

Figure 15: Mean accuracy of individual tasks after training for the top 5 performing approaches on Easy-CHASY

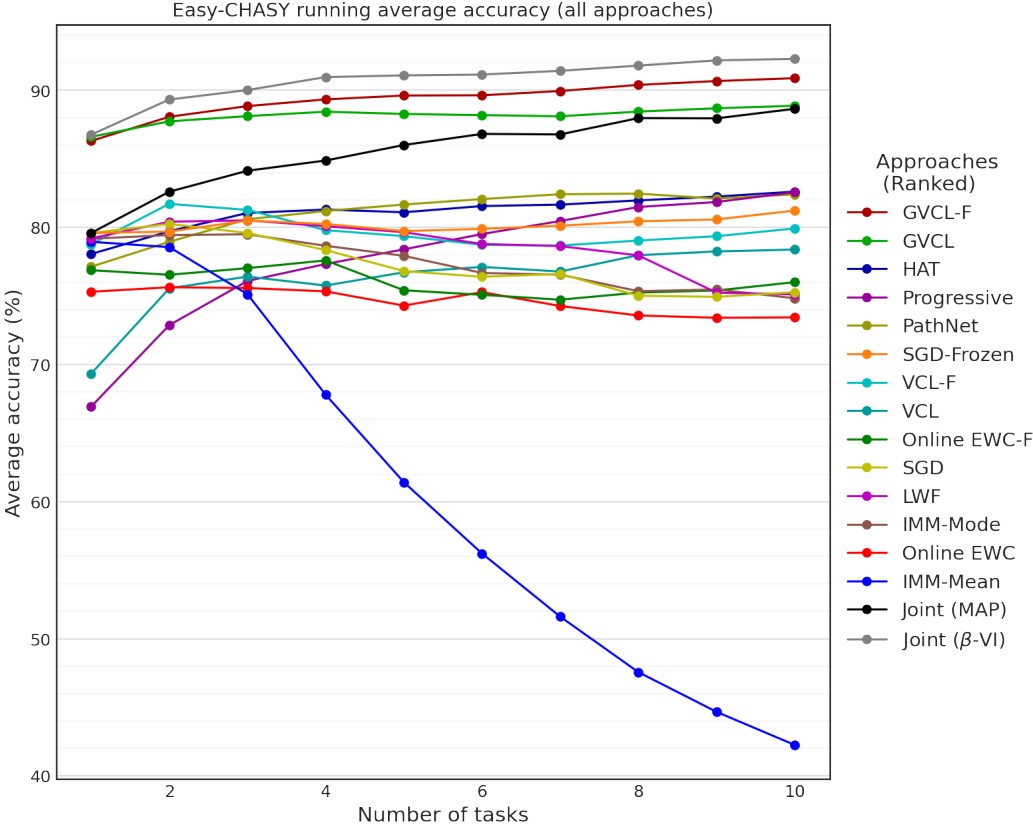

Figure 16: Running average accuracy of individual tasks after training for the all approaches on Easy-CHASY

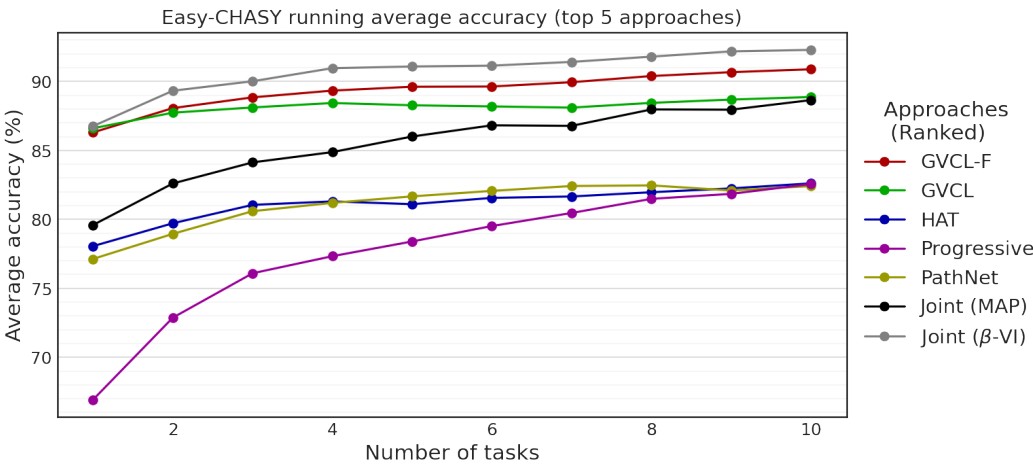

Figure 17: Running average accuracy of individual tasks after training for the top 5 approaches on Easy-CHASY

## J.2   HARD-CHASY ADDITIONAL RESULTS

| Metric | ACC (%) | BWT (%) | FWT (%) | NET (%) |
|---|---|---|---|---|
| GVCL-F | $\mathbf{69.5 \pm 0.6}$ | $-0.1 \pm 0.1$ | $-1.6 \pm 0.7$ | $-1.7 \pm 0.6$ |
| GVCL | $64.4 \pm 0.6$ | $-0.6 \pm 0.2$ | $-6.3 \pm 0.6$ | $-6.8 \pm 0.6$ |
| | | | | |
| HAT | $62.5 \pm 5.4$ | $-0.8 \pm 0.4$ | $-3.7 \pm 5.5$ | $-4.5 \pm 5.4$ |
| PathNet | $64.8 \pm 0.8$ | $0.0 \pm 0.0$ | $-2.2 \pm 0.8$ | $-2.2 \pm 0.8$ |
| VCL | $45.8 \pm 1.4$ | $-11.9 \pm 1.6$ | $-13.5 \pm 2.2$ | $-25.4 \pm 1.4$ |
| VCL-F | $65.0 \pm 0.8$ | $-2.7 \pm 0.8$ | $-3.4 \pm 0.6$ | $-6.1 \pm 0.8$ |
| Online EWC | $56.4 \pm 1.7$ | $-7.1 \pm 1.7$ | $-3.4 \pm 1.3$ | $-10.5 \pm 1.7$ |
| Online EWC-F | $56.7 \pm 6.4$ | $-8.8 \pm 5.9$ | $-1.4 \pm 0.9$ | $-10.2 \pm 6.4$ |
| Progressive | $65.2 \pm 1.6$ | $0.0 \pm 0.0$ | $-1.8 \pm 1.6$ | $-1.8 \pm 1.6$ |
| IMM-mean | $35.5 \pm 0.8$ | $-1.0 \pm 0.8$ | $-30.5 \pm 1.2$ | $-31.5 \pm 0.8$ |
| imm-mode | $44.3 \pm 4.3$ | $-22.2 \pm 5.4$ | $-0.5 \pm 1.1$ | $-22.7 \pm 4.3$ |
| LWF | $46.4 \pm 2.5$ | $-23.0 \pm 2.8$ | $2.4 \pm 1.0$ | $-20.6 \pm 2.5$ |
| SGD | $47.1 \pm 2.2$ | $-21.0 \pm 2.7$ | $1.2 \pm 0.7$ | $-19.8 \pm 2.2$ |
| SGD-Frozen | $61.6 \pm 1.4$ | $0.0 \pm 0.0$ | $-5.3 \pm 1.4$ | $-5.3 \pm 1.4$ |
| | | | | |
| Separate (MAP) | $54.1 \pm 1.2$ | - | - | $0.0 \pm 0.0$ |
| Separate ($\beta$-VI) | $71.2 \pm 0.5$ | - | - | $0.0 \pm 0.0$ |
| | | | | |
| Joint (MAP) | $66.4 \pm 0.6$ | - | - | $-0.6 \pm 0.6$ |
| Joint ($\beta$-VI + FiLM) | $70.4 \pm 0.8$ | - | - | $-0.8 \pm 0.8$ |

Table 5: Performance metrics of GVCL-F, GVCL and various baseline algorithms on Hard-CHASY. Separate and joint training results for both MAP and $\beta$-VI models are also presented

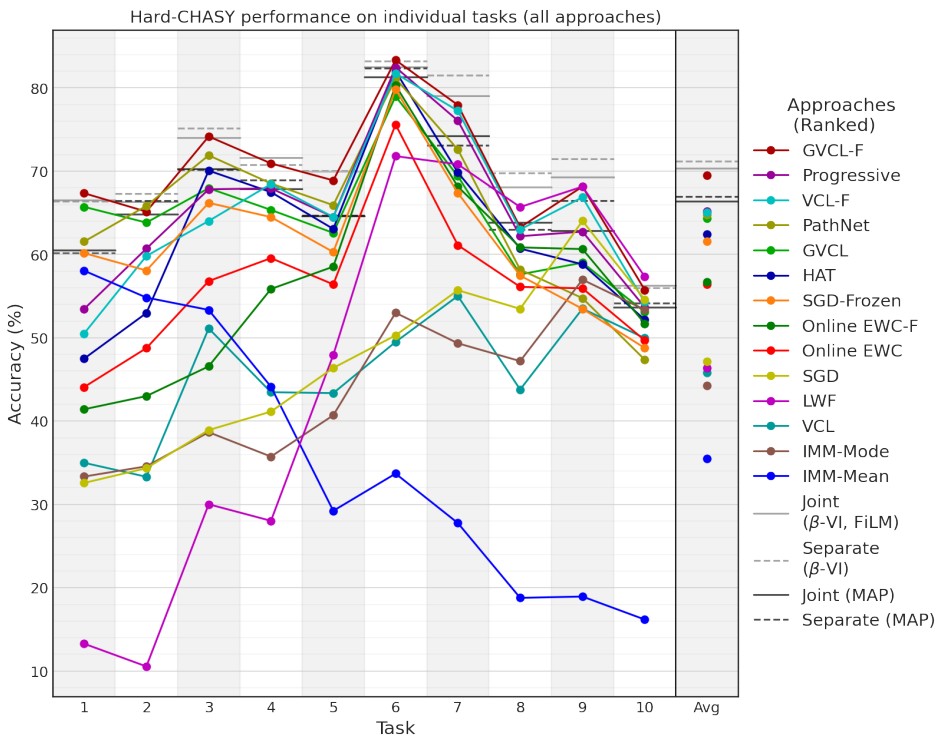

Figure 18: Mean accuracy of individual tasks after training for all approaches on Hard-CHASY

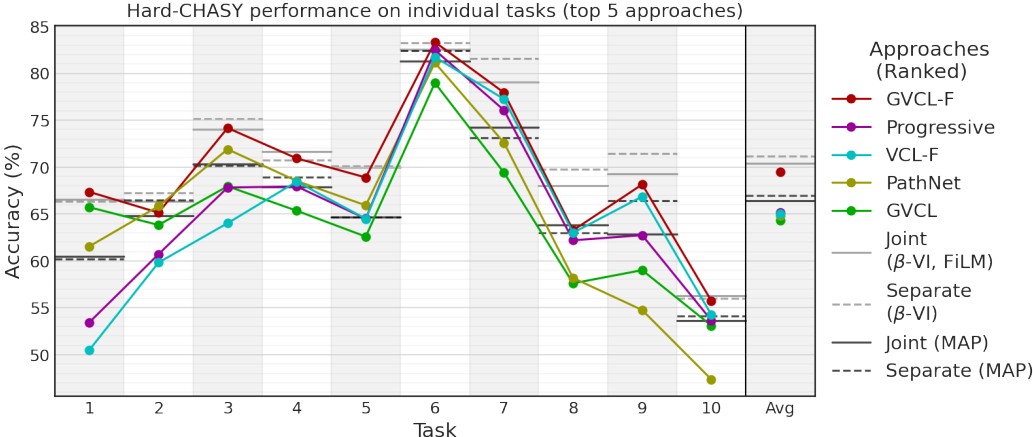

Figure 19: Mean accuracy of individual tasks after training for the top 5 performing approaches on Hard-CHASY

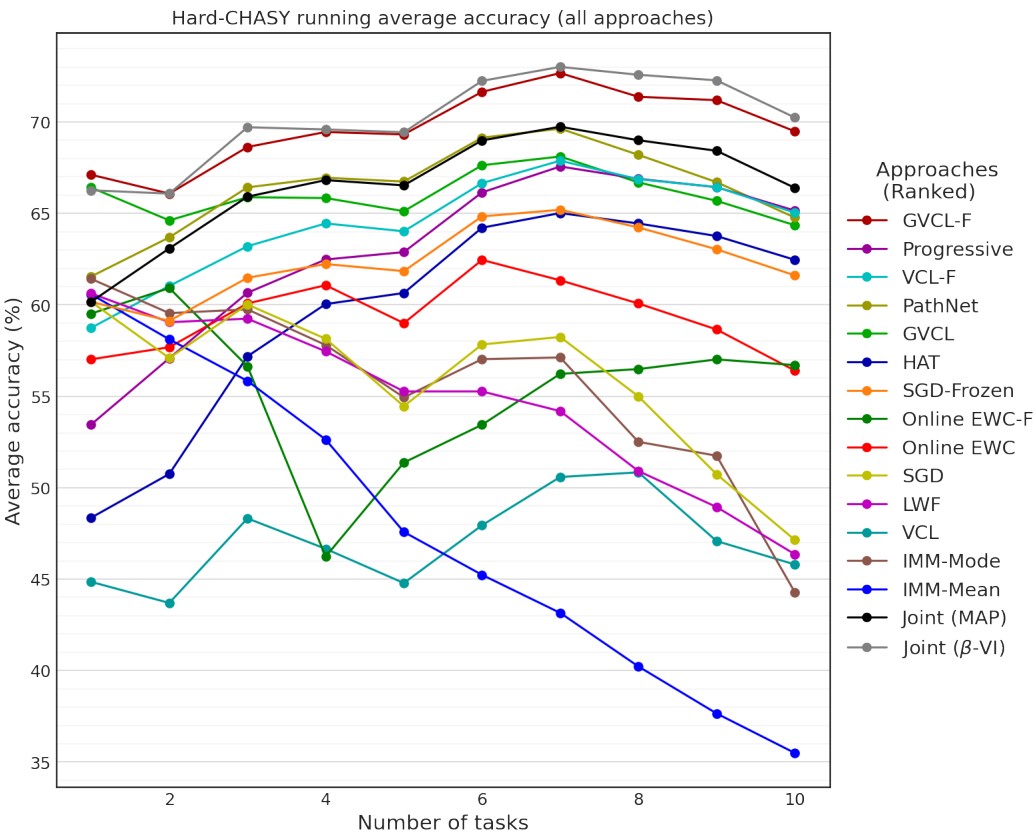

Figure 20: Running average accuracy of individual tasks after training for the all approaches on Hard-CHASY

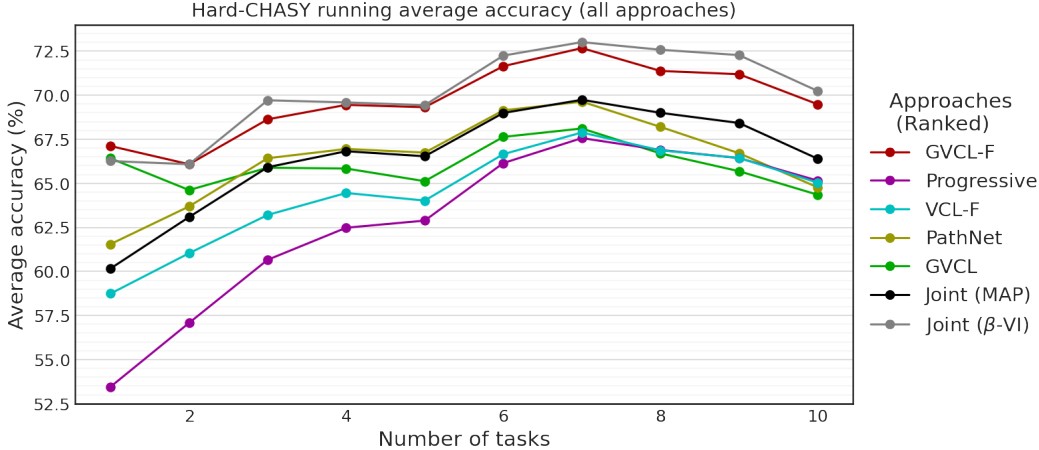

Figure 21: Running average accuracy of individual tasks after training for the top 5 approaches on Hard-CHASY

## J.3  *Split*-MNIST ADDITIONAL RESULTS

| Metric | ACC (%) | BWT (%) | FWT (%) | NET (%) |
|---|---|---|---|---|
| GVCL-F | **98.6 ± 0.1** | 0.0 ± 0.0 | −0.1 ± 0.1 | **−0.0 ± 0.1** |
| GVCL | 94.6 ± 0.7 | −4.0 ± 0.7 | −0.0 ± 0.0 | −4.1 ± 0.7 |
| | | | | |
| HAT | 98.3 ± 0.1 | −0.2 ± 0.0 | −0.1 ± 0.1 | −0.3 ± 0.1 |
| PathNet | 95.2 ± 1.8 | 0.0 ± 0.0 | −3.3 ± 1.8 | −3.3 ± 1.8 |
| VCL | 92.4 ± 1.2 | −5.5 ± 1.1 | −0.8 ± 0.1 | −6.3 ± 1.2 |
| VCL-F | 94.8 ± 0.9 | −3.3 ± 0.9 | −0.6 ± 0.1 | −3.9 ± 0.9 |
| Online EWC | 94.0 ± 1.4 | −3.8 ± 1.4 | −0.8 ± 0.1 | −4.6 ± 1.4 |
| Online EWC-F | 94.1 ± 0.7 | −0.3 ± 0.6 | −4.1 ± 0.3 | −4.4 ± 0.7 |
| Progressive | 98.4 ± 0.0 | 0.0 ± 0.0 | −0.2 ± 0.0 | −0.2 ± 0.0 |
| IMM-mean | 90.5 ± 1.1 | **0.5 ± 0.1** | −8.5 ± 1.2 | −8.0 ± 1.1 |
| imm-mode | 95.4 ± 0.2 | −1.7 ± 0.3 | −1.5 ± 0.1 | −3.1 ± 0.2 |
| LWF | 97.4 ± 0.2 | −1.1 ± 0.1 | −0.1 ± 0.1 | −1.2 ± 0.2 |
| SGD | 76.2 ± 1.7 | −22.4 ± 1.7 | 0.0 ± 0.1 | −22.4 ± 1.7 |
| SGD-Frozen | 91.7 ± 0.2 | 0.0 ± 0.0 | −6.9 ± 0.2 | −6.9 ± 0.2 |
| | | | | |
| Separate (MAP) | 98.6 ± 0.0 | - | - | 0.0 ± 0.0 |
| Separate ($\beta$-VI) | 98.7 ± 0.0 | - | - | 0.0 ± 0.0 |
| | | | | |
| Joint (MAP) | 98.7 ± 0.0 | - | - | 0.1 ± 0.0 |
| Joint ($\beta$-VI + FiLM) | 98.8 ± 0.0 | - | - | 0.1 ± 0.0 |

Table 6: Performance metrics of GVCL-F, GVCL and various baseline algorithms on *Split*-MNIST. Separate and joint training results for both MAP and $\beta$-VI models are also presented

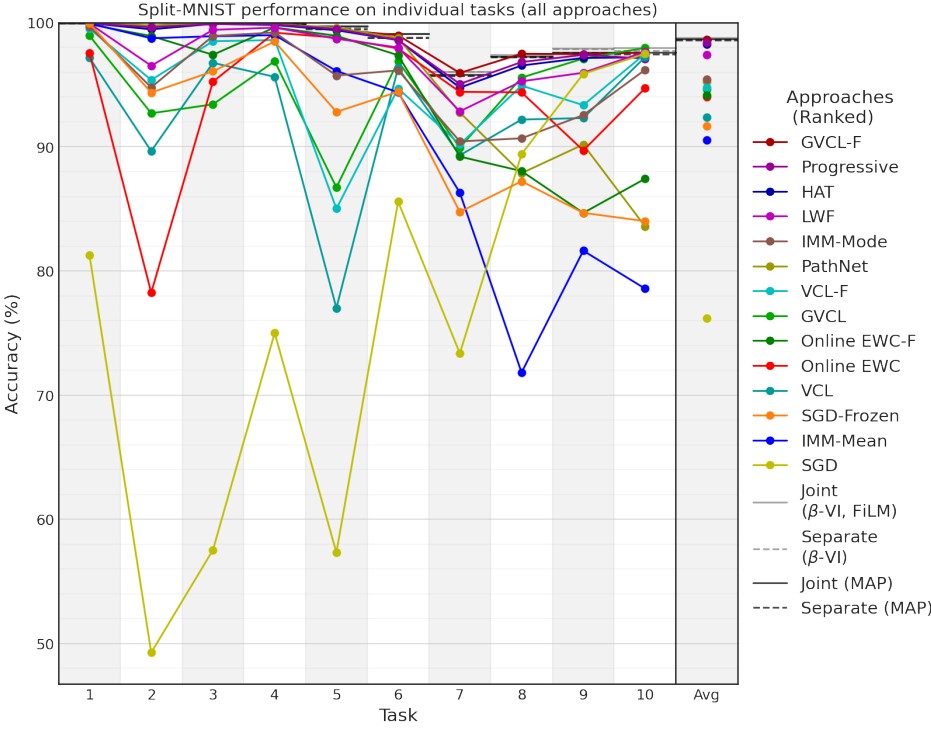

Figure 22: Mean accuracy of individual tasks after training for all approaches on *Split*-MNIST

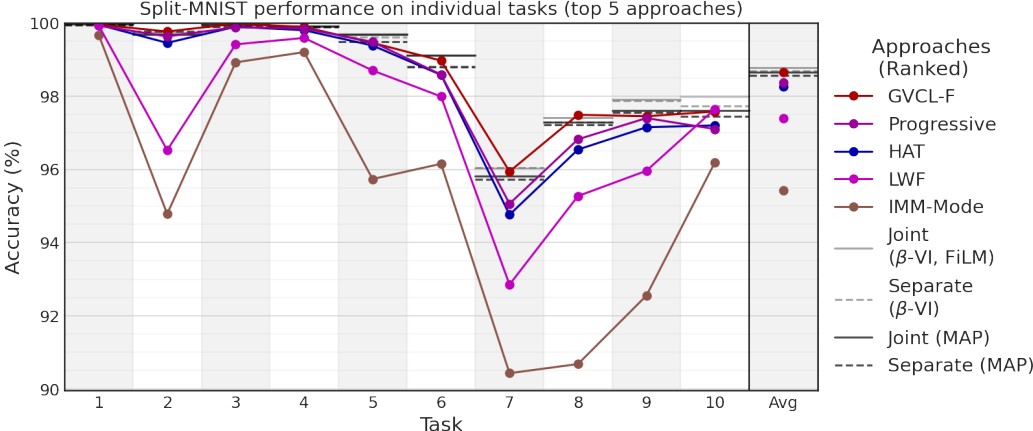

Figure 23: Mean accuracy of individual tasks after training for the top 5 performing approaches on *Split*-MNIST

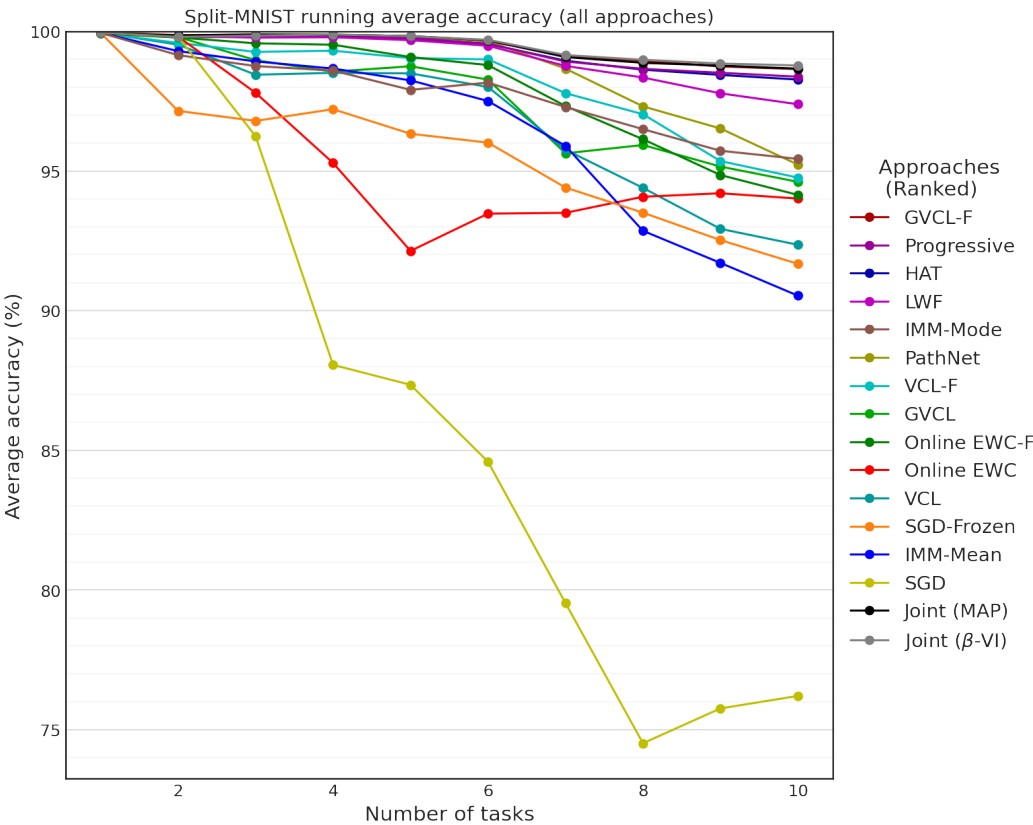

Figure 24: Running average accuracy of individual tasks after training for the all approaches on *Split*-MNIST

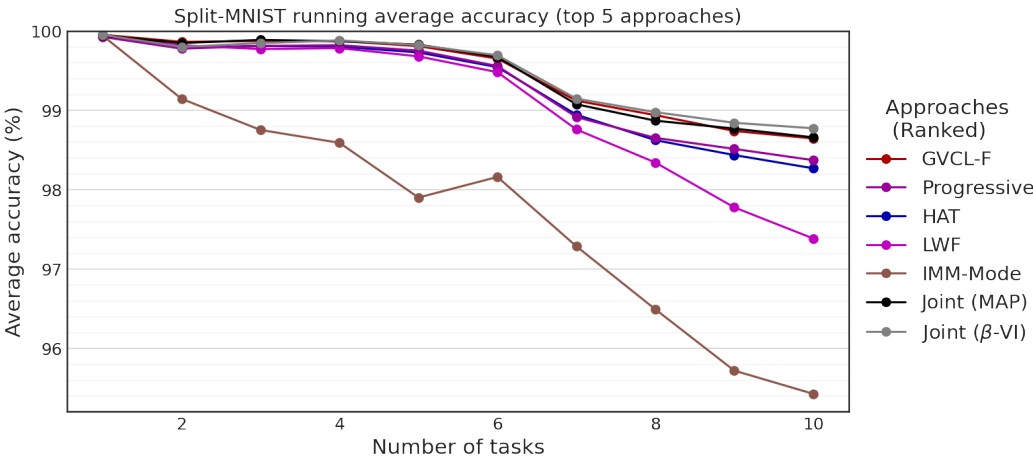

Figure 25: Running average accuracy of individual tasks after training for the top 5 approaches on *Split*-MNIST

### J.4 *Split*-CIFAR ADDITIONAL RESULTS

| Metric | ACC (%) | BWT (%) | FWT (%) | NET (%) |
|---|---|---|---|---|
| GVCL-F | $\mathbf{80.0 \pm 0.5}$ | $-0.3 \pm 0.2$ | $8.8 \pm 0.5$ | $\mathbf{8.5 \pm 0.5}$ |
| GVCL | $70.6 \pm 1.7$ | $-2.3 \pm 1.4$ | $1.3 \pm 1.0$ | $-1.0 \pm 1.7$ |
| HAT | $77.3 \pm 0.3$ | $-0.1 \pm 0.1$ | $6.8 \pm 0.2$ | $6.7 \pm 0.3$ |
| PathNet | $68.7 \pm 0.8$ | $0.0 \pm 0.0$ | $-1.9 \pm 0.8$ | $-1.9 \pm 0.8$ |
| VCL | $44.2 \pm 14.2$ | $-23.9 \pm 12.2$ | $-3.5 \pm 2.1$ | $-27.4 \pm 14.2$ |
| VCL-F | $56.2 \pm 2.8$ | $-19.5 \pm 3.2$ | $4.1 \pm 0.8$ | $-15.4 \pm 2.8$ |
| Online EWC | $77.1 \pm 0.2$ | $-0.5 \pm 0.3$ | $6.9 \pm 0.3$ | $6.4 \pm 0.2$ |
| Online EWC-F | $77.1 \pm 0.2$ | $-0.4 \pm 0.2$ | $6.9 \pm 0.3$ | $6.5 \pm 0.2$ |
| Progressive | $70.7 \pm 0.8$ | $0.0 \pm 0.0$ | $0.1 \pm 0.8$ | $0.1 \pm 0.8$ |
| IMM-mean | $67.6 \pm 0.6$ | $-0.2 \pm 0.3$ | $-2.9 \pm 0.8$ | $-3.1 \pm 0.6$ |
| imm-mode | $74.9 \pm 0.3$ | $-6.2 \pm 0.3$ | $10.5 \pm 0.4$ | $4.3 \pm 0.3$ |
| LWF | $73.8 \pm 0.9$ | $-8.0 \pm 0.8$ | $\mathbf{11.2 \pm 0.2}$ | $3.2 \pm 0.9$ |
| SGD | $74.7 \pm 0.4$ | $-6.5 \pm 0.4$ | $\mathbf{10.6 \pm 0.8}$ | $4.1 \pm 0.4$ |
| SGD-Frozen | $70.3 \pm 0.4$ | $0.0 \pm 0.0$ | $-0.3 \pm 0.4$ | $-0.3 \pm 0.4$ |
| Separate (MAP) | $70.6 \pm 0.6$ | - | - | $0.0 \pm 0.0$ |
| Separate ($\beta$-VI) | $71.6 \pm 0.2$ | - | - | $0.0 \pm 0.0$ |
| Joint (MAP) | $80.9 \pm 0.3$ | - | - | $10.2 \pm 0.3$ |
| Joint ($\beta$-VI + FiLM) | $79.8 \pm 1.0$ | - | - | $8.2 \pm 1.0$ |

Table 7: Performance metrics of GVCL-F, GVCL and various baseline algorithms on *Split*-CIFAR. Separate and joint training results for both MAP and $\beta$-VI models are also presented

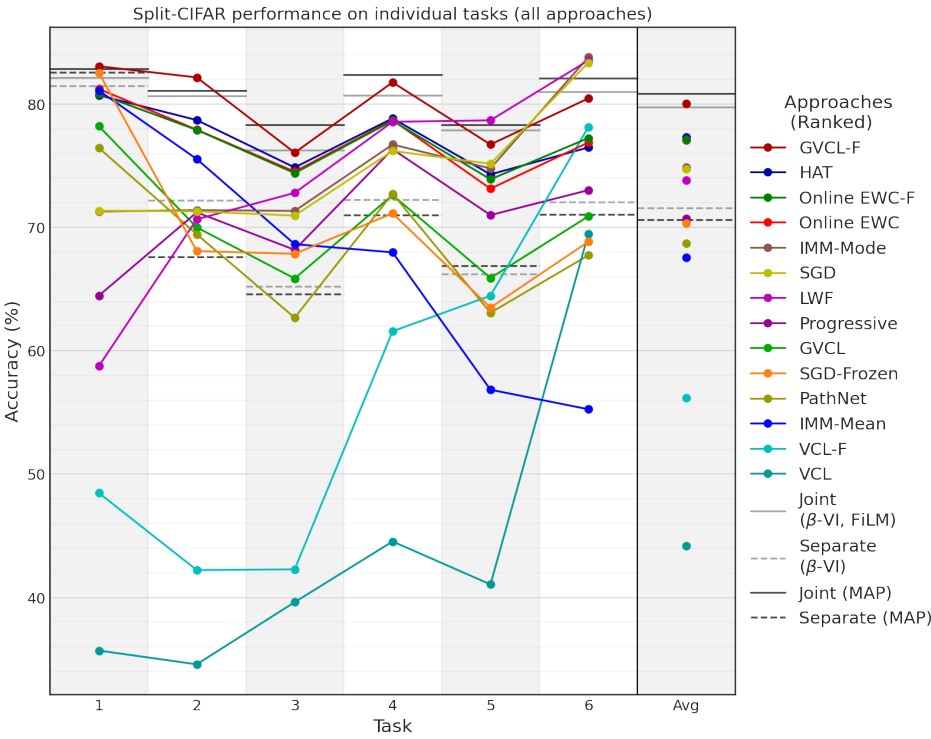

Figure 26: Mean accuracy of individual tasks after training for all approaches on *Split*-CIFAR

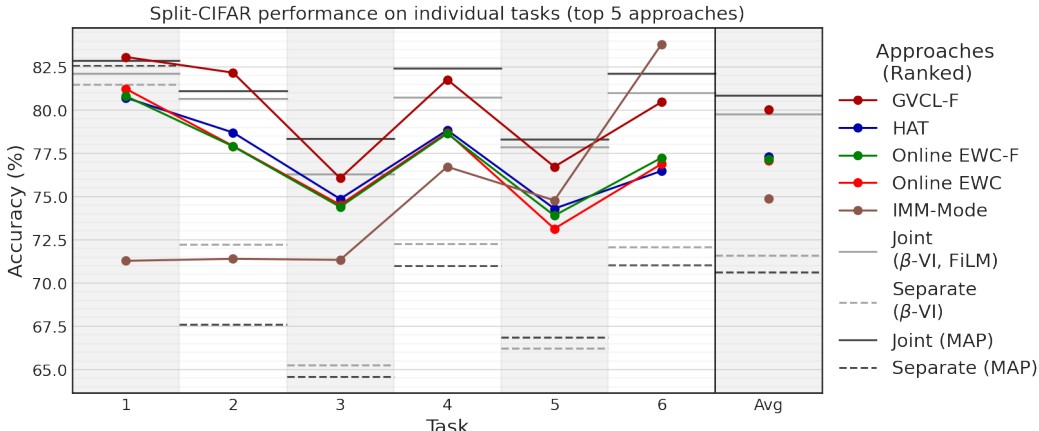

Figure 27: Mean accuracy of individual tasks after training for the top 5 performing approaches on *Split*-CIFAR

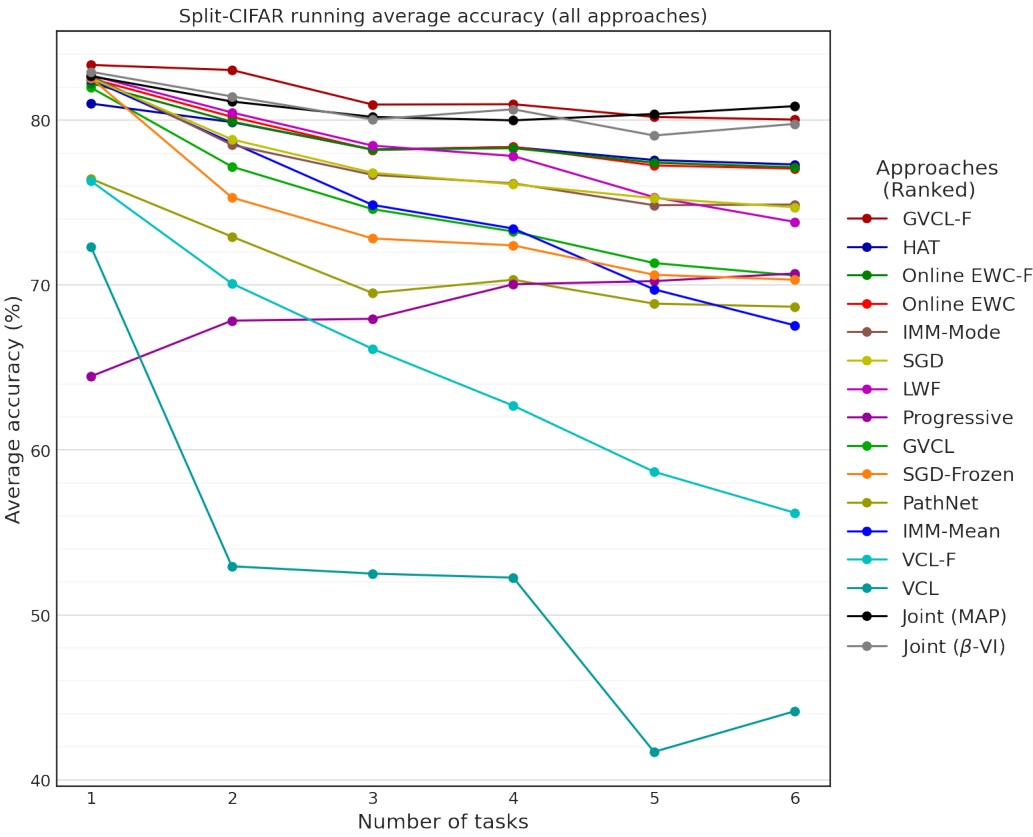

Figure 28: Running average accuracy of individual tasks after training for the all approaches on *Split*-CIFAR

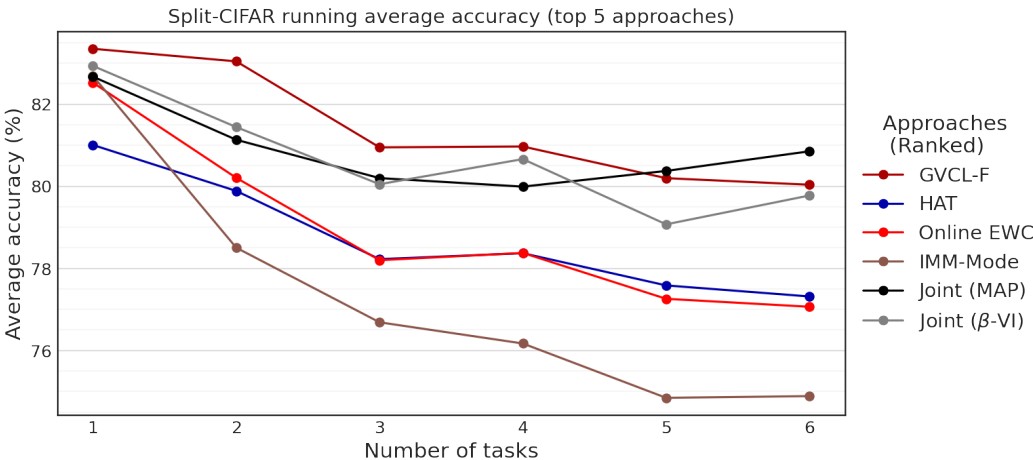

Figure 29: Running average accuracy of individual tasks after training for the top 5 approaches on *Split*-CIFAR

## J.5 MIXED VISION TASKS ADDITIONAL RESULTS

| Metric | ACC (%) | BWT (%) | FWT (%) | NET (%) |
|---|---|---|---|---|
| GVCL-F | **80.0 ± 1.2** | −0.9 ± 1.3 | −4.8 ± 1.6 | **−5.6 ± 1.2** |
| GVCL | 49.0 ± 2.8 | −13.1 ± 1.6 | −23.5 ± 3.4 | −36.7 ± 2.8 |
| | | | | |
| HAT | **80.3 ± 1.0** | −0.1 ± 0.1 | −5.8 ± 1.0 | **−5.9 ± 1.0** |
| PathNet | 76.8 ± 2.0 | 0.0 ± 0.0 | −9.5 ± 2.0 | −9.5 ± 2.0 |
| VCL | 26.9 ± 2.1 | −35.0 ± 5.6 | −23.7 ± 3.8 | −58.8 ± 2.1 |
| VCL-F | 55.5 ± 2.0 | −18.2 ± 2.1 | −11.9 ± 2.4 | −30.1 ± 2.0 |
| Online EWC | 62.8 ± 5.2 | −18.7 ± 5.8 | −4.8 ± 0.7 | −23.4 ± 5.2 |
| Online EWC-F | 70.5 ± 4.0 | −11.8 ± 4.3 | −3.9 ± 0.5 | −15.7 ± 4.0 |
| Progressive | 77.6 ± 0.4 | 0.0 ± 0.0 | −8.6 ± 0.4 | −8.6 ± 0.4 |
| IMM-mean | 53.8 ± 2.0 | −4.4 ± 1.7 | −28.0 ± 3.3 | −32.4 ± 2.0 |
| imm-mode | 36.6 ± 18.7 | −9.1 ± 7.0 | −40.5 ± 11.9 | −49.6 ± 18.7 |
| LWF | 25.8 ± 4.3 | −57.3 ± 4.5 | −3.1 ± 0.6 | −60.4 ± 4.3 |
| SGD | 35.4 ± 3.9 | −50.5 ± 3.9 | **−0.4 ± 0.0** | −50.9 ± 3.9 |
| SGD-Frozen | 52.9 ± 3.9 | 0.0 ± 0.0 | −33.3 ± 3.9 | −33.3 ± 3.9 |
| | | | | |
| Separate (MAP) | 86.3 ± 0.1 | - | - | 0.0 ± 0.0 |
| Separate (β-VI) | 85.7 ± 0.1 | - | - | 0.0 ± 0.0 |
| | | | | |
| Joint (MAP) | 84.3 ± 0.1 | - | - | −2.0 ± 0.1 |
| Joint (β-VI + FiLM) | 83.8 ± 0.2 | - | - | −1.8 ± 0.2 |

Table 8: Performance metrics of GVCL-F, GVCL and various baseline algorithms on Mixed Vision tasks. Separate and joint training results for both MAP and β-VI models are also presented

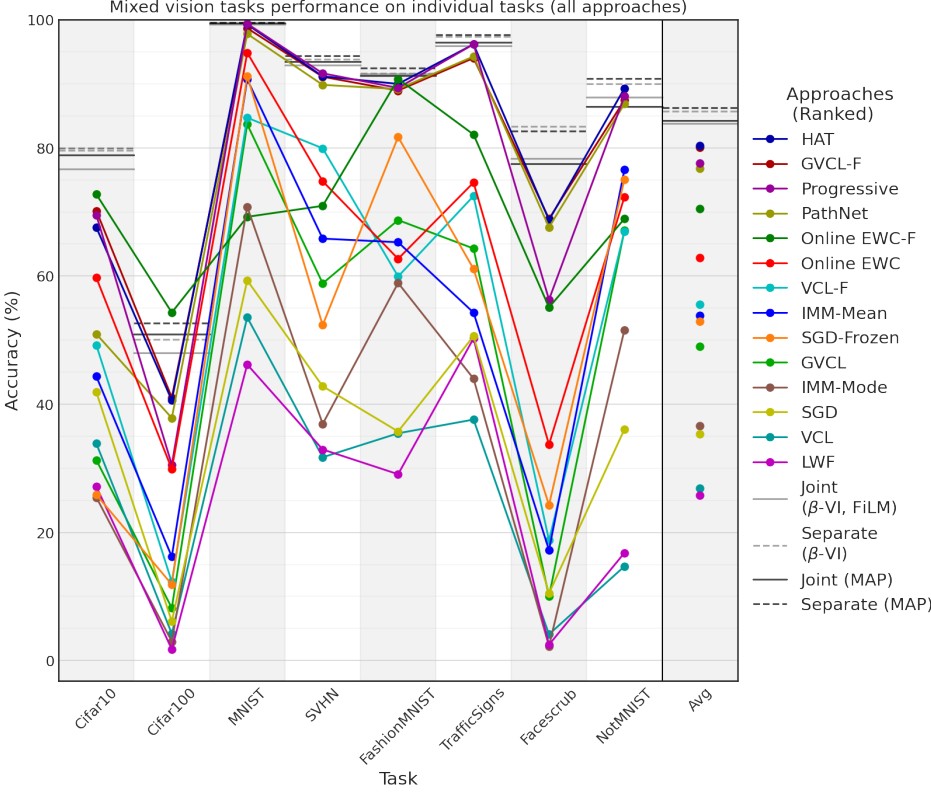

Figure 30: Mean accuracy of individual tasks after training for all approaches on mixed vision tasks

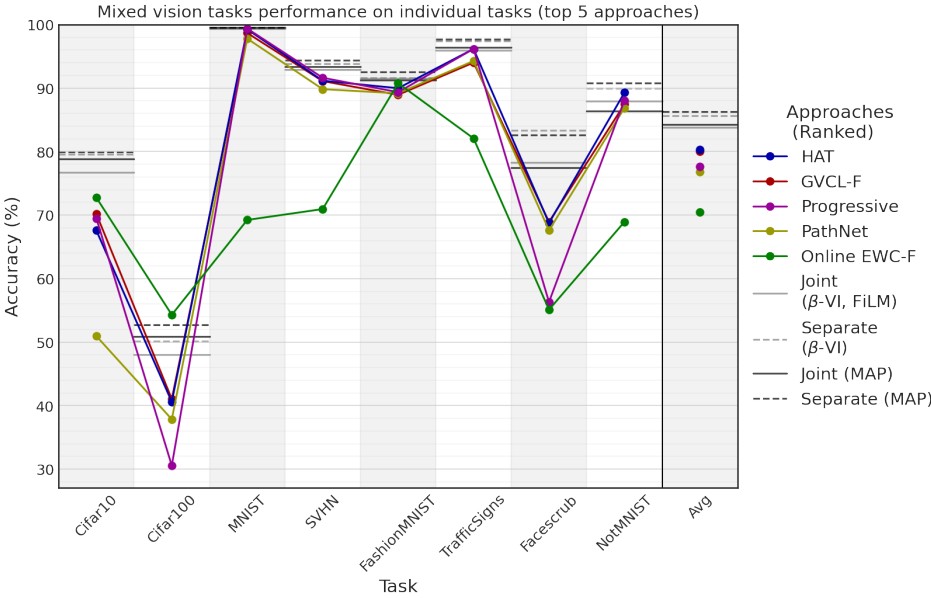

Figure 31: Mean accuracy of individual tasks after training for the top 5 performing approaches on mixed vision tasks

|  | CIFAR10 | CIFAR100 | MNIST | SVHN | F-MNIST | TrafficSigns | Facescrub | NotMNIST | Average |
|---|---|---|---|---|---|---|---|---|---|
| GVCL-F | 0.79% | **0.01%** | **0.04%** | **0.73%** | **0.25%** | **0.10%** | **0.11%** | **0.53%** | **0.32%** |
| HAT | **0.12%** | 0.40% | 0.13% | 2.55% | 0.94% | 0.42% | 5.05% | 3.88% | 1.69% |

Table 9: ECE of all 8 mixed vision tasks for a model trained continually using GVCL-F or HAT. F-MNIST stands for FashionMNIST.

| | | | | | | | | | | | | | | | | | |
|---|---|---|---|---|---|---|---|---|---|---|---|---|---|---|---|---|---|
| Cluster 1 | $\nu$ | $\psi$ | $\gamma$ | $\varphi$ | $\forall$ | $\vartheta$ | $\vee$ | $\Psi$ | $\mathcal{N}$ | $\Omega$ | $\diagup$ | $\Omega$ | $\mathcal{H}$ | $\checkmark$ | $\cup$ | $\sqcup$ | $\heartsuit$ | $\nabla$ |
| Cluster 2 | $\longrightarrow$ | $\rightarrow$ | $\rightarrow$ | $\hookrightarrow$ | $\leftrightarrow$ | $\mapsto$ | $\rightharpoonup$ | $\Rightarrow$ | $\Longleftrightarrow$ | $\twoheadrightarrow$ | $\leftarrow$ | $\sim$ | $\sim$ | $\div$ | $\Rightarrow$ | $\rightleftharpoons$ | $\Leftrightarrow$ | $\dashv$ |
| Cluster 3 | $\tau$ | $\eta$ | $\lceil$ | $\lceil$ | $\Gamma$ | $\Pi$ | $\cap$ | $\Pi$ | $\top$ | $\pi$ | $\Box$ | $\neg$ | $\mathrm{^\circ C}$ | $\uparrow$ | $\rceil$ | $\mathrm{I}$ | | |
| Cluster 4 | $\mid$ | $\mid$ | $\dagger$ | $\lfloor$ | $\parallel$ | $\int$ | $\parallel$ | $\ell$ | $/$ | $\{$ | $\dagger$ | $\backslash$ | $\backslash$ | $\downarrow$ | $\langle$ | | | |
| Cluster 5 | $\mathcal{A}$ | $\Lambda$ | $\wedge$ | $\Delta$ | $\lambda$ | $\triangle$ | $\mathbb{A}$ | $\mathcal{M}$ | $\mu$ | $\chi$ | $\times$ | $\prec$ | | | | | | |
| Cluster 6 | $\therefore$ | $\ddots$ | $\perp$ | $\perp$ | $\cdot\cdot$ | $\cdot$ | $\vdots$ | $\cdot\cdot$ | $\pm$ | $\angle$ | $\mathcal{L}$ | | | | | | | |
| Cluster 7 | $\leq$ | $\cong$ | $\subseteq$ | $\simeq$ | $\lesssim$ | $\equiv$ | $\leqslant$ | $\triangleq$ | $\subsetneq$ | $\preceq$ | $\sqsubseteq$ | | | | | | | |
| Cluster 8 | $\oint$ | $\backprime$ | $f$ | $\$$ | $\xi$ | $\zeta$ | $\mathbb{B}$ | $\beta$ | $\rho$ | $\}$ | $\S$ | | | | | | | |
| Cluster 9 | $\aleph$ | $\alpha$ | $\propto$ | $\bowtie$ | $\ltimes$ | $\infty$ | $\omega$ | $\approx$ | $\ltimes$ | $\rtimes$ | $\mathbb{N}$ | | | | | | | |
| Cluster 10 | $\varnothing$ | $\emptyset$ | $\varnothing$ | $\varnothing$ | $\phi$ | $\Phi$ | $\sharp$ | $\mathbb{H}$ | $\notin$ | $\boxtimes$ | | | | | | | | |
| Cluster 11 | $\Sigma$ | $\sum$ | $\gtrsim$ | $\Xi$ | $\geq$ | $\mathscr{L}$ | $\geqslant$ | $\supseteq$ | $\mathbb{Z}$ | $\exists$ | | | | | | | | |
| Cluster 12 | $\epsilon$ | $\varepsilon$ | $\mathcal{E}$ | $\in$ | $\varrho$ | $\mathbb{C}$ | $\delta$ | $\subset$ | $\mathbb{E}$ | | | | | | | | | |
| Cluster 13 | $\#$ | $\natural$ | $\neq$ | $\star$ | $\hbar$ | $\mathscr{H}$ | $\%$ | $*$ | $\not\equiv$ | | | | | | | | | |
| Cluster 14 | $\circ$ | $\circ$ | $\mathcal{O}$ | $\sigma$ | $\mathbb{G}$ | $\surd$ | $\mathcal{F}$ | $\sigma$ | | | | | | | | | | |
| Cluster 15 | $\sigma$ | $\&$ | $\mathscr{C}$ | $\clubsuit$ | $\mathbb{Q}$ | $\mathring{a}$ | $\ae$ | $\mathbb{R}$ | | | | | | | | | | |
| Cluster 16 | $\Theta$ | $\theta$ | $\odot$ | $\odot$ | $\copyright$ | $\oplus$ | $\wp$ | $\otimes$ | | | | | | | | | | |
| Cluster 17 | $\mathcal{D}$ | $\partial$ | $\ni$ | $\supset$ | $\mathcal{P}$ | $\diamond$ | | | | | | | | | | | | |
| Cluster 18 | $\models$ | $\vDash$ | $\vdash$ | $\kappa$ | $\succ$ | | | | | | | | | | | | | |
| Cluster 19 | $\bullet$ | $\blacksquare$ | | | | | | | | | | | | | | | | |
| Cluster 20 | $\nVdash$ | $\mathbb{1}$ | | | | | | | | | | | | | | | | |

Figure 32: Clusters of symbols found by performing K-means clustering with $K = 20$ based on the embedding layer of a model trained with variational inference on a 200-way classification task on the 200 most common symbols in the HASYv2 dataset. Easy-CHASY is made by taking the first symbol from each cluster as the first task, then the second, and so on, up to 10 tasks. Hard-CHASY is made by taking the clusters with the most classes in order (clusters 1-10).

# K    CLUSTERED HASYV2 (CHASY)

The HASYv2 dataset is a dataset consisting over 32x32 black/white handwritten Latex characters. There are a total of 369 classes, and over 150 000 total samples (Thoma, 2017).

We constructed 10 classification tasks, each with a varying number of classes ranging from 20 to 11. To construct these tasks, we first trained a mean-field Bayesian neural network on a 200-way classification task on the 200 classes with the most total samples. To get an embedding for each class, we use the activations of the second-last layer. Then, we performed K-means clustering with 20 clusters on the means of the embedding generated by each class when the samples of the classes were input into the network. Doing this yielded the classes shown in figure 32. Now, within each cluster are classes which are deemed "similar" by the network. To make the 10 classification tasks, we then took classes from each cluster sequentially (in order of the class whose mean was closest to the cluster's mean), so that each task contains at most 1 symbol from each cluster. Doing this ensures that tasks are similar to one another, since each task consists of classes which are different in similar ways. With the classes selected, the training set is made by selecting 16 samples of each classes, and using the remaining as the test set. This procedure was used to generate the "easy" set of tasks, which should have the maximum amount of similarity between tasks. We also constructed a second set of tasks, the "hard" set, in which each task is individually difficult. This was done by selecting each task to be classification within each cluster, selecting clusters with the most number of symbols first. This corresponds to clusters 1-10 in figure 32. With the classes for each task selected, 16 samples from each class are used in the training set, and the remainder are used as the test set. Excess samples are discarded so that the test set class distribution is also uniform within each task.

It was necessary to perform this clustering procedure as we found it difficult to produce sizable transfer gains if we simply constructed tasks by taking the classes with the most samples. While we were able to have gains of up to 3% from joint training on 10 20-way classification tasks with the tasks chosen by class sample count, these gains were significantly diminished when performing MAP estimation as opposed to MLE estimation, and reduced even further when performing VI. Because one of our benchmark continual learning methods is VCL, showing transfer when trained using VI is necessary.

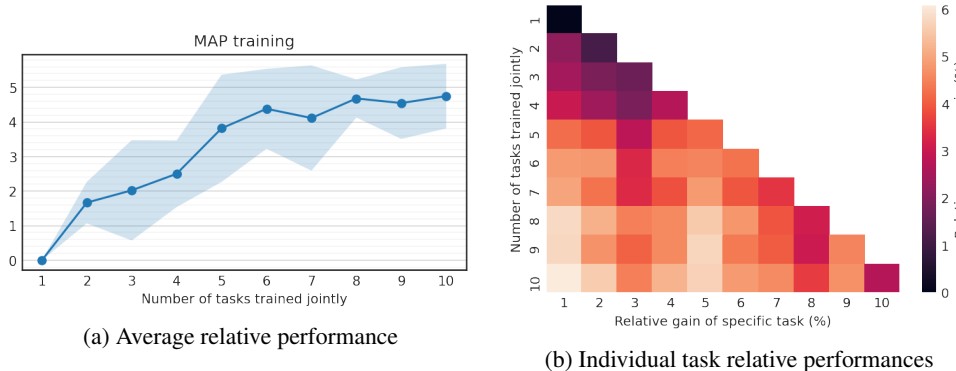

(a) Average relative performance

(b) Individual task relative performances

Figure 33: Relative test-set accuracy of models trained jointly on the easy set of tasks relative to individual training for MAP estimation. Figure 33a shows the means aggregated over all tasks while figure 33b shows the performance differences for individual tasks. Performance increases near monotonically as more tasks are added, achieving an average of around 4.7% gain with 10 tasks

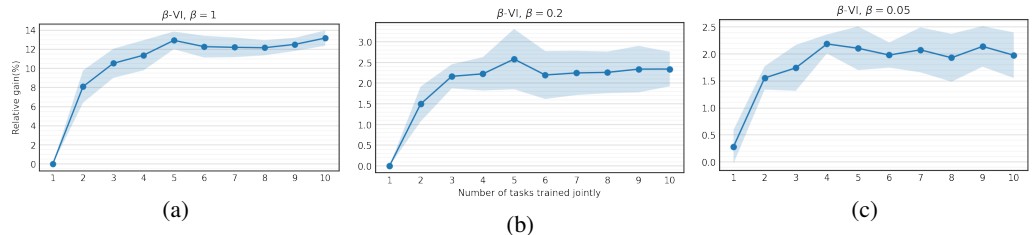

(a)                      (b)                      (c)

Figure 34: Relative performance of models trained jointly on the easy set of tasks relative to individual training for variational inference with various KL-reweighting coefficients $\beta$. Performance gains reach around 2.0% with 10 tasks in the worst case, which is less than with MAP training but still significant

Figures 33a and 34 show the performance gains of joint training over separate training on this new dataset, for both MAP, and KL-reweighted VI, respectively. Figure 33b shows how relative test set accuracy varies for each specific task for these training procedures.

