# OpenReview forum: "Generalized Variational Continual Learning"
_ICLR.cc/2021/Conference — ICLR 2021 Poster_

### Official Review · AnonReviewer3 · 2020-10-24
**An interesting perspective but lack of preciseness and kind of unclear.**

**Rating:** 4
**Confidence:** 5

**Review:**

This paper proposed a generalized variational continual learning (GVCL) framework using the \beta - ELBO, and then combined with FiLM layers. The idea is interesting but there is a lack of preciseness. The pros and cons are as follows.

Pros:
1. The proposed GVCL proposed a different and interesting perspective on the online EWC, viewed as a special case of \beta \to 0;
2. FiLM layers are introduced to combine with GVCL, which lead to significant improvement in the performance;
3. Various experiments are performed, showing some level of advantages.

Cons:
1. The new perspective that online EWC could be viewed as a special case of the GVCL framework is lacking preciseness. First of all, as described in Sec. 2.3, the result of the \beta-ELBO, even with \beta \to 0, does not lead to the key hyper parameter \lambda in online EWC. To compensate this, the authors introduce a modified KL divergence to make them similar. However, it is not justified, from a unified Bayesian or some other theoretical perspective , why the previous \beta-ELBO needs to be modified. It is kind of wired to start from the Bayesian  framework and then go back to the non-Bayesian perspective to design a Bayesian algorithm to improve the performance, and then claim that the previous non-Bayesian algorithm is a special case of the unified Bayesian framework. Moreover, as described in Sec. 2.3, the  resultant GVCL when \beta \to 0 is actually different from the previous online EWC algorithm. As a result, strictly speaking, it is not approperiate to claim that the online EWC could be recovered as a limiting case.

2. If it is true that the proposed GVCL is a generalization of VCL and Online EWC, which allows interpolation between the two, then it is expected and reasonable that the GVCL alone (without additional FiLM layers) should perform at least the same as VCL and online EWC. Otherwise, the statement is not true and there is no advantage of the proposed GVCL framework . However, as shown in experimental results, e.g., Table 1, GVCL alone performs worse than Online EWC in large datasets, which is really wired. The authors also acknowledged this point and claimed that this is due to the difficulty in optimizing GVCL with small \beta. It would be better to make such statement more precise because this is really important point for this paper. Otherwise, it implies that the so-called interpolation between VCL and online EWC has no additional advantage.

3. Regarding the results of the GVCL and GVCL-F, it seems that the improvement mainly comes from the FiLM layers, rather than the GVCL framework itself.  To make this more clear and for a more fair comparison, it is highly suggested to compare other methods (online EWC, VCL, HAT, etc) with FiLM layers. Otherwise, the current improvement of the performance is unclear. In addition, the improvement of GVCL-F over the baseline is not consistent.

---

> ### Author Response · Authors · 2020-11-17
> **Reviewer 3 Response**
>
> Many thanks for your comments. They have helped us improve the paper. We respond to your questions below.
>
> 1. It is kind of weird to start from the Bayesian framework and then go back to the non-Bayesian perspective.
>
> We did not intend to claim that our approach is entirely Bayesian, and our resulting algorithm is not. Rather, we considered this work to be Bayesian inspired: we looked at the Bayesian approach (VCL) and made careful adjustments to fix key issues affecting it. Please also see the general response, point 2 for more discussion of these points.
>
> In the latest version of the paper, we re-interpreted $\lambda$ so that it falls into the general theme of the paper as tempering different parts of the ELBO in light of the cold-posterior effect. Note that tempering is not Bayesian, but rather a commonly-used workaround for dealing with the poor performance of Bayesian methods.
>
> 2. Moreover, as described in Sec. 2.3, the resultant GVCL when $\beta \to 0$ is actually different from the previous online EWC algorithm
>
> We assume you are referring to the difference in $\tilde{\lambda}$ and $\lambda$. These two approaches differ only in their treatment of the prior variance. In the original EWC paper, and the paper proposing Online-EWC, it is unclear what the treatment of the prior covariance should be, hence this difference arises due to an ambiguity in the original algorithm. Furthermore, note that with increasingly small $\beta$, the difference between $\tilde{\lambda}$ and $\lambda$ vanishes, as the prior term becomes negligible compared to the Hessian.
>
> We have changed the text to make this clearer in Section 2.3.
>
> 3. GVCL ... should perform at least the same as VCL and online EWC.
>
> As we mentioned previously, GVCL performs worse than Online EWC sometimes due to optimization issues with convergence as $\beta \to 0$. However, note that GVCL-F always outperforms VCL-F and EWC-F as expected. We also note that there is a benefit to having a unifying framework that encompasses a range of existing approaches allowing them to be better understood.
>
> We have now added a toy example of GVCL in a toy 2d regression dataset showing the convergent behaviour of GVCL to Online EWC. In this toy example, it takes 10 times longer to achieve convergence for very small values of $\beta$ (1e-4) compared to $\beta = 1$. Note that these values of $\beta$ are smaller than we had in our neural network examples, and given that the toy example has only 3 parameters yet still took a very long time to optimize, it is likely that we cannot practically reach the Online EWC limit on a neural network of even modest size.
>
> 4. Regarding the results of the GVCL and GVCL-F, it seems that the improvement mainly comes from the FiLM layers
>
> This is not a correct interpretation of the results -- the improvement comes from both GVCL and FiLM layers, with significant contributions coming from both. We show this explicitly in the new Section 5.4, which as you suggested, looks at the performance gains from adding FiLM layers to GVCL, VCL and Online EWC. It shows for example:
>
> Split CIFAR: moving from VCL to GVCL is 25% gain, adding FiLM layers is a further 11% gain
> Mixed Vision: moving from VCL to GVCL is 24% gain, adding FiLM layers is a further 30% gain
>
> Also note adding FiLM layers to Online EWC only gives 0.1% and 7.7% improvement on these datasets respectively. See Table 2 and General Response point 3 for additional information.
>
> So we see that FiLM layers alone provide little benefit to Online EWC, and a key insight in this paper is how FiLM layers interact with variational methods to fix the pruning issue. Unlike HAT, because of the interaction with the prior, we do not need more complex training procedures to learn the FiLM parameters. Note that we did not include HAT + FiLM layers since it already has per-task channel-wise gating layers.

---

> > ### Comment · AnonReviewer3 · 2020-11-24
> > **Thank you for your response**
> >
> > Thank you for the response with clarifications. So the improvement comes from both GVCL and FiLM layers, rather than the GVCL framework alone. As a result, I think it is essential to highlight this in the main text, or even in the title. Otherwise, it gives one impression that GVCL is the solution.

---

> > > ### Author Response · Authors · 2020-11-24
> > > **Changes to the abstract and introduction**
> > >
> > > Thank you for your response. As you suggested, we have now modified the Abstract and the Introduction in the latest version of the paper. This is to make the importance of the synergy between GVCL and FiLM layers more obvious. For example, the last paragraph of the Introduction now explicitly says that experiments are included with GVCL+FiLM layers, and we added a sentence, “In Section 5.4 we show that FiLM layers provide a disproportionate improvement to variational methods, confirming our hypothesis in Section 3.”

---

### Official Review · AnonReviewer1 · 2020-10-25
**GVCL**

**Rating:** 8
**Confidence:** 5

**Review:**

This paper proposes Generalized Variational Continual Learning (GVCL). It is shown that Online EWC and VCL are special cases of GVCL, along with other theoretical contributions. Further, GVCL is augmented with FiLM to alleviate weaknesses of VCL and GVCL. GVCL and GVCL-F are applied to a number of continual learning tasks and demonstrate competitive performance.

Although GVCL and GVCL-F do not outperform baselines, particularly in hard settings (split-mnist and mixed vision), GVCL is an original and excellent contribution. The paper is clear and well-written, the proposed algorithm is theoretically motivated and analysed, experiments are comprehensive, demonstrating the empirical performance of GVCL.

I have the following comments:
- It would be interesting to have VCL and Online EWC added to Figures 2 and 3.
- Why is GVFL significantly worse than baselines for split-mnist (Figure 2c)?
- Why is split-mnist omitted from Figure 3?
- The supplementary material contains some analysis on the effect and sensitivity of the value of $\beta$ on the performance of the algorithm. This should be extended and presented in the main paper.

Minor:
- "the node is *effective* shut off" -> effectively

---

> ### Author Response · Authors · 2020-11-17
> **Reviewer 1 Response**
>
> Many thanks for your comments. They have helped us improve the paper. We respond to your questions below.
>
> 1. It would be interesting to have VCL and Online EWC added to Figures 2 and 3.
>
> We purposefully included only the best algorithms in Figures 2 and 3 so that the y-axis covers an appropriate range. VCL and Online EWC perform relatively poorly, so we left them out so that it is easy to compare the performance of the better algorithms. We have added additional figures to Appendix K which show these plots, along with the performance of additional algorithms.
>
> 2. Why is GVCL significantly worse than baselines for split-mnist (Figure 2c)?
>
> The baselines in Figure 2c are potentially confusing, and it is unfair to compare vanilla GVCL to them. HAT and GVCL-F store task-specific parameters, and hence have growing memory demands with number of tasks. Therefore the natural comparison for HAT is GVCL-F, and not GVCL (note that Joint-MAP is not a continual learning algorithm). It appears that this is much more important for Split-MNIST than the CHASY benchmarks.
>
> 3. Why is split-mnist omitted from Figure 3?
>
> Many of the tasks all effectively are at 100% performance, so including it does not provide much useful information. However, we have included the plot in Appendix K.
>
> 4. The supplementary material contains some analysis on the effect and sensitivity of the value of $\beta$ on the performance of the algorithm. This should be extended and presented in the main paper.
>
> We have added more explanation in the main text (end of Section 2.2). We have also now added a small toy example showing the convergent behaviour of GVCL to Online EWC in Appendix B, which includes a range of $\beta$ values.

---

### Official Review · AnonReviewer2 · 2020-10-29
**Official Blind Review #2**

**Rating:** 7
**Confidence:** 4

**Review:**

This work considers online variational Bayesian approaches to continual learning. The authors propose a beta-ELBO objective which they claim interpolates between Gaussian variational inference (beta = 1) and Laplace’s approximation (beta = 0).
Furthermore, the authors propose task-specific, non-probabilistic (point estimation) FiLM layers that apply an element-wise transformation to the activations.

Theory / Contribution:
The two contributions seem quite orthogonal to each other and each of them is rather minor in novelty.
It is obvious that using beta=0 leads to MAP estimates from which Laplace’s approximation can be computed. However, I am quite confused what exactly the authors do here and there could be a major mistake:
From the paper, I am not sure if the authors a) compute Laplace’s approximation in the end, at the resulting mean of q, for any beta value? As far as I understand, the authors instead b) only optimise the variance through the beta-ELBO.
However, in this case, the resulting approximation would *not* identical to Laplace’s approximation!
I need clarification what the authors are doing here.
Consider the case of beta=0, the covariance will be the dirac distribution as the authors note in Sec. 2.2 or the supplementary material. The authors then go on and write the optimal covariance matrix for which the derivative of the beta-ELBO is zero.
You have first postulated that the covariance is zero, in order to be able to pull out the expectation, and then you again allow for a non-zero beta-elbo-minimizing covariance. This would be a contraction. This makes me guess you do compute Laplace’s approximation instead. But then it is not discussed how you deal with beta>0.


Related work:
The related work section is rather short mentioning only very few related approaches. More effort is required here.


Experiments:
The experimental evaluation is thorough and seems promising. Although I am wondering why e.g. Fig. 2 does not include VCL and EWC. Figure 8 in the supplementary material probably has some legends mixed up, or the explanations that small beta values cause locally measured locally are wrong? For Fig. a), the largest beta=10 seems to be a good approximation and also the most local. In case of Fig. B) and C) it is unclear / subjective (from visually inspecting the likelihood function) which is the best approximation. In A), beta=0.1 is the least local approximation, in B) beta=10 and in C) beta=1. I cannot follow the intuition provided here.


Summary:
I am sceptical about the correctness regarding the equivalence between VI and Laplace’s approximation; the exact approach proposed in the paper is unclear and may be based on a contradiction. In case I have a misunderstanding here, I hope the authors will point this out and update the manuscript.


Update after Rebuttal:
The authors provided clarifications and improved the manuscript.
In particular, the authors now detail the two special cases (beta=0, beta=1) and how it relates to EWC and VCL.
I am no longer sceptical that the claims regarding the equivalence to EWC in case of beta=0 is correct.
Based on this, I changed my evaluation and now suggest acceptance.

---

> ### Author Response · Authors · 2020-11-17
> **Reviewer 2 Response**
>
> Many thanks for your comments. They have helped us improve the paper. We respond to your questions below.
>
> 1. The two contributions seem quite orthogonal to each other and each of them is rather minor in novelty.
>
> We disagree that these contributions are minor. In this paper, we show that several continual learning algorithms all arise from a single unifying framework, and certain choices and hyperparameters in each algorithm all arise by making different choices in the tempering of posteriors, priors, and likelihoods. For example:
>
> * VCL occurs when no tempering is performed
> * Online-EWC, Online-Structured Laplace and SOLA are instances of GVCL where $\beta \to 0$, and the choice of $Q$ distribution is changed
> * $\lambda$ arises by tempering the posterior and prior using the same temperature in the KL-divergence
> * Online-EWC’s $\gamma$ arises by tempering the posterior and prior using different temperatures in the KL-divergence
>
> A unifying algorithm immediately opens the door for different choices of these parameters, and lets us understand the relationship between and broader context of these algorithms. Naturally, it means that improvements and innovations in one of these algorithms can readily be applied to others and paves the way for rapid and systematic progress.
>
> Our second contribution, the usage of FiLM layers, addresses a key limitation in variational methods, which is particularly problematic in the continual learning setting. While this contribution is orthogonal to GVCL, it is particularly synergistic. What differs between our version of FiLM layers and other similar algorithms, such as HAT, is its synergy with variational methods. Because of the pruning effect and the prior, no special algorithm is needed to fit these FiLM layers, and the resulting gain for variational methods is over 10%, compared to merely 2% for non-variational methods. We have added a new section in the revised version of the paper to make this clear (Section 5.4).
>
> 2. I am not sure if the authors a) compute Laplace’s approximation in the end, at the resulting mean of q, for any beta value?
>
> This is a misunderstanding. We have responded to this point in the general response (point 1), but add further clarification below.
>
> To be clear, we never compute Laplace’s approximation directly. We update $\Sigma$ using the $\beta$-ELBO and show that this recovers a version of Laplace’s approximation in a limiting case. In the derivation of this result, we did not assume $\beta = 0$, but rather assume that $\beta$ is very close to zero as we take the limit. When this is done, there is a cancellation in the $\beta$-ELBO whereby the beta-dependence in the previous posterior cancels with the beta term in the $\beta$-ELBO, giving rise to the EWC regularisation (see equations 10 and 11 in Appendix C).
>
> In our derivation, we did not assume the covariance was zero, but we assumed it was $\textit{near}$-zero, so we can still apply normal arithmetic to it.
>
> We agree that in Appendix C, the statement that “q approaches a delta function” was rather imprecise. To amend this, we have now added a proof that moving from Equation 8 to 9 is valid for small $\beta$. This is included in Appendix C.1.
>
> 3. Related work: The related work section is rather short mentioning only very few related approaches. More effort is required here.
>
> Note that many related works are mentioned previously in the text (e.g. 16 unique texts in sections 2 and 3), not only in the Related Works section. Also, we have a longer related work section in Appendix I, which we were unable to include in the main text due to space constraints. As we now have additional space (from gaining an additional page), we are happy to expand this section in the main text. Please let us know of the specific references you have in mind.
>
> 4. I am wondering why e.g. Fig. 2 does not include VCL and EWC
>
> We have included the requested plots in Appendix J. We only included the best performing algorithms in figure 2 since VCL and EWC performance lies well below the range of the graph.

---

> > ### Author Response · Authors · 2020-11-17
> > **Reviewer 2 Response Part 2**
> >
> > 5. Figure 8 in the supplementary material probably has some legends mixed up, or the explanations that small beta values cause locally measured locally are wrong
> >
> > The legends are correct, and there is likely some misunderstanding here. We clarify that by “local” we refer to the immediate vicinity around a point, i.e. if we zoomed in very close to a point. We also reiterate that a “good” and a “local” approximation are not to be conflated, particularly in the toy examples presented.
> >
> > For Figures 8b and 8c, there is a cusp at the mode of the distribution. Therefore, a local approximation would approximate the function immediately surrounding that cusp, and ignore the regions far away. We see that this is exactly what $\beta = 0.1$ (very small $\beta$) does: it has a very sharp curve. Similarly, for Figure 1, if zoomed in very close to the mode of (a), we would notice that the true distribution is nearly flat. This means that a local fit would match that flatness, and ignore the fact that it begins to curve further away from the mode. $\beta = 0.1$  does exactly this. It appears to be a bad fit because of the scale of our graphs, but if we zoomed in very closely to the mode, it would appear to be the best fit, because it is the most local fit.

---

> > > ### Comment · AnonReviewer2 · 2020-11-17
> > > **Thanks for clarification, one more potential misunderstanding to be clarified**
> > >
> > > Thanks for the clarification. Fig. 8 now makes sense to me.
> > >
> > > Regarding 2), I am still trying to understand whether or not the variance resulting from optimizing the beta-ELBO is the same as the variance you would obtain when computing the Hessian at mu_t.
> > >
> > > Lets start again with i) beta=0 (exactly zero) and then again ii) beta --> 0 (close to zero).
> > > In i), it is clear that we will always obtain just the MLE estimate (not even MAP) with 0 covariance in every step, as we will always completely ignore the KL term. Now I don't see how for ii) the covariance suddenly jumps to the correct covariance for an infinitesimal change.
> > >
> > > Lets try to get there through your results in App. C.
> > > Step 1) dL/dSigma = 0: We obtain an optimal (local minimum with derivative zero) precision matrix in (9) under the condition that beta is close to zero. I also agree with the resulting recursion of the precision.
> > > Note: So far, we have not concluded whether or not this optimal precision matrix is the same as we would get from Laplace's approximation. It is just whatever we get from optimizing the beta-ELBO, and in case of beta=0, precision will be inf.
> > > Step 2) Eq. (10) now looks at the ELBO when we use the previous posterior resulting from optimizing the beta-ELBO.
> > > Now, I agree that if beta is only *close* to zero, it will cancel (for the Hessian).
> > > And I also agree that the beta-ELBO then looks like the optim. from Laplace propagation.
> > > But that is only for optimizing mu!
> > > So I agree also that each iteration (time-step) t should in theory find the same mu_t as we would for Laplace Propagation.
> > > However, my understanding is that we will have a too sharp posterior at t-1 and then because we down-weight the KL term through beta, it will act as if it had the correct variance. So in the limiting case, we have a dirac distribution and it is weighted with zero, canceling completely.
> > >
> > > In other words, the resulting posterior approximation at every iteration is not identical to Laplace's approximation, but the mean is. This means that posterior predictives will not use the right posterior (although we could apply Laplace's approximation just for posterior predictives while doing continual learning as proposed).
> > >
> > > Do you think there is still a misunderstanding or would you agree?

---

> > > > ### Author Response · Authors · 2020-11-17
> > > > **Clarification of misunderstanding with Online-EWC**
> > > >
> > > > Thank you for your quick response, and for your time.
> > > >
> > > > You are correct when you state that the resulting posterior is not identical to that recovered using Laplace’s approximation: the means are the same, but the covariance is near-zero. Therefore the approximate posterior over the weights has no uncertainty. This is in fact exactly how Online EWC performs predictions (Schwarz et al., 2018, “Progress & Compress:  A scalable framework for continual learning”, Section 4). That is, predictions at test-time are made with deterministic weights (no uncertainty), and the Hessian / the link to Laplace’s approximation is only used for updating this mean parameter value.
> > > >
> > > > If Laplace’s approximation is used to form a non-deterministic posterior which is used to make predictions, then that is a slightly different algorithm to Online EWC, which we agree does not fall under the GVCL family.

---

> > > > > ### Comment · AnonReviewer2 · 2020-11-17
> > > > > **posterior approximation**
> > > > >
> > > > > Thanks for the clarifications!
> > > > >
> > > > > Laplace propagation and Online Structured Laplace Approximations (Ritter et. al) do compute the Hessians though.
> > > > > I understand now that you do not need to compute the Hessians for learning continually because of the cancellation. However for the posterior predictive distribution, computing the "right" covariance would be necessary.
> > > > > I would have expected that the variational resulting distribution from optimizing the ELBO is a posterior approximation, which is the case for VCL and for Laplace's approximation, but for GVCL this is arguably not the case.
> > > > > I think this should be discussed.
> > > > >
> > > > > I suppose, we could do a Laplace's approximation in GVCL as well, if we compute the Hessian at mu, ignoring the learnt variance (which does not correspond to either variational or Laplace approximation).

---

> > > > > > ### Author Response · Authors · 2020-11-18
> > > > > > **GVCL Posterior**
> > > > > >
> > > > > > Thanks again for your response.
> > > > > >
> > > > > > 1. In the next revision, we will add a discussion of how predictions are performed in Section 2.2 noting that the VCL predictive is recovered when $\beta=1$ and the Online EWC predictive is recovered when $\beta \to 0$ as the weight uncertainty disappears. In the paper, we typically use values of $\beta$ between 0.2 and 0.05, so there is still some uncertainty retained.
> > > > > >
> > > > > > 2. It is not clear to us whether Ritter et al. do use weight uncertainties when making predictions e.g. in section 2, paragraph 1 (https://papers.nips.cc/paper/2018/file/f31b20466ae89669f9741e047487eb37-Paper.pdf),
> > > > > > the paper says their aim is to find a MAP estimate to the posterior over all datasets. This would imply that they do not predict with uncertainty for *all* of the algorithms, including e.g. the one called "Online Laplace", even though this might be confusing. Furthermore, they describe EWC as “approximat[ing] the posterior .. with a Gaussian” (section 3 paragraph 2), but EWC also does not use weight uncertainty either, casting doubt on whether they use uncertainty too.  Moreover, the only (non-official) implementation we have found also only uses deterministic weights (https://github.com/hannakb/KFA).
> > > > > >
> > > > > > Indeed, generally, if you use Laplace's approximation for Bayesian neural networks with Monte Carlo sampling for forming the predictive, you get very poor results (see e.g. https://arxiv.org/abs/1906.11537). So you have to either temper the posterior by a large amount (e.g. by removing all uncertainty) or use the linearisation approximation discussed in the above paper.

---

> > > > > > > ### Comment · AnonReviewer2 · 2020-11-19
> > > > > > > **review adjust scores**
> > > > > > >
> > > > > > > Thanks again for further clarifications.
> > > > > > > I will read the paper again (after the revision) and update my scores.

---

> > > > > > > > ### Author Response · Authors · 2020-11-23
> > > > > > > > **New revision with clarifications**
> > > > > > > >
> > > > > > > > We have now updated the paper to include the following discussion of the predictive distribution at the end of section 2.2:
> > > > > > > >
> > > > > > > > When performing inference with GVCL at test time, we use samples from the unmodified $q(\theta)$ distribution. This means that when $\beta = 1$, we recover the VCL predictive, and as $\beta \to 0$, the posterior collapses as described earlier, meaning that the weight samples are effectively deterministic. This is in line with the inference procedure given by Online EWC and its variants.
> > > > > > > > In practice, we use values of $\beta = 0.05 - 0.2$ in Section 5, meaning that some uncertainty is retained, but not all. We can increase the uncertainty at inference time by using an additional tempering step, which we describe, along with further generalizations in Appendix D.
> > > > > > > >
> > > > > > > > If you have any suggestions, we are happy to modify the paragraph, or add more details in one of the appendices.
> > > > > > > > The other changes in the latest revision are given in the latest general response.

---

### Official Review · AnonReviewer4 · 2020-11-02
**Interesting take on Unifying VCL and EWC**

**Rating:** 7
**Confidence:** 4

**Review:**

The authors propose Generalized VCL in this paper, which consists of multiple ideas: first, the authors introduce a beta-Elbo, which facilitates downweighting the KL-term of VCL. If beta taken to the limit towards zero, the authors show that the beta-elbo recovers the online EWC learning criterion, which draws an interesting link between VCL and EWC.
The authors also discuss reweighting terms to introduce a parameter lambda as in EWC, which they incorporate via a lambda-kl divergence term.
Finally, furnished with this learning objective that interpolates between VCL and EWC, the authors propose to combine the learning objective with the architectural choice of Film layers, which they show facilitate overcoming the pruning behavior that their method inherits from VCL by offering ways to prune nodes without injecting noise into the network.

Experiments are broad on multiple interesting datasets and quite clearly show that their proposed combined model performs best.

Positives:
The paper draws an interesting unification between EWC and VCL, and in fact also other related works, as subtle modifications in a regularizer. This by itself is an interesting contribution. The fact that the authors study the interplay of their learning arlgorithm with architectural biases, i.e. overcoming early pruning via film layers, is also a valuable idea that I find not just interesting in itself, but also stylistically valuable as an approach to studying  deep learning. While the Film layers per se also appear somewhat ad hoc, their empirical benefits -particuarly when paired with the lambda-elbo, are impressive and well put together.

Criticisms:
While I really enjoy the derivation of the beta-elbo in the zero limit, I found the introduction of the reweighting terms in Sec. 2.3 to be ad hoc and not particularly well justified. It feels as if it is reverse engineered to match the desired criterion from EWC. I think the authors should dig deeper here for better justifications for such choices, as they did a good job having a mathematically interesting framework to derive earlier.

Additionally, the film layers work great, but I maybe missed if they are the main attraction powering performance or if it is the combination with the new ELBO. Would film layers with VCL do equally well? This is empirically confusing, it would be great to get some more help to understand the relative merits of each components here and clarify more how these pieces fit together empirically. I do enjoy the appendix discussing this qualitatively, but I would like to understand it quantitatively better, as theoretically film layers plus VCL (without this paper's innovations) should also benefit similarly.

One additional criticism is that the title is somewhat misleading, as it does not generalize VCL to broader settings, but rather collapses it towards the limit beta towards zero. The title raised hopes for a richer variational treatment rather than a unification to EWC and an architecture change. The authors might want to consider tweaking the title to sth that is closer to the paper's actual contributions.


Overall:
This paper takes an interesting approach towards adding to the EWC and VCL literature by unifying them and offering an architectural fix for a key problem in these scenarios. While the contributions are mixed and not consistently derived from clear modeling assumptions, their interplay is well studied and highly relevant to the understanding and improvement of practical continual learning. I also want to again applaud the authors for studying and explaining the interplay of pruning and film layers, I enjoyed reading the supplementary information on this. I wish more papers that discover methods that perform well empirically would study the interplays of algorithm and architecture similarly to expose interesting effects.

---

> ### Author Response · Authors · 2020-11-17
> **Reviewer 4 Response**
>
> Many thanks for your comments. They have helped us improve the paper. We respond to your questions below.
>
> 1. I found the introduction of the reweighting terms in Sec. 2.3 to be ad hoc and not particularly well justified...I think the authors should dig deeper here for better justifications for such choices…
>
> We agree that the writing in Section 2.3 could be improved and have re-written this section connecting the introduction of the parameter $\lambda$ to the literature on cold posteriors. Please also see the general response, point 2 for more information.
>
> 2. ”Additionally, the film layers work great, but I maybe missed if they are the main attraction powering performance or if it is the combination with the new ELBO. Would film layers with VCL do equally well?”
>
> We have added Section 5.4, which shows the relative performance gain from adding FiLM layers to VCL and Online EWC. We see that GVCL and VCL both see large performance gains while Online EWC only receives marginal gains. This suggests that FiLM layers are particularly synergistic with VI based methods. Additionally, GVCL+FiLM outperforms VCL+FiLM.
>
> 3. The title is somewhat misleading
>
> We selected the title as the paper generalizes several existing continual learning algorithms under a single variational framework, related by the choice of Q distribution class and choices related to the tempering of distributions. We are happy to consider alternative titles and would be open to hear your suggestions.

---

> > ### Comment · AnonReviewer4 · 2020-11-23
> > **Thank you for your response**
> >
> > Thank you for your efforts to address my comments and to improve the paper.
> >
> > 1.
> > I find the link to cold posteriors interesting, it would explain the choices more consistently from an inferential point of view. This analysis can also be deepened somewhat in future work, as the importance of 'cold' posteriors seems under-explored in the context of continual learning.
> >
> > 2.
> > Thank you for clarifying that. I suspected FiLM layers would also work well with VCL itself in this case, I am glad this gets confirmed, as it may inform users running VCL systems that structure akin to FiLM layers may immediately help improve their systems. It is also quite interesting that GVCL outperforms VCL when using FiLM layers. I think this is overall a valuable addition to the paper and makes the case for this 'second' idea in the manuscript more succinctly.
> >
> > 3.
> > I would be presumptuous to suggest a title and I exclude this from my evaluation of the paper of course, but my mental hash function for this manuscript stored it under 'unifying VCL and EWC' rather than 'generalizing VCL' as the suggested changes are not strictly advances in the field of approximate inference.  I don't know if my hash function is useful enough to the authors for considering titles that would more directly inform the reader about the content, it is anecdotal.
> >
> > Overall, I remain convinced this is a strong paper and continue to suggest acceptance.

---

### Author Response · Authors · 2020-11-17
**General Response**

We thank all the reviews for their thoughtful criticisms and suggestions. Here, we address the main points raised by reviewers and outline the changes we have made to the paper in response. More specific points are addressed in the response to each of the individual reviewers.


1. Reviewer 2 has concerns about the mathematical limit that connects generalized variational inference to Laplace’s approximation

The reviewer has misunderstood the theory in our paper associated with the limit $\beta \to 0$. We briefly outline the key result:

* Consider running GVCL with a common $\beta$ value used across all tasks
* Now take the limit of this procedure as $\beta$ tends to zero
* In this case, all the approximate posteriors (qs) limit to deltas around the MAP estimate
* The inverse variances (precisions) of these approximate posteriors tend to sums of the Hessians at the MAP value scaled by 1/$\beta$
*Critically, the objective functions for each task become equal to online EWC due to a cancellation of the terms involving beta

We have improved the discussion of this limit by adding more detail to Appendix C and C.1, in particular we justify the argument that for small $\beta$ the expectation in Equation 8 becomes approximately the Hessian at the mean (Equation 9).


2. Reviewers 3 and 4 are worried that the introduction of $\lambda$ -- which is necessary to recover Online EWC in a general way -- is not theoretically-well justified

It is true that, from a Bayesian perspective, it is not straightforward to justify the introduction of the parameter $\lambda$. In the revised version of the paper, we add a theoretical explanation for reweighting the quadratic term with $\lambda$ as well (Section 2.3).

In the re-written Section 2.3, we show that $\lambda$ arises if we make use of tempering, as has been proposed in the context of cold posteriors (Wenzel et al. 2020). Specifically, at each step we temper the previous posterior before applying variational inference. We believe that this new interpretation sheds light on the relationship between the effectiveness of $\lambda$ and cold posteriors.

We would also like to reiterate that our final algorithm cannot be strictly considered “Bayesian,” nor do we claim it to be. Rather, there is a general trend in the Bayesian Deep Learning community whereby Bayesian methods are used to develop new algorithmic approaches to deep learning problems and then relaxations of these approaches are considered, with additional parameters, that perform better empirically than the pure-Bayesian method. EWC was developed using this approach (and indeed this resulted in the same $\lambda$ parameter being introduced without rigorous justification) and more recent work has followed this example (Kirkpatrick et al. 2016, Ritter et al. 2018, Osawa et al. 2019 , Pan et al. 2020, Wenzel et al. 2020,  Higgins et al 2017, Alemi et al. 2017). This class of approaches has been called ‘Bayesian Inspired’ and we see the current work as belonging to this pragmatic vein.

In this paper we take the more strictly Bayesian VCL algorithm, and improve it by addressing the main shortcomings of variational Bayesian methods. Namely, we address the poor data fit problem by considering tempered likelihoods with $\beta$ and $\lambda$, and fix the pruning issue using FiLM layers.


3. Reviewers 3 and 4 question “whether the performance gains are from FiLM Layers or GVCL”

We have added results on all benchmarks with Online EWC + FiLM layers, and VCL + FiLM layers in section 5.4. These results show that (i) FiLM layers provide a significant benefit to the variational algorithms (VCL and GVCL), while not so much for Online EWC, (ii) GVCL + FiLM outperforms all competing algorithms, with both innovations contributing to the improved performance.

To summarise, in the revised Section 5.4 in Table 2 and Appendix J we see:
* VCL+FiLM >> VCL and GVCL+FiLM >> GVCL, while EWC + FiLM $\approx$ EWC.
* GVCL+FiLM > GVCL > VCL+FiLM >> VCL


4. AnonReviewers 1 and 2 have some questions about the omission of certain baseline algorithms from figures

For Figures 2 and 3, we only included the GVCL, GVCL-F, and the top performing baseline algorithm. This was done to keep the figure uncluttered, and to keep the y-axis in a reasonable range as these baseline algorithms perform poorly. However, as requested by reviewers, we have now added some extra figures in Appendix J.

---

> ### Author Response · Authors · 2020-11-17
> **General Response Part 2**
>
> Here we note the changes we have made to the revised version of the paper:
>
> 1. A derivation of the quadratic term multiplier based on tempering the posterior and prior in Section 2.3.
> 2. Section 5.4 (and Table 2), which includes performance gains from adding FiLM layers to EWC and VCL.
> 3. Appendix B, which empirically shows the convergence of GVCL to Online EWC in a toy example and highlights the difficulty of 4. achieving this limit in practice.
> 5. Appendix C.1, which includes a proof that the delta-function argument used in Equation 8 to 9 is valid.
> 6. Appendix C.3, which shows how Online-EWC’s $\gamma$ arises by tempering the posterior and prior in the KL-divergence by slightly different temperatures
> 7. Additional results of EWC + FiLM and VCL + FiLM added to the full result tables in Appendix J, as well as additional figures showing each algorithm’s performance on each task (similar to Figures 2 and 3).

---

> > ### Author Response · Authors · 2020-11-23
> > **23/11 General Response**
> >
> > In the latest revision, we made some minor changes to some figures and section 2.2, which we outline here:
> >
> > 1. Added a paragraph clarifying how predictions are made with GVCL at the end of section 2.2
> > 2. Updated outdated figures in appendix K which describe the easy-CHASY benchmark
> > 3. Fixed an error in the newly added plots from the last revision in appendix J where the joint training lines were plotted wrong
> > 4. fixed a problem with the SGD, Separate, SGD-Frozen, and Joint (MAP) training on CHASY datasets where early stopping was performed on slightly the wrong criteria. This affects table 1 and figure 2ab (in the backwards and forwards transfer metrics by ~0.2% and the plot of the joint training line). The change is very minor and does not affect our analysis or conclusions
> >
> > 24/11 Small revision:
> > 1. Modified the abstract and introduction so that it is more clear that the improvements are from both GVCL and FiLM layers

---

### Decision · Program_Chairs · 2021-01-07
**Final Decision**

**Decision:**

Accept (Poster)

**Comment:**

Three of four reviewers are in favour of accepting the paper. Some reviewers raised valid criticism regarding the derivations, interpretation of the mathematical analysis and experimental results. So clearly some aspects of the paper could and should be clarified in accordance with the points raised by the reviewers. However, all in all the paper contains enough contributions to warrant publication.